# CROSS-DOMAIN POLICY OPTIMIZATION VIA BELLMAN CONSISTENCY AND HYBRID CRITICS

**Ming-Hong Chen**[1]* **Kuan-Chen Pan**[1]* **You-De Huang**[1]* **Xi Liu**[2] **Ping-Chun Hsieh**[1]

[1]National Yang Ming Chiao Tung University, Hsinchu, Taiwan
[2]Applied Machine Learning, Meta AI, Menlo Park, CA, USA
{mhchen1224.cs12, pinghsieh}@nycu.edu.tw

## ABSTRACT

Cross-domain reinforcement learning (CDRL) is meant to improve the data efficiency of RL by leveraging the data samples collected from a source domain to facilitate the learning in a similar target domain. Despite its potential, cross-domain transfer in RL is known to have two fundamental and intertwined challenges: (i) The source and target domains can have distinct state space or action space, and this makes direct transfer infeasible and thereby requires more sophisticated inter-domain mappings; (ii) The transferability of a source-domain model in RL is not easily identifiable a priori, and hence CDRL can be prone to negative effect during transfer. In this paper, we propose to jointly tackle these two challenges through the lens of *cross-domain Bellman consistency* and *hybrid critic*. Specifically, we first introduce the notion of cross-domain Bellman consistency as a way to measure transferability of a source-domain model. Then, we propose $Q$Avatar, which combines the Q functions from both the source and target domains with an adaptive hyperparameter-free weight function. Through this design, we characterize the convergence behavior of $Q$Avatar and show that $Q$Avatar achieves reliable transfer in the sense that it effectively leverages a source-domain Q function for knowledge transfer to the target domain. Through experiments, we demonstrate that $Q$Avatar achieves favorable transferability across various RL benchmark tasks, including locomotion and robot arm manipulation. Our code is available at https://rl-bandits-lab.github.io/Cross-Domain-RL/.

## 1 INTRODUCTION

Cross-domain reinforcement learning (CDRL) serves as a practical framework to improve the sample efficiency of RL from the perspective of transfer learning, which leverages the pre-trained models from a source domain to enable knowledge transfer to the target domain, under the presumption that the data collection and model training are much less costly in the source domain (*e.g.,* simulators). A plethora of the existing CDRL methods focuses on knowledge transfer across environments that share the same state-action spaces but with different transition dynamics. This setting has been extensively studied from a variety of perspectives, such as reward augmentation (Eysenbach et al., 2021; Liu et al., 2022), data filtering (Xu et al., 2023), and latent representations (Lyu et al., 2024). Despite the above progress, to fully realize the promise of CDRL, there are two fundamental challenges to tackle: (i) *Distinct state and/or action spaces between domains*: To support flexible transfer across a wide variety of domains, the generic CDRL is required to address the discrepancies in the state and action spaces between source and target domains. Take robot control as an example. One common scenario is to apply direct policy transfer between robot agents of different morphologies (Zhang et al., 2021), which naturally leads to a discrepancy in representations. This discrepancy significantly complicates the transfer of either data samples or learned source-domain models. (ii) *Unknown transferability of a source-domain model to the target domain*: CDRL conventionally presumes that the source-domain model can achieve effective transfer under a properly learned cross-domain correspondence. However, in practice, given that the data budget of the target domain is limited, it is rather difficult to determine a priori the transferability of a source-domain model. Indeed, it has

---

*Equal contribution.

been widely observed that transfer learning from the source domain can have a negative impact on the target domain (Weiss et al., 2016; Pan & Yang, 2009).

As a consequence, despite that CDRL has been shown to succeed in various scenarios, without a proper design, the performance of CDRL could actually be much worse than the vanilla target-domain model learned without using any source knowledge. Notably, to tackle (i), several approaches have been proposed to address such representation discrepancy by learning state-action correspondence, either in the typical RL (You et al., 2022) or unsupervised settings (Zhang et al., 2021; Gui et al., 2023). However, existing solutions are all oblivious to the issues of model transferability between the domains. Hence, one fundamental research question about CDRL remains largely open:

*How to achieve effective transfer in CDRL under distinct state-action spaces without the knowledge of the transferability of the pre-trained source-domain model?*

In this paper, we affirmatively address the above question by revisiting cross-domain state-action correspondence through the lens of *cross-domain Bellman consistency*, which quantifies the transferability of a source-domain model. To enable reliable transfer across varying levels of source-model transferability, we introduce a novel CDRL framework, *QAvatar*, which integrates source-domain and target-domain critics. Drawing an analogy from the movie *Avatar*, where humans remotely control genetically engineered bodies to adapt to alien environments, *Q*Avatar updates the target-domain policy via a weighted combination of the target- and source-domain Q functions, while learning the state-action correspondence by minimizing a cross-domain Bellman loss.

To validate this idea, we first present a tabular prototype of *Q*Avatar and show that it attains a tight sub-optimality bound under an adaptive, hyperparameter-free weight function, regardless of source model transferability. This ensures improved sample efficiency while avoiding poor transfer. Building on this, we develop a practical version by combining *Q*Avatar with a normalizing flow-based mapping for learning state-action correspondence.

The main contributions of this paper can be summarized as follows: 1) We propose the *Q*Avatar framework that achieves knowledge transfer between two domains with distinct state and action spaces for improving sample efficiency. We then present a prototypical *Q*Avatar algorithm and establish its convergence property. 2) We further substantiate the *Q*Avatar framework by proposing a practical implementation with a normalizing-flow-based state-action mapping. This further demonstrates the compatibility of *Q*Avatar with off-the-shelf methods for learning state-action correspondence. 3) Through experiments and an ablation study, we show that *Q*Avatar outperforms the CDRL benchmark algorithms on various RL benchmark tasks.

## 2 RELATED WORK

**CDRL across domains with *distinct* state and action spaces.** The existing approaches can be divided into two main categories: (i) *Manually designed latent mapping*: In (Ammar & Taylor, 2012; Gupta et al., 2017; Ammar et al., 2012), the trajectories are mapped manually from the source domain and the target domain to a common latent space. The distance between latent states can then be calculated to find the correspondence of the states from the different domains. (ii) *Learned inter-domain mapping*: In (Taylor et al., 2008; Zhang et al., 2021; You et al., 2022; Gui et al., 2023; Zhu et al., 2024), the inter-domain mapping is mainly learned by enforcing dynamics alignment (or termed dynamics cycle consistency in (Zhang et al., 2021)). Additional properties have also been incorporated as auxiliary loss functions in learning the inter-domain mapping, including domain cycle consistency (Zhang et al., 2021), effect cycle consistency (Zhu et al., 2024), maximizing mutual information between states and embeddings (You et al., 2022) However, the existing approaches all presume that the domains are sufficiently similar and do not have any performance guarantees. By contrast, we propose a reliable CDRL method that can achieve transfer regardless of source-domain model quality or domain similarity with guarantees.

**CDRL across domains with *identical* state and action spaces.** Various methods have been proposed for the case where source and target domains share the same state and action spaces but are subject to dynamics mismatch. Existing methods include (i) using the samples from both source and target domains jointly for learning (Eysenbach et al., 2021; Liu et al., 2022; Xu et al., 2023; Lyu et al., 2024), (ii) explicit characterization of domain similarity (Behboudian et al., 2022; Sreenivasan et al.,

2023), and (iii) using both Q-functions for Q-learning updates (Wang et al., 2020). However, given the assumption on identical state-action spaces, they are not readily applicable to our CDRL setting.

## 3  PRELIMINARIES

In this section, we provide the problem statement and basic building blocks of CDRL as well as the useful notation needed by subsequent sections. For a set $\mathcal{X}$, we let $\Delta(\mathcal{X})$ denote the set of probability distributions over $\mathcal{X}$. As in typical RL, we model each environment as an infinite-horizon discounted Markov decision process (MDP) denoted by $\mathcal{M} := (\mathcal{S}, \mathcal{A}, P, r, \gamma, \mu)$, where (i) $\mathcal{S}$ and $\mathcal{A}$ represent the state space and action space, (ii) $P : \mathcal{S} \times \mathcal{A} \rightarrow \Delta(\mathcal{S})$ denotes the transition function, (iii) $r : \mathcal{S} \times \mathcal{A} \rightarrow [0, 1]$ is the reward function (without loss of generality, we presume the rewards lie in the $[0, 1]$ interval), (iv) $\gamma \in [0, 1)$ is the discounted factor, and (v) $\mu \in \Delta(\mathcal{S} \times \mathcal{A})$ denotes the initial state-action distribution. Notably, the use of an initial distribution over states and actions is a standard setting in the literature of natural policy gradient (NPG) (Agarwal et al., 2021a; Ding et al., 2020; Yuan et al., 2022; Agarwal et al., 2020; Zhou et al., 2024). Given any policy $\pi : \mathcal{S} \rightarrow \Delta(\mathcal{A})$, let $\tau = (s_0, a_0, r_1, \cdots)$ denote a (random) trajectory generated under $\pi$ in $\mathcal{M}$, and the expected total discounted reward under $\pi$ is $V^\pi_\mathcal{M}(\mu) := \mathbb{E}[\sum_{t=0}^\infty \gamma^t r(s_t, a_t) | \pi; s_0, a_0 \sim \mu]$. We use $Q^\pi_\mathcal{M}(s, a)$ and $V^\pi_\mathcal{M}(s)$ to denote the Q function and value function of a policy $\pi$. We also define the state-action visitation distribution (also known as the occupancy measure in the MDP literature) of $\pi$ as $d^\pi(s, a) := (1 - \gamma)\big(\mu(s, a) + \sum_{t=1}^\infty \gamma^t \mathbb{P}(s_t = s, a_t = a; \pi, \mu)\big)$, for each $(s, a)$.

**Problem Statement of Cross-Domain RL.** In typical CDRL, the knowledge transfer involves two MDPs, namely the source-domain MDP $\mathcal{M}_{\text{src}} := (\mathcal{S}_{\text{src}}, \mathcal{A}_{\text{src}}, P_{\text{src}}, r_{\text{src}}, \gamma, \mu_{\text{src}})$ and the target-domain MDP $\mathcal{M}_{\text{tar}} := (\mathcal{S}_{\text{tar}}, \mathcal{A}_{\text{tar}}, P_{\text{tar}}, r_{\text{tar}}, \gamma, \mu_{\text{tar}})$[1]. Notably, in addition to distinct state and action spaces, the two domains can have different reward functions, transition dynamics, and initial distributions. We assume that the two MDPs share the same discounted factor $\gamma$, which is rather mild. Moreover, the trajectories of the two domains are completely unpaired. Let $\Pi_{\text{tar}}$ be the set of all stationary Markov policies for $\mathcal{M}_{\text{tar}}$.

The goal of the RL agent is to learn a policy $\pi^*$ in the target domain such that the expected total discounted reward is maximized, *i.e.,* $\pi^* := \arg\max_{\pi \in \Pi_{\text{tar}}} V^\pi_{\mathcal{M}_{\text{tar}}}(\mu_{\text{tar}})$. To improve sample efficiency via knowledge transfer (compared to learning from scratch), in CDRL, the target-domain agent is granted access to $(\pi_{\text{src}}, Q_{\text{src}}, V_{\text{src}})$, which denotes a policy and the corresponding Q and value functions pre-trained in $\mathcal{M}_{src}$. Notably, we make no assumption on the quality of $\pi_{\text{src}}$ (and hence $\pi_{\text{src}}$ may not be optimal to $\mathcal{M}_{\text{src}}$), despite that $\pi_{\text{src}}$ shall exhibit acceptable performance in practice.

In this paper, we focus on designing a reliable CDRL algorithm in that it effectively leverages a source-domain Q function $Q_{\text{src}}$ for knowledge transfer to the target domain, regardless of the quality of $Q_{\text{src}}$ and domain similarity.

**Inter-Domain Mapping Functions.**   To address the discrepancy in state-action spaces in CDRL, learning an inter-domain mapping is one common block of many CDRL algorithms. Specifically, there are a variety of ways to construct the mapping functions, such as handcrafted functions (Ammar & Taylor, 2012), encoders and decoders trained by cycle consistency (You et al., 2022) like cycle-GAN (Zhu et al., 2017), neural networks trained by dynamics alignment of the MDPs (Gui et al., 2023). Moreover, mapping functions have various candidate target spaces, such as a latent space, state or action spaces of the target domain (*i.e.,* from $\mathcal{S}_{\text{src}}, \mathcal{A}_{\text{src}}$ to $\mathcal{S}_{\text{tar}}, \mathcal{A}_{\text{tar}}$), and state or action spaces of the source domain (*i.e.,* from $\mathcal{S}_{\text{tar}}, \mathcal{A}_{\text{tar}}$ to $\mathcal{S}_{\text{src}}, \mathcal{A}_{\text{src}}$).

For example, Gui et al. (2023) proposed learning two mappings, $G_1 : \mathcal{S}_{\text{tar}} \rightarrow \mathcal{S}_{\text{src}}$ and $G_2 : \mathcal{A}_{\text{src}} \rightarrow \mathcal{A}_{\text{tar}}$, via dynamics alignment, which infers the unknown mapping between unpaired trajectories of $\mathcal{M}_{\text{src}}$ and $\mathcal{M}_{\text{tar}}$ by aligning one-step state transitions. However, this unsupervised approach provides no performance guarantee and can suffer from identification issues. By contrast, we propose learning inter-domain state and action mappings, $\phi : \mathcal{S}_{\text{tar}} \rightarrow \mathcal{S}_{\text{src}}$ and $\psi : \mathcal{A}_{\text{tar}} \rightarrow \mathcal{A}_{\text{src}}$, using a cross-domain Bellman-like loss with guarantees (Section 4). Appendix D.1 shows a toy example where cycle consistency fails, but the Bellman-like loss leverages target rewards to learn a better mapping.

---

[1]Throughout this paper, we use the subscripts "src" and "tar" to represent the objects in the source and target domains, respectively.

**Tabular Approximate Q-Natural Policy Gradient.** Natural Policy Gradient (NPG) (Kakade, 2001; Agarwal et al., 2019) is a classical RL algorithm. In this paper, we adopt NPG under two assumptions to analyze CDRL: (i) **Tabular setting:** finite state and action spaces, with independent parameters for each state–action pair $(s, a)$; (ii) **Approximate Q-function:** the true $Q^\pi$ is inaccessible due to limited data, so we use an empirical approximation from samples. At iteration $t$, we first collect data $\mathcal{D}^{(t)}$ by executing $\pi^{(t)}$, then obtain $Q^{(t)}$ by minimizing the standard TD loss for least-squares policy evaluation (LSPE) (Lagoudakis & Parr, 2001; Yu & Bertsekas, 2009; Lazaric et al., 2012)[2]

$$\mathcal{L}_{\mathrm{TD}}(Q^{(t)}; \pi^{(t)}, \mathcal{D}^{(t)}) := \hat{\mathbb{E}}_{(s,a,r,s')\in\mathcal{D}^{(t)}}\left[\left|r + \gamma\mathbb{E}_{a'\sim\pi^{(t)}}[Q^{(t)}(s', a')] - Q^{(t)}(s, a)\right|^2\right]. \quad (1)$$

Finally, we perform a one-step policy improvement: $\pi^{(t+1)}(a|s) \propto \pi^{(t)}(a|s)\exp\left(\eta Q^{(t)}(s, a)\right)$, where $\eta$ is the learning rate. This update improves the policy while staying close to the original.

**Notation.** Throughout this paper, for any policy $\pi$ and any real-valued function $h : \mathcal{S} \times \mathcal{A} \to \mathbb{R}$, we use $h(s, \pi)$ and $\bar{h}(s, a; \pi)$ as the shorthand for $\mathbb{E}_{a\sim\pi(\cdot|s)}[h(s, a)]$ and $h(s, a) - \mathbb{E}_{a\sim\pi(\cdot|s)}[h(s, a)]$, respectively. For any real vector $z$ and $p \geq 1$, we let $\|z\|_p$ be the $\ell_p$-norm of $z$. For any real-valued function $f : \mathcal{S} \times \mathcal{A} \to \mathbb{R}$, we use $\|f\|_{d^{\pi^{(t)}}}$ as the shorthand for $\mathbb{E}_{(s,a)\sim d^{\pi^{(t)}}}[f(s, a)]$.

## 4 METHODOLOGY

In this section, we first describe the concept of cross-domain Bellman consistency and accordingly propose the $Q$Avatar framework in the tabular setting (*i.e.*, $\mathcal{S}_{\mathrm{tar}}$ and $\mathcal{A}_{\mathrm{tar}}$ are finite). We then extend this framework to a practical deep RL implementation.

### 4.1 CROSS-DOMAIN BELLMAN CONSISTENCY

To motivate Source domain Q-function transfer, we

---

**Algorithm 1** Direct Q Transfer (DQT)

---
**Require:** Source-domain Q function $Q_{\mathrm{src}}$, total iterations $T$, and $\eta = (1 - \gamma)\sqrt{1/T}$.
1: Initialize $\pi^{(1)}$ as a uniformly random policy.
2: **for** iteration $t = 1, \cdots, T$ **do**
3:     Select $\phi^{(t)}$ and $\psi^{(t)}$
4:     Update target-domain policy as in (3).
5: **end for**
6: **Return** $\pi_{\mathrm{tar}}^{(T)} \sim \mathrm{Uniform}(\{\pi^{(1)}, \cdots, \pi^{(T)}\})$.

---

present the sub-optimal gap of traditional NPG. First, we describe the definitions of state-action distribution coverage and TD error.

**Definition 1** (Coverage). *Given a target-domain policy $\pi^\dagger$ in $\mathcal{M}_{tar}$, we say that $\pi^\dagger$ has coverage $C_{\pi^\dagger}$ if for any policy $\pi \in \Pi_{tar}$, we have $\|d^{\pi^\dagger}/d^\pi\|_\infty \leq C_{\pi^\dagger}$.*

**Assumption 1.** *The initial distribution is exploratory, i.e., $\mu_{tar}(s, a) > 0$, for all $s, a$.*

Notably, $C_{\pi^\dagger}$ is finite if $\|d^{\pi^\dagger}/\mu_{\mathrm{tar}}\|_\infty$ is finite (since $\|\mu_{\mathrm{tar}}/d^\pi\|_\infty \leq 1/(1 - \gamma)$ for all $\pi$ by the definition of $d^\pi$), which holds under an exploratory initial distribution with $\mu_{\mathrm{tar}}(s, a) > 0$ for all $(s, a)$—a standard assumption in the NPG literature (Agarwal et al., 2021a; Ding et al., 2020; Yuan et al., 2022; Agarwal et al., 2020; Zhou et al., 2024). Intuitively, coverage enables direct comparison of Bellman errors between policies. We also use $\mu_{\mathrm{tar,min}}$ as shorthand for $\min_{s,a}\mu_{\mathrm{tar}}(s, a)$.

**Definition 2** (TD Error). *For each state-action pair $(s, a)$ and $t \in \mathbb{N}$, the TD error $\epsilon_{td}^{(t)}(s, a)$ is defined as $\epsilon_{td}^{(t)}(s, a) := \left|Q_{tar}^{(t)}(s, a) - r_{tar}(s, a) - \gamma\mathbb{E}_{s'\sim P_{tar}(\cdot|s,a), a'\sim\pi^{(t)}(\cdot|s')}[Q_{tar}^{(t)}(s', a')]\right|$.*

**Proposition 1.** *Under the tabular and approximate-Q settings, and Assumption 1, the average sub-optimality of Q-NPG over $T$ iterations is upper bounded by*

$$\frac{1}{T}\sum_{t=1}^T \mathbb{E}_{s\sim\mu_{tar}}\left[V^{\pi^*}(s) - V^{\pi^{(t)}}(s)\right]$$

$$\leq \underbrace{\frac{[\log|\mathcal{A}_{tar}| + 1]}{\sqrt{T}(1 - \gamma)}}_{(a)} + \underbrace{\frac{C_0}{T}\sum_{t=1}^T \left\|\left|Q_{tar}^{(t)} - Q^{\pi^{(t)}}\right|\right\|_{d^{\pi^{(t)}}}}_{(b)} \leq \underbrace{\frac{[\log|\mathcal{A}_{tar}| + 1]}{\sqrt{T}(1 - \gamma)}}_{(a)} + \underbrace{\frac{C_1}{T}\sum_{t=1}^T \|\epsilon_{td}^{(t)}\|_{d^{\pi^{(t)}}}}_{(c)}, \quad (2)$$

*where $C_0 := 2C_{\pi^*}/(1 - \gamma)$ and $C_1 := 2C_{\pi^*}/((1 - \gamma)^3\mu_{tar, min})$.*

---
[2]LSPE under linear function approximation includes the tabular case via one-hot features:

The detailed proof of Proposition 1 is provided in Appendix B. The upper bound of the sub-optimality gap has two parts. Term (a) characterizes Q-NPG learning and converges at $O(1/\sqrt{T})$, while term (b) (or equivalently term (c)) accounts for approximation error at each iteration, which can be made arbitrarily small with enough samples (Agarwal et al., 2021a). In CDRL, limited data amplifies term (b), potentially preventing convergence to the optimal policy. To mitigate this issue, instead of learning $Q^{(t)}$ from scratch to approximate $Q^{\pi^{(t)}}$, we leverage a pre-trained source-domain Q-function $Q_{src}(\phi^{(t)}(s), \psi^{(t)}(a))$ with inter-domain mapping $\phi^{(t)}$ and $\psi^{(t)}$ to approximate $Q^{\pi^{(t)}}$. Here, the inter-domain mappings $\phi^{(t)}$ and $\psi^{(t)}$ are introduced to address the state–action representation mismatch. For more specifically, we present Direct Q Transfer (DQT) method, in each iteration $t$, DQT proceeds in two steps: (i) It first updates $\phi^{(t)}$ and $\psi^{(t)}$, *e.g.,* by gradient descent on some loss function. (ii) The policy is updated by an NPG policy improvement step based on the pre-trained source-domain $Q_{\text{src}}$ and inter-domain mappings $\phi^{(t)}, \psi^{(t)}$ as

$$\pi^{(t+1)}(a|s) \propto \pi^{(t)}(a|s) \exp\left(\eta Q_{\text{src}}(\phi^{(t)}(s), \psi^{(t)}(a))\right), \tag{3}$$

where $\eta$ is the step size. The pseudo code is in Algorithm 1. Before characterizing the convergence behavior, we describe the cross-domain Bellman error used in Proposition 2.

**Definition 3** (Cross-Domain Bellman Error). *Given a pre-trained source-domain $Q_{src}$, inter-domain correspondences $\phi, \psi$, and target-domain policy $\pi$, for each state-action pair $(s, a)$, the cross-domain Bellman error is defined as $\epsilon_{cd}(s, a; \phi, \psi, Q_{src}, \pi) := \left| Q_{src}(\phi(s), \psi(a)) - r_{tar}(s, a) - \gamma \mathbb{E}_{s' \sim P_{tar}(\cdot|s,a), a' \sim \pi(\cdot|s')}[Q_{src}(\phi(s'), \psi(a'))] \right|.$*

**Proposition 2.** *Under the DQT method in Algorithm 1 and Assumption 1, the average sub-optimality over $T$ iterations is upper bounded as*

$$\frac{1}{T} \sum_{t=1}^{T} \mathbb{E}_{s \sim \mu_{tar}}\left[V^{\pi^*}(s) - V^{\pi^{(t)}}(s)\right] \leq \underbrace{\frac{[\log|\mathcal{A}_{tar}| + 1]}{\sqrt{T}(1 - \gamma)}}_{(a)} + \underbrace{\frac{C_0}{T} \sum_{t=1}^{T} \left\| \left| Q_{src}(\phi^{(t)}, \psi^{(t)}) - Q^{\pi^{(t)}} \right| \right\|_{d^{\pi^{(t)}}}}_{(b)}$$

$$\leq \underbrace{\frac{[\log|\mathcal{A}_{tar}| + 1]}{\sqrt{T}(1 - \gamma)}}_{(a)} + \underbrace{\frac{C_1}{T} \sum_{t=1}^{T} \left\| \epsilon_{cd}(Q_{src}, \phi^{(t)}, \psi^{(t)}) \right\|_{d^{\pi^{(t)}}}}_{(c)}, \tag{4}$$

*where $C_0 := 2C_{\pi^*}/(1 - \gamma)$ and $C_1 := 2C_{\pi^*}/((1 - \gamma)^3 \mu_{tar, min})$.*

The detailed proof of Proposition 2 is in Appendix B. The main insights are: (i) Similar to Proposition 1, the upper bound has two terms. Term (a) characterizes Q-NPG learning, while the sub-optimality gap is mainly determined by the approximation error from $Q_{src}$, equivalent to the cross-domain Bellman error (term (c)). (ii) Minimizing this error requires $\phi$ and $\psi$ that reduce term (c). Motivated by Equation (4), we define cross-domain Bellman consistency.

**Definition 4** (Cross-Domain Bellman Consistency). *Let $\delta \geq 0$. A source-domain critic $Q_{src}$ is said to be $\delta$-Bellman-consistent under target domain policy $\pi$ if there exist a pair of inter-domain mapping $(\phi, \psi)$ such that $\|\epsilon_{cd}(Q_{src}, \phi, \psi)\|_{d^\pi}$ is no more than $\delta$.*

**Transferability of a Source-Domain Model.** Given a source-domain critic $Q_{\text{src}}$, if for any iteration $t$ there exist inter-domain mappings $\phi^{(t)}$ and $\psi^{(t)}$ such that $Q_{\text{src}}$ is $\delta$-Bellman-consistent under $\pi^{(t)}$, then term (c) in (4) is bounded by $C_1\delta$. Thus, the transferability of a source-domain model is captured by $\delta$. In the perfect transfer scenario, where source and target domains are identical and $Q_{\text{src}}$ is optimal, setting $\phi$ and $\psi$ as identity mappings ensures small $\delta$ for all $t$, yielding a small sub-optimality gap for sufficiently large $T$.

**Limitations of DQT.** By Proposition (2), a limitation of DQT is that with a poorly transferable source critic, the cross-domain Bellman error at each iteration $t$ is large, so term (c) in (4) dominates the bound and prevents effective cross-domain transfer.

## 4.2 THE $Q$AVATAR ALGORITHM

To address DQT's limitation, we propose $Q$Avatar, which uses a hybrid critic consisting of a weighted combination of a learned target-domain Q function and a given source-domain Q function to enable

---

**Algorithm 2** $Q$Avatar

---

**Require:** Source-domain Q function $Q_{\text{src}}$.
1: Initialize the state mapping function $\phi^{(0)}$, the action mapping function $\psi^{(0)}$, number of on-policy samples per iteration $N_{\text{tar}}$, the target-domain policy $\pi^{(0)}$, weight decay function $\alpha(0) = 0$, and $\eta = (1 - \gamma)\sqrt{1/T}$.
2: **for** iteration $t = 1, \cdots, T$ **do**
3:     Sample $\mathcal{D}_{\text{tar}}^{(t)} = \{(s, a, r, s')\}$ of $N_{\text{tar}}^{(t)}$ on-policy samples using $\pi^{(t)}$ in the target domain.
4:     Update $Q_{\text{tar}}$ by minimizing the TD loss in (1), i.e., $Q_{\text{tar}}^{(t)} \leftarrow \arg\min_{Q_{\text{tar}}} \mathcal{L}_{\text{TD}}(Q_{\text{tar}}; \pi^{(t)}, \mathcal{D}_{\text{tar}}^{(t)})$.
5:     Update $\phi$ and $\psi$ by minimizing (5), i.e., $\phi^{(t)}, \psi^{(t)} \leftarrow \arg\min_{\phi,\psi} \mathcal{L}_{\text{CD}}(\phi, \psi; Q_{\text{src}}, \pi^{(t)}, \mathcal{D}_{\text{tar}}^{(t)})$.
6:     Defined weight parameter $\alpha(t) = \|\epsilon_{\text{td}}^{(t)}\|_{\mathcal{D}_{\text{tar}}^{(t)}} / (\|\epsilon_{\text{cd}}(Q_{\text{src}}, \phi^{(t)}, \psi^{(t)})\|_{\mathcal{D}_{\text{tar}}^{(t)}} + \|\epsilon_{\text{td}}^{(t)}\|_{\mathcal{D}_{\text{tar}}^{(t)}})$
7:     Update the target-domain policy by adapting NPG to CDRL as in (6).
8: **end for**
9: **Return** Target-domain policy $\pi_{\text{tar}}^{(T)} \sim \text{Uniform}(\{\pi^{(1)}, \cdots, \pi^{(T)}\})$.

---

reliable cross-domain knowledge transfer. This design allows $Q$Avatar to improve sample efficiency in favorable scenarios while avoiding reliance on poorly transferable source models. Specifically, $Q$Avatar comprises three major components:

- **Inter-domain mapping**: Under $Q$Avatar, we propose to learn the inter-domain mappings $\phi : \mathcal{S}_{\text{tar}} \rightarrow \mathcal{S}_{\text{src}}$ and $\psi : \mathcal{A}_{\text{tar}} \rightarrow \mathcal{A}_{\text{src}}$ by minimizing the cross-domain Bellman loss as

$$\mathcal{L}_{\text{CD}}(\phi, \psi; Q_{\text{src}}, \pi_{\text{tar}}, \mathcal{D}_{\text{tar}}) := \hat{\mathbb{E}}_{(s,a,r_{\text{tar}},s') \in \mathcal{D}_{\text{tar}}} \left[ \left| r_{\text{tar}} + \gamma \mathbb{E}_{a' \sim \pi_{\text{tar}}}[Q_{\text{src}}(\phi(s'), \psi(a'))] - Q_{\text{src}}(\phi(s), \psi(a)) \right|^2 \right],$$
(5)

  where $Q_{\text{src}}$ is the pre-trained source-domain Q function and $\mathcal{D}_{\text{tar}} = \{(s, a, r_{\text{tar}}, s')\}$ denotes a set of target-domain samples drawn under $\pi_{\text{tar}}$. Intuitively, the loss in (5) looks for a pair of mapping functions $\phi, \psi$ such that $Q_{\text{src}}$ aligns as much with the target-domain transitions as possible.

- **Target-domain Q function**: To implement the hybrid critic, $Q$Avatar maintains a target-domain Q function $Q_{\text{tar}}$, serving as the critic of the current target-domain policy. At each iteration $t$, $Q_{\text{tar}}$ is obtained via policy evaluation by minimizing the TD loss $\mathcal{L}_{\text{TD}}(Q_{\text{tar}}; \pi_{\text{tar}}, \mathcal{D}_{\text{tar}})$, where $\mathcal{D}_{\text{tar}} = (s, a, r, s')$ are target-domain samples (Equation 1).

- **NPG-like policy update with a weighted Q-function combination**: $Q$Avatar leverages both $Q_{\text{src}}$ and $Q_{\text{tar}}$ for policy updates. At each iteration $t$,

$$\pi^{(t+1)}(a|s) \propto \pi^{(t)}(a|s) \cdot \exp\left(\eta\left((1 - \alpha(t))Q_{\text{tar}}^{(t)}(s, a) + \alpha(t)Q_{\text{src}}(\phi^{(t)}(s), \psi^{(t)}(a))\right)\right),$$
(6)

  where $\alpha : \mathbb{N} \rightarrow [0, 1]$ is a weight function (see Section 4.3).

The pseudo code of $Q$Avataris provided in Algorithm 2.

**Remark 1.** In line 6 of Algorithm 1 and line 8 of Algorithm 2, DQT and $Q$Avatar output the final policy by selecting uniformly from all intermediate policies which is a standard procedure linking average sub-optimality to policy performance. In experiments, the last-iterate policy suffices and performs well.

## 4.3 THEORETICAL JUSTIFICATION OF $Q$AVATAR

In this section, we present the theoretical result of $Q$Avatar and thereby describe how to choose the proper decay parameter $\alpha(\cdot)$.

**Definition 5** (Cross-Domain Action Value Function). *For each state-action pair $(s, a)$ and $t \in \mathbb{N}$, the cross-domain action value function $f^{(t)}(s, a)$ is defined as $f^{(t)}(s, a) := (1 - \alpha(t))Q_{tar}^{(t)}(s, a) + \alpha(t)Q_{src}(\phi^{(t)}(s), \psi^{(t)}(a))$.*

We are ready to present the main theoretical result, and the detailed proof is provided in Appendix B.

**Proposition 3.** *(Average Sub-Optimality) Under the QAvatar in Algorithm 2 and Assumption 1, the average sub-optimality over $T$ iterations can be upper bounded as*

$$
\frac{1}{T}\sum_{t=1}^{T}\mathbb{E}_{s\sim\mu_{tar}}\Big[V^{\pi^*}(s)-V^{\pi^{(t)}}(s)\Big]
$$

$$
\leq \underbrace{\frac{[\log|\mathcal{A}_{tar}|+1]}{\sqrt{T}(1-\gamma)}}_{(a)} + \underbrace{\frac{C_0}{T}\sum_{t=1}^{T}\mathbb{E}_{(s,a)\sim d^{\pi^{(t)}}}\Big[\big|f^{(t)}(s,a)-Q^{\pi^{(t)}}(s,a)\big|\Big]}_{(b)} \tag{7}
$$

$$
\leq \underbrace{\frac{[\log|\mathcal{A}_{tar}|+1]}{\sqrt{T}(1-\gamma)}}_{(a)} + \underbrace{\frac{C_1}{T}\sum_{t=1}^{T}\Big(\alpha(t)\|\epsilon_{cd}(Q_{src},\phi^{(t)},\psi^{(t)})\|_{d^{\pi^{(t)}}} + (1-\alpha(t))\|\epsilon_{td}^{(t)}\|_{d^{\pi^{(t)}}}\Big)}_{(c)}, \tag{8}
$$

*where $C_0 := 2C_{\pi^*}/(1-\gamma)$ and $C_1 := 2C_{\pi^*}/((1-\gamma)^3\mu_{tar,\,min})$.*

Notably, the term (a) in (8) reflects the learning progress of NPG, and term (c) reflects the transferability of a source-domain critic $Q_{\text{src}}$ and the error of policy evaluation for the target-domain policy.

**A Hyperparameter-Free Design of $\alpha(t)$.** Based on (8), for each iteration $t$, term (c) can be minimized by choosing $\alpha(t)$ as an indicator function, i.e., set to 1 when $\|\epsilon_{\text{cd}}(Q_{\text{src}},\phi^{(t)},\psi^{(t)})\|_{d^{\pi^{(t)}}} < \|\epsilon_{\text{td}}^{(t)}\|_{d^{\pi^{(t)}}}$, and 0 otherwise. In practice, estimating the two error terms is noisy, so using an indicator can cause large fluctuations in $\alpha(t)$ and unstable training. To address this, we propose a smoother variant: $\alpha(t) = \|\epsilon_{\text{td}}^{(t)}\|_{d^{\pi^{(t)}}}/(\|\epsilon_{\text{cd}}(Q_{\text{src}},\phi^{(t)},\psi^{(t)})\|_{d^{\pi^{(t)}}} + \|\epsilon_{\text{td}}^{(t)}\|_{d^{\pi^{(t)}}})$. Notably, this design is *hyperparameter-free* and incurs minimal deployment overhead.

**Key Implications of Proposition 3:** (1) Effective transfer lowers the upper bound of average sub-optimality: In an ideal case with perfect mappings $\phi^*,\psi^*$ such that $L_{\text{CD}}(\phi^*,\psi^*;Q_{\text{src}},\pi_{\text{tar}},\mathcal{D}_{\text{tar}}) = 0$ for any $\pi_{\text{tar}}$, we obtain $\|\epsilon_{\text{cd}}(Q_{\text{src}},\phi^*,\psi^*)\|_{d^{\pi_{\text{tar}}}} = 0$. Then $\alpha(t) = \mathbf{1}$ at all $t$, making term (c) in (8) vanish. The bound thus reduces to term (a), which becomes negligible as $T$ grows. (2) QAvatar avoids being trapped by low-transfer critics. For a source critic only $\delta$-Bellman-consistent with large $\delta$, $\|\epsilon_{\text{cd}}(Q_{\text{src}},\phi,\psi)\|_{d^{\pi^{(t)}}}$ remains large, so $\alpha(t) \approx 0$. Consequently, term (c) reduces to the standard TD error.

## 4.4 PRACTICAL IMPLEMENTATION OF QAVATAR

We extend the QAvatar framework in Algorithm 2 to a practical deep RL implementation. The pseudo code is provided in Algorithm 3 in Appendix.

- **Learning the target-domain policy and the Q function.** To go beyond the tabular setting, we extend QAvatar by connecting NPG with soft policy iteration (SPI) (Haarnoja et al., 2018). In the entropy-regularized RL setting, SPI is known to be a special case of NPG (Cen et al., 2022). Based on this connection, we choose to integrate QAvatar with soft actor-critic (SAC) (Haarnoja et al., 2018), *i.e.,* updating the target-domain critic $Q_{\text{tar}}$ by the critic loss of SAC and updating the target-domain policy $\pi^{(t)}$ by the SAC policy loss with the weighted combination of $Q_{\text{tar}}$ and $Q_{\text{src}}$ of QAvatar.

- **Learning the inter-domain mapping functions with an augmented flow model.** Similar to the tabular setting, we learn inter-domain mappings by minimizing the cross-domain Bellman loss. In practical RL problems, the state and action spaces are usually bounded, so the outputs of $\phi : \mathcal{S}_{\text{tar}} \to \mathcal{S}_{\text{src}}$ and $\psi : \mathcal{A}_{\text{tar}} \to \mathcal{A}_{\text{src}}$ must lie within feasible regions. As discussed in Section 2, adversarial learning is commonly used to address this (Taylor et al., 2008; Zhang et al., 2021; Gui et al., 2023; Zhu et al., 2024), but it can lead to unstable training. Therefore, we adopt the method of (Brahmanage et al., 2023) in the action-constrained RL literature (Hung et al., 2025), training a normalizing flow to map the outputs of the mapping functions into the feasible regions.

## 5 EXPERIMENTS

### 5.1 SETUP

**Benchmark CDRL Methods.** We compare $Q$Avatar with recent CDRL benchmarks under different state-action spaces, including Cross-Morphology-Domain Policy Adaptation (CMD) (Gui et al., 2023), Cross-domain Adaptive Transfer (CAT) (You et al., 2022), and Policy Adaptation by Representation mismatch (PAR) (Lyu et al., 2024). For a fair comparison, all methods use the same source-domain models, including policy and corresponding Q-networks, pre-trained with SAC. We also evaluate both PPO-based CAT, the original version in (You et al., 2022), and SAC-based CAT. Notably, CMD is an enhanced version of (Zhang et al., 2021) that integrates dynamics cycle consistency to learn state-action correspondences.

To demonstrate sample efficiency, we also compare $Q$Avatar with standard SAC (Haarnoja et al., 2018), which learns from scratch in the target domain, and with direct fine-tuning (FT) of the source models (Ha et al., 2024), equivalent to SAC with source feature initialization. Both serve as competitive baselines. Hyperparameters are provided in Appendix F.

**Evaluation Environments.**

- **Locomotion**: We use the standard MuJoCo environments, including HalfCheetah-v3 and Ant-v3, as the source domains and follow the same procedure as in (Zhang et al., 2021; Xu et al., 2023) to modify them for the target domains. The detailed morphologies are in Appendix F.

- **Robot arm manipulation**: We leverage Robosuite, a popular package for robot learning released by (Zhu et al., 2020) and evaluate our algorithm on door opening and table wiping. For each task, we use the Panda robot arm as the source domain and set the UR5e robot arm as the target domain.

- **Goal Navigation**: A natural transfer scenario occurs when the source and target domains share the same goal but differ in robot type. We use the Safety-Gym benchmark (Ray et al., 2019) and evaluate transfer from Car to Doggo, keeping the goal unchanged, specifically using CarGoal0 as the source and DoggoGoal0 as the target domain.

The dimensions of the state and action spaces of all the source-target pairs are in Table 3 in Appendix F. All the results reported below are averaged over 5 random seeds.

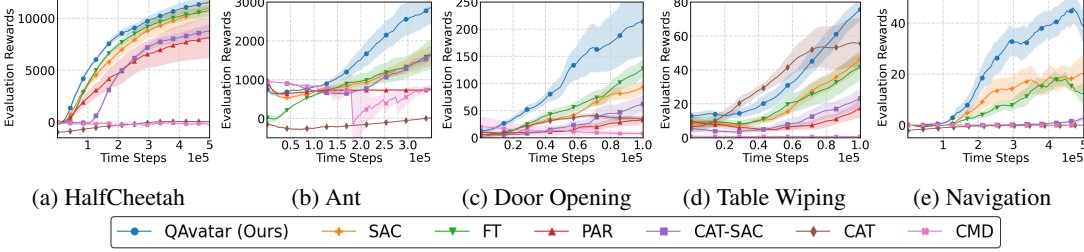

Figure 1: Training curves of $Q$Avatar and benchmark methods: (a)-(b) Locomotion tasks; (d)-(e) Robot arm manipulation tasks in Robosuite; (f) Navigation task from CarGoal0 to DoggoGoal0.

### 5.2 EXPERIMENTAL RESULTS

**Does $Q$Avatar improve data efficiency?**

**Learning curves**: As shown by Figure 1, we observe that $Q$Avatar achieves improved data efficiency via cross-domain transfer than SAC throughout the training process in all the tasks, despite that these tasks have rather different dimensions as shown in Table 3.

CAT-SAC achieves moderate results on MuJoCo but transfers slowly to other tasks, as CAT-like methods lack guarantees and depend on parameter-based transfer, i.e., weighted combinations of source and target policy layers. Such methods assume shared feature representations (Zhuang et al., 2020), which often fails when domains differ. FT improves data efficiency over SAC on MuJoCo but learns slowly in Robosuite due to dissimilar state–action representations from different robot arms. CMD generally performs poorly and can be unstable (e.g., in Ant) owing to its adversarial mapping module. We attribute CMD's weakness to its unsupervised design, which ignores target-domain rewards.

**Time to threshold**: We provide Table 1 to mark the time to threshold. It shows that $Q$Avatar requires only about 44% of the environment steps to achieve the threshold than SAC does in the best case.

**Aggregated performance**: To ensure a reliable comparison, we follow the guidelines of (Agarwal et al., 2021b) and calculate the interquartile mean (IQM) using rliable, which enables evaluation at an aggregated level. Figure 2 shows that $Q$Avatarindeed achieves significantly better performance than all baselines.

Table 1: Time to threshold of $Q$Avatar and SAC.

| Environment | Threshold | $Q$Avatar | SAC | $Q$Avatar / SAC |
|---|---|---|---|---|
| HalfCheetah | 6000 | 126K | 176K | 0.71 |
| Ant | 1600 | 206K | 346K | 0.59 |
| Door Opening | 90 | 48K | 98K | 0.49 |
| Table Wiping | 45 | 72K | 98K | 0.73 |
| Navigation | 20 | 218K | 490K | 0.44 |

Figure 2: Aggregated IQMs (with 95% stratified bootstrap CIs) across tasks.

**How does $Q$Avatar perform under strong positive and negative transfer?** We consider a task where the source domain is standard 'Ant-v3' and the target changes the goal to move backward, with all else unchanged. Here, $Q_{\text{src}}$ and $Q_{\text{tar}}$ are adversarial due to opposite goals. We evaluate QAvatar in two scenarios: (a) **Learning state/action mapping**: strong transferability exists, as Ant is symmetric along the front-back axis, allowing a perfect mapping. (b) **Fixing mapping as identity**: a strong negative transfer case, since $Q_{\text{src}}$ provides adversarial reward signals. As shown in Figure 3, $Q$Avatar captures both positive transfer (high $\alpha(t)$) and negative transfer (low $\alpha(t)$), demonstrating that $\alpha(t)$ reflects transferability.

**Performance of $Q$Avatar with a low-quality source domain:** We evaluate this scenario in the Cheetah environment (Section 5.1) using a low-quality source model with a total return of 1000 (vs. $\sim$7000 for the expert). Figure 4 illustrates the learning process and $\alpha(t)$ of QAvatar. Results show that when the source model is of low quality, $\alpha(t)$ decreases to a small value by the end of training, mitigating the effect of negative transfer.

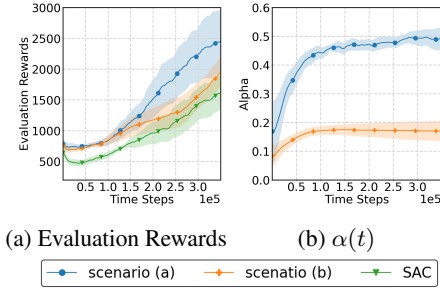

(a) Evaluation Rewards   (b) $\alpha(t)$

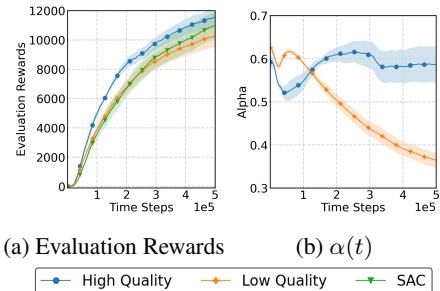

(a) Evaluation Rewards   (b) $\alpha(t)$

Figure 3: The training curve and the values of $\alpha(t)$ for $Q$Avatar under strongly positive and strongly negative transfer scenarios.

Figure 4: The training curve and the values of $\alpha(t)$ in the Cheetah environment with a low-quality source model.

**Does $Q$Avatar still perform reliably well when the source and target with two unrelated transfer scenario?** We evaluate transfer from original Hopper-v3 in MuJoCo to the table-wiping task in Robosuite. The configurations of these environments are provided in Section 5.1. Figure 9a shows that even when the source and target domains share no structural similarity, $Q$Avatar still performs reliably and does not suffer from negative transfer.

**How $Q$Avatar perform on non-stationary environment?** We use the Ant environment and introduce stochasticity by adding $\mathcal{N}(0, 0.1)$ noise to rewards and $\mathcal{N}(0, 0.05)$ to actions, following (Tessler et al., 2019). As shown in Figure 9b, despite stochastic rewards and transitions, the inter-domain mapping is effectively learned, enabling positive transfer and faster learning in the target domain.

**$Q$Avatar on image-based experiment.** We additionally evaluate $Q$Avatar on image-based continuous control tasks from the DeepMind Control Suite (DMC) (Tassa et al., 2018). In DMC, each observation consists of a stack of three 84×84 RGB frames, and we apply an action repeat of 4. The SAC setup,

the flow-model training, and the cross-domain transfer protocol is describe in Appendix D.4. For cross-domain experiments, we consider two transfer scenarios: The source model is trained with SAC on the walker_walk task, and the target tasks are walker_run and cheetah_run, respectively. Figure 5 indicate that $Q$Avatar achieves substantially higher performance than SAC trained from scratch on both target tasks, and notably, $Q$Avatar succeeds even when SAC struggles to learn effectively on cheetah_run.

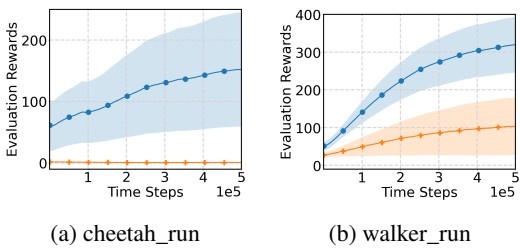

(a) cheetah_run        (b) walker_run

Figure 5: Performance comparison on DMC tasks.

**Sensitivity test on the choice of $N_\alpha$.** In $Q$Avatar, $\alpha_t$ is updated periodically, and $N_\alpha$ determines only the frequency of this closed-form update. As long as $N_\alpha$ is not too large (which would delay updates) or too small (which may cause $\alpha_t$ to fluctuate too rapidly and introduce instability), the overall learning behavior remains largely unaffected. We evaluate three update intervals, $N_\alpha = 300, 1000, 3000$, in the Cheetah and Table Wiping environments, whose configurations are provided in Section 5.1. As shown in Figure 6 and 7, the results indicate that (1) **the learned $\alpha_t$ trajectories are highly similar across settings**, and (2) **performance exhibits only mild sensitivity**. None of the choices lead to degradation or instability, suggesting that the method is reasonably robust to the selection of $N_\alpha$ within this range.

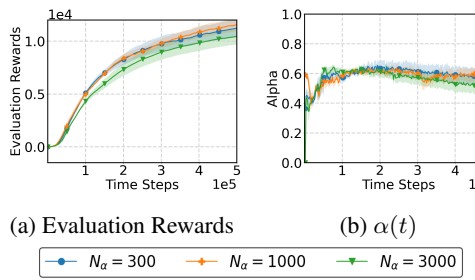

(a) Evaluation Rewards        (b) $\alpha(t)$

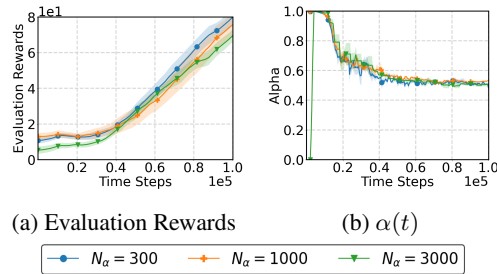

(a) Evaluation Rewards        (b) $\alpha(t)$

Figure 6: The training curves and the values of $\alpha(t)$ for $Q$Avatar under three settings of the $N_\alpha$ value in Cheetah environment.

Figure 7: The training curves and the values of $\alpha(t)$ for $Q$Avatar under three settings of the $N_\alpha$ value in Table Wiping environment.

**Extension: $Q$Avatar with more than one source model.** $Q$Avatar can be readily extended for transfer from multiple source model. Similar to the idea of one source critic transfer, the weight $\alpha_i(t)$ for the $i$-th source critic $Q_{\text{src},i}$, $\alpha_i(t) = (1/\|\epsilon_{\text{cd}}(Q_{\text{src},i}, \phi_i^{(t)}, \psi_i^{(t)})\|_{d^{\pi^{(t)}}})/(1/\|\epsilon_{\text{td}}^{(t)}\|_{d^{\pi^{(t)}}} + \sum_{j=1}^{N} 1/\|\epsilon_{\text{cd}}(Q_{\text{src},j}, \phi_i^{(t)}, \psi_i^{(t)})\|_{d^{\pi^{(t)}}})$. Consider a two-source to one-target transfer scenario: (i) Source domain 1 (denoted by "src1") is Ant-v3 with the both front legs disabled; (ii) Source domain 2 (denoted by "src2") is Ant-v3 with the both back legs disabled. (iii) Target domain (denoted by "tar") is the original Ant-v3 with no modifications. Figure 9c shows $Q$Avatar in multi-source cross-domain transfer can achieve higher transferability by leveraging the knowledge from two source domains.

## 6  CONCLUDING REMARKS

We propose cross-domain Bellman consistency as a measure of source-model transferability, and introduce $Q$Avatar, the first CDRL method that reliably handles distinct state-action representations with performance guarantees. Using a hybrid critic and a hyperparameter-free weighting scheme, $Q$Avatar achieves robust knowledge transfer even with weak source models. Experiments confirm its effectiveness for cross-domain RL. The general ideas of Bellman consistency and hybrid critics can be extended to cross-domain transfer in other learning settings, such as preference-based RL and imitation learning (Fickinger et al., 2022; Chu et al., 2026). A limitation of our formulation is the assumption that target-domain data collection is costlier than training compute. Since $Q$Avatar takes about twice the training time of SAC due to inter-domain mappings and the flow model, further acceleration would be needed when training efficiency is critical.

## ACKNOWLEDGMENTS

This research was partially supported by the National Science and Technology Council (NSTC) of Taiwan under Grant Numbers 114-2628-E-A49-002 and 114-2634-F-A49-002-MBK. This work was also partially supported by the Center for Intelligent Team Robotics and Human-Robot Collaboration under the "Top Research Centers in Taiwan Key Fields Program" of the Ministry of Education (MOE), Taiwan. The authors also thank the National Center for High-performance Computing (NCHC) for providing computational and storage resources.

## ETHICS STATEMENT

We conduct our research entirely in simulated environments, using no human participants or sensitive data. This work fully complies with the code of ethics.

## REPRODUCIBILITY STATEMENT

The code for our experiments is provided in the supplementary material, along with a README file detailing the commands required to run the experiments. Furthermore, a comprehensive list of package dependencies is included to facilitate the recreation of the experimental environment.

## USE OF LARGE LANGUAGE MODELS (LLMS)

Large language models (LLMs) were used solely for linguistic editing of the manuscript and had no involvement in the study design, methodology, experiments, or interpretation of the results.

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

APPENDICES

## A  SUPPORTING LEMMAS

**Lemma 1** (Performance difference lemma). *For any two policies $\pi$ and $\pi'$, we have*

$$V^{\pi'}(\mu) - V^\pi(\mu) = \frac{1}{1-\gamma}\mathbb{E}_{s,a\sim d^{\pi'}}[A^\pi(s,a)],$$

*where $A^\pi(s,a) := Q^\pi(s,a) - V^\pi(s)$ is the advantage function.*

*Proof.* This can be directly obtained from Lemma 6.1 in (Kakade & Langford, 2002). □

**Lemma 2** ((Agarwal et al., 2019), Chapter 4). *Let $\tau = (s_0, a_0, s_1, a_1, \cdots)$ denote the (random) trajectory generated under a policy $\pi$ in an infinite-horizon MDP $\mathcal{M}$. For any function $f : \mathcal{S}\times\mathcal{A}\to \mathbb{R}$, we have*

$$\mathbb{E}_\tau\left[\sum_{t=0}^\infty \gamma^t f(s_t,a_t)\right] = \frac{1}{1-\gamma}\mathbb{E}_{(s,a)\sim d^\pi}\left[f(s,a)\right]. \tag{9}$$

**Lemma 3** (Importance Ratio). *Given a fixed policy $\pi$ and a fixed state-action pair $(s,a)$, let $p_k(s,a)$ denote the probability of reaching $(s,a)$ under an initial distribution $d^\pi$ and policy $\pi$ after $k$ time steps. Then, for any $k\in\mathbb{N}$, we have*

$$\frac{p_k(s,a)}{d^\pi(s,a)} \le \frac{1}{(1-\gamma)\mu(s,a)}. \tag{10}$$

*Proof.* To begin with, recall the definition of $d^\pi$ as

$$d^\pi(s,a) := (1-\gamma)\left(\mu(s,a) + \sum_{t=1}^\infty \gamma^t P(s_t = s, a_t = a; \pi, \mu)\right) \equiv \sum_{t=0}^\infty \gamma^t P(s_t = s, a_t = a; \pi, \mu). \tag{11}$$

Let $s_{\text{next},k}$ and $a_{\text{next},k}$ denote the state and action after $k$ time steps. Then, we can write down $p_k(s,a)$:

$$p_k(s,a) = \sum_{(s',a')\in\mathcal{S}\times\mathcal{A}} \mathbb{P}(s_{\text{next},k} = s, a_{\text{next},k} = a|s', a'; \pi)d^\pi(s', a') \tag{12}$$

$$= \sum_{(s',a')\in\mathcal{S}\times\mathcal{A}} \mathbb{P}(s_{\text{next},k} = s, a_{\text{next},k} = a|s', a'; \pi)\cdot(1-\gamma)\cdot\sum_{t=0}^\infty \gamma^t \mathbb{P}(s_t = s', a_t = a'; \pi, \mu) \tag{13}$$

$$= (1-\gamma)\sum_{t=0}^\infty \gamma^t \sum_{s',a'\in\mathcal{S}\times\mathcal{A}} \mathbb{P}(s_{\text{next},k} = s, a_{\text{next},k} = a|s', a'; \pi, \mu)\cdot\mathbb{P}(s_t = s', a_t = a'; \pi, \mu) \tag{14}$$

$$= (1-\gamma)\sum_{t=0}^\infty \gamma^t \mathbb{P}(s_{t+k} = s, a_{t+k} = a; \pi, \mu). \tag{15}$$

Then, we have

$$\frac{p_k(s,a)}{d^\pi(s,a)} = \frac{(1-\gamma)\sum_{t=0}^\infty \gamma^t \mathbb{P}(s_{t+k} = s; a_{t+k} = a; \pi, \mu)}{(1-\gamma)\sum_{t=0}^\infty \gamma^t \mathbb{P}(s_t = s, a_t = a; \pi, \mu)} \tag{16}$$

$$= \frac{\sum_{t=0}^\infty \gamma^t \mathbb{P}(s_{t+k} = s, a_{t+k} = a; \pi, \mu)}{\sum_{t=0}^\infty \gamma^t \mathbb{P}(s_t = s, a_t = a; \pi, \mu)} \tag{17}$$

$$\le \frac{\sum_{t=0}^\infty \gamma^t}{\sum_{t=0}^\infty \gamma^t \mathbb{P}(s_t = s; \pi, \mu)} \tag{18}$$

$$= \frac{1}{1-\gamma}\cdot\frac{1}{\sum_{t=0}^\infty \gamma^t \mathbb{P}(s_t = s; \pi, \mu)}, \tag{19}$$

where (18) holds by $\mathbb{P}(s_{t+k} = s, a_{t+k} = a; \pi, \mu) \leq 1$ and (19) holds by taking the sum of an infinite geometric sequence. By the fact that $\sum_{t=0}^{\infty} \gamma^t \mathbb{P}(s_t = s, a_t = a; \pi, \mu) = \mu(s, a) + \sum_{t=1}^{\infty} \gamma^t \mathbb{P}(s_t = s, a_t = a; \pi, \mu)$, we have

$$\frac{1}{1-\gamma} \cdot \frac{1}{\sum_{t=0}^{\infty} \gamma^t \mathbb{P}(s_t = s, a_t = a; \pi, \mu)} = \frac{1}{1-\gamma} \cdot \frac{1}{\mu(s, a) + \sum_{t=1}^{\infty} \gamma^t \mathbb{P}(s_t = s, a_t = a; \pi, \mu)} \tag{20}$$

$$\leq \frac{1}{(1-\gamma)\mu(s, a)} \tag{21}$$

where (21) holds by $\sum_{t=1}^{\infty} \gamma^t \mathbb{P}(s_t = s, a_t = a; \pi, \mu) \geq 0$. $\qquad \square$

**Lemma 4.** *Let $\nu^{(t)} : \mathcal{S} \times \mathcal{A} \to \mathbb{R}$ and $\pi^{(t)}$ denote any tabular function used in the policy update and the policy at iteration $t$. That is,*

$$\pi^{(t+1)}(a \mid s) \propto \pi^{(t)}(a \mid s) \exp\left(\eta \nu^{(t)}(s, a)\right).$$

*Then, we assume that $\|\nu^{(t)}\|_\infty \leq 1/(1-\gamma)$ and setting learning rate $\eta = (1-\gamma)\sqrt{1/T}$ and optimal policy $\pi^*$, we have*

$$\sum_{t=1}^{T} \mathbb{E}_{(s,a) \sim d^{\pi^*}} \left[\bar{\nu}^{(t)}(s, a)\right] \leq \frac{\sqrt{T} \left[\log|\mathcal{A}_{tar}| + 1\right]}{1-\gamma}$$

*Proof.* Let $\bar{\nu}^{(t)}(s, a) := \nu^{(t)}(s, a) - \nu^{(t)}(s, \pi^{(t)}(s))$. According to the policy update rule, at iteration $t$, the policy $\pi^{(t+1)}$ for the next iteration is updated by the formula:

$$\pi^{(t+1)}(a \mid s) = \frac{\pi^{(t)}(a \mid s) \exp\left(\eta \nu^{(t)}(s, a)\right)}{\sum_{a'} \pi^{(t)}(a' \mid s) \exp\left(\eta \nu^{(t)}(s, a')\right)} = \frac{\pi^{(t)}(a \mid s) \exp\left(\eta \bar{\nu}^{(t)}(s, a)\right)}{\sum_{a'} \pi^{(t)}(a' \mid s) \exp\left(\eta \bar{\nu}^{(t)}(s, a')\right)}. \tag{22}$$

Let $Z_t := \sum_{a'} \pi^{(t)}(a' \mid s) \exp\left(\eta \bar{\nu}^{(t)}(s, a')\right)$. By multiplying both sides of (22) by $Z_t$, taking the logarithm, and then taking the expectation on both sides w.r.t $(s, a) \sim d^{\pi^*}$, we obtain

$$\mathbb{E}_{(s,a) \sim d^{\pi^*}} \left[\eta \bar{\nu}^{(t)}(s, a)\right] = \mathbb{E}_{(s,a) \sim d^{\pi^*}} \left[\log Z_t + \log \pi^{(t+1)}(a \mid s) - \log \pi^{(t)}(a \mid s)\right]. \tag{23}$$

Next, we bound the term $\log Z_t$. Note that $\eta \bar{\nu}^{(t)}(s, a) \leq \sqrt{1/T} \leq 1$ and the fact that $\exp(x) < 1 + x + x^2$ for any $x \leq 1$, we have

$$\log Z_t = \log\left(\sum_{a' \in \mathcal{A}} \pi^{(t)}(a' \mid s) \exp\left(\eta \bar{\nu}^{(t)}(s, a')\right)\right) \tag{24}$$

$$\leq \log\left(\sum_{a' \in \mathcal{A}} \pi^{(t)}(a' \mid s) \left[1 + \left(\eta \bar{\nu}^{(t)}(s, a')\right) + \left(\eta \bar{\nu}^{(t)}(s, a')\right)^2\right]\right) \tag{25}$$

$$\leq \log\left(1 + \frac{\eta^2}{(1-\gamma)^2}\right) \tag{26}$$

$$\leq \frac{\eta^2}{(1-\gamma)^2}, \tag{27}$$

where (26) is because $\sum_{a' \in \mathcal{A}} \pi^{(t)}(a' \mid s) \bar{\nu}^{(t)}(s, a') = 0$ and $\|\nu^{(t)}\|_\infty \leq 1/(1-\gamma)$, (27) is follow the fact that $\log(1 + x) \leq x$ for any $x \geq 0$. Then, we have

$$\mathbb{E}_{(s,a) \sim d^{\pi^*}} \left[\eta \bar{\nu}^{(t)}(s, a)\right] \leq \mathbb{E}_{(s,a) \sim d^{\pi^*}} \left[\log \pi^{(t+1)}(a \mid s) - \log \pi^{(t)}(a \mid s) + \frac{\eta^2}{(1-\gamma)^2}\right]. \tag{28}$$

By taking the summation over iterations on both sides of (28), we have

$$
\sum_{t=1}^{T} \mathbb{E}_{(s,a)\sim d^*}\left[\eta\bar{\nu}^{(t)}(s,a)\right]
$$

$$
\leq \frac{T\eta^2}{(1-\gamma)^2} + \mathbb{E}_{(s,a)\sim d^{\pi^*}}\left[\log\pi^{(T+1)}(a\mid s) - \log\pi^{(1)}(a\mid s)\right].
$$

Using the fact that $\log(\pi(a\mid s)) \leq 0$ and $\pi^{(1)}(a\mid s) = \frac{1}{|\mathcal{A}_{\text{tar}}|}$, we have

$$
\sum_{t=1}^{T} \mathbb{E}_{(s,a)\sim d^{\pi^*}}\left[\bar{\nu}^{(t)}(s,a)\right] \leq \frac{T\eta}{(1-\gamma)^2} + \frac{\log|\mathcal{A}_{\text{tar}}|}{\eta}.
$$

By setting $\eta = (1-\gamma)\sqrt{1/T}$, we have

$$
\sum_{t=1}^{T} \mathbb{E}_{(s,a)\sim d^{\pi^*}}\left[\bar{\nu}^{(t)}(s,a)\right] \leq \frac{\sqrt{T}\left[\log|\mathcal{A}_{\text{tar}}| + 1\right]}{1-\gamma}
$$

$\square$

**Lemma 5.** *Let $\nu^{(t)} : \mathcal{S}\times\mathcal{A}\to\mathbb{R}$ and $\pi^{(t)}$ denote value function used in the policy update and the policy at iteration $t$. That is,*

$$
\pi^{(t+1)}(a\mid s) \propto \pi^{(t)}(a\mid s)\exp\left(\eta\nu^{(t)}(s,a)\right). \tag{29}
$$

*Then, by assuming that $\|\nu^{(t)}\|_\infty \leq 1/(1-\gamma)$ and setting the learning rate $\eta = (1-\gamma)\sqrt{1/T}$ and optimal policy $\pi^*$, we have*

$$
\frac{1}{T}\sum_{t=1}^{T}\mathbb{E}_{s\sim\mu_{tar}}\left[V^{\pi^*}(s) - V^{\pi^{(t)}}(s)\right]
$$

$$
\leq \frac{\left[\log|\mathcal{A}_{tar}| + 1\right]}{\sqrt{T}(1-\gamma)} + \frac{2C_{\pi^*}}{1-\gamma}\frac{1}{T}\sum_{t=1}^{T}\mathbb{E}_{(s,a)\sim d^{\pi^{(t)}}}\left[\left|\nu^{(t)}(s,a) - Q^{\pi^{(t)}}(s,a)\right|\right]
$$

*Proof.*

$$
V^{\pi^*}(\mu_{\text{tar}}) - V^{\pi^{(t)}}(\mu_{\text{tar}})
$$

$$
= \frac{1}{1-\gamma}\mathbb{E}_{(s,a)\sim d^{\pi^*}_{\text{tar}}}\left[A^{\pi^{(t)}}(s,a)\right] \tag{30}
$$

$$
= \frac{1}{1-\gamma}\mathbb{E}_{(s,a)\sim d^{\pi^*}_{\text{tar}}}\left[\bar{\nu}^{(t)}(s,a) - \bar{\nu}^{(t)}(s,a) + A^{\pi^{(t)}}(s,a)\right] \tag{31}
$$

$$
= \frac{1}{1-\gamma}\mathbb{E}_{(s,a)\sim d^{\pi^*}_{\text{tar}}}\left[\bar{\nu}^{(t)}(s,a)\right] + \frac{1}{1-\gamma}\mathbb{E}_{(s,a)\sim d^{\pi^*}_{\text{tar}}}\left[-\bar{\nu}^{(t)}(s,a) + A^{\pi^{(t)}}(s,a)\right] \tag{32}
$$

$$
\leq \frac{1}{1-\gamma}\mathbb{E}_{(s,a)\sim d^{\pi^*}_{\text{tar}}}\left[\bar{\nu}^{(t)}(s,a)\right] + \frac{1}{1-\gamma}\mathbb{E}_{(s,a)\sim d^{\pi^*}_{\text{tar}}}\left[\left|-\bar{\nu}^{(t)}(s,a) + A^{\pi^{(t)}}(s,a)\right|\right], \tag{33}
$$

where (30) holds by the performance difference lemma (cf. Lemma 1), (31) is obtained by adding $\bar{\nu}^t(s,a) - \bar{\nu}^t(s,a)$, (32) is obtained by rearranging the terms in (31), and (33) holds by $x \leq |x|$, for all $x \in \mathbb{R}$. By the fact that $\|\frac{d^{\pi^*}}{d^{\pi^{(t)}}}\|_\infty \leq C$, we have

$$
\frac{1}{1-\gamma}\mathbb{E}_{(s,a)\sim d^{\pi^*}}\left[\bar{\nu}^{(t)}(s,a)\right] + \frac{1}{1-\gamma}\mathbb{E}_{s,a\sim d^{\pi^*}}\left[\left|-\bar{\nu}^{(t)}(s,a) + A^{\pi^{(t)}}(s,a)\right|\right]
$$

$$
\leq \frac{1}{1-\gamma}\mathbb{E}_{(s,a)\sim d^{\pi^*}}\left[\bar{\nu}^{(t)}(s,a)\right] + \frac{1}{1-\gamma}C\cdot\mathbb{E}_{s,a\sim d^{\pi^{(t)}}}\left[\left|-\bar{\nu}^{(t)}(s,a) + A^{\pi^{(t)}}(s,a)\right|\right] \tag{34}
$$

$$
. \tag{35}
$$

Recall the definitions that $\bar{\nu}^{(t)}(s,a) := \nu^{(t)}(s,a) - \nu^{(t)}(s,\pi^{(t)}(s))$ and $A^{\pi^{(t)}}(s,a) := Q^{\pi^{(t)}}(s,a) - Q^{\pi^{(t)}}(s,\pi^{(t)}(s))$. Then, we have

$$
\mathbb{E}_{(s,a)\sim d^{\pi^{(t)}}}\left[\left|\bar{\nu}^{(t)}(s,a) - A^{\pi^{(t)}}(s,a))\right|\right]
$$

$$
= \mathbb{E}_{(s,a)\sim d^{\pi^{(t)}}}\left[\left|\nu^{(t)}(s,a) - \nu^{(t)}(s,\pi^{(t)}(s)) - Q^{\pi^{(t)}}(s,a) + Q^{\pi^{(t)}}(s,\pi^{(t)}(s)))\right|\right] \tag{36}
$$

$$
\leq \mathbb{E}_{(s,a)\sim d^{\pi^{(t)}}}\left[\left|\nu^{(t)}(s,a) - Q^{\pi^{(t)}}(s,a)\right| + \left|Q^{\pi^{(t)}}(s,\pi^{(t)}(s)) - \nu^{(t)}(s,\pi^{(t)}(s))\right|\right] \tag{37}
$$

where (37) holds by the fact that $|x+y| \leq |x|+|y|$ for any $x,y \in \mathbb{R}$. Then, by linearity of expectation, we obtain

$$
\mathbb{E}_{(s,a)\sim d^{\pi^{(t)}}}\left[\left|\nu^{(t)}(s,a) - Q^{\pi^{(t)}}(s,a)\right| + \left|Q^{\pi^{(t)}}(s,\pi^{(t)}(s)) - \nu^{(t)}(s,\pi^{(t)}(s))\right|\right]
$$

$$
= \mathbb{E}_{(s,a)\sim d^{\pi^{(t)}}}\left[\left|\nu^{(t)}(s,a) - Q^{\pi^{(t)}}(s,a)\right|\right] + \mathbb{E}_{s\sim d^{\pi^{(t)}}}\left[\left|Q^{\pi^{(t)}}(s,\pi^{(t)}(s)) - \nu^{(t)}(s,\pi^{(t)}(s))\right|\right] \tag{38}
$$

$$
= \mathbb{E}_{(s,a)\sim d^{\pi^{(t)}}} 2\left[\left|\nu^{(t)}(s,a) - Q^{\pi^{(t)}}(s,a)\right|\right] \tag{39}
$$

where (39) holds by Jensen's inequality. Then, substituting the result from (39) into (34), we have

$$
\frac{1}{1-\gamma}\mathbb{E}_{(s,a)\sim d^{\pi^*}}\left[\bar{\nu}^{(t)}(s,a)\right] + \frac{1}{1-\gamma}C \cdot \mathbb{E}_{s,a\sim d^{\pi^{(t)}}}\left[\left|-\bar{\nu}^{(t)}(s,a) + A^{\pi^{(t)}}(s,a)\right|\right] \tag{40}
$$

$$
\leq \frac{1}{1-\gamma}\mathbb{E}_{(s,a)\sim d^{\pi^*}}\left[\bar{\nu}^{(t)}(s,a)\right] + \frac{2C}{1-\gamma} \cdot \mathbb{E}_{(s,a)\sim d^{\pi^{(t)}}}\left[\left|\nu^{(t)}(s,a) - Q^{\pi^{(t)}}(s,a)\right|\right] \tag{41}
$$

Next, summing over all iterations and combining with Lemma 4, we have

$$
\frac{1}{T}\sum_{t=1}^{T}\mathbb{E}_{s\sim\mu_{\text{tar}}}\left[V^{\pi^*}(s) - V^{\pi^{(t)}}(s)\right]
$$

$$
\leq \frac{[\log|\mathcal{A}_{\text{tar}}|+1]}{\sqrt{T}(1-\gamma)} + \frac{2C}{1-\gamma}\frac{1}{T}\sum_{t=1}^{T}\mathbb{E}_{(s,a)\sim d^{\pi^{(t)}}}\left[\left|\nu^{(t)}(s,a) - Q^{\pi^{(t)}}(s,a)\right|\right] \tag{42}
$$

$\square$

Recall that for any policy $\pi$, we use $d^\pi$ to denote the discounted state-action visitation distribution under policy $\pi$ in the target domain.

**Lemma 6.** *Under Algorithm 2, for any $t \in \mathbb{N}$, we have*

$$
\mathbb{E}_{(s,a)\sim d^{\pi^{(t)}}}\left[\left|f^t(s,a) - Q^{\pi^{(t)}}(s,a)\right|\right]
$$

$$
\leq \frac{1}{(1-\gamma)^2\mu_{tar,min}}\left[(1-\alpha(t))\mathbb{E}_{(s,a)\sim d^{\pi^{(t)}}}\left[\epsilon_{td}^{(t)}(s,a)\right] + \alpha(t)\mathbb{E}_{(s,a)\sim d^{\pi^{(t)}}}\left[\epsilon_{cd}(s,a;Q_{src},\phi^{(t)},\psi^{(t)},\pi^{(t)})\right]\right] \tag{43}
$$

*Proof.* Recall the definition of $f^{(t)} := (1-\alpha(t))Q_{\text{tar}}^{(t)}(s,a) + \alpha(t)Q_{\text{src}}(\phi^{(t)}(s),\psi^{(t)}(a))$, we have

$$
\mathbb{E}_{(s,a)\sim d^{\pi^{(t)}}}\left[\left|f^{(t)}(s,a) - Q^{\pi^{(t)}}(s,a)\right|\right]
$$

$$
= \mathbb{E}_{(s,a)\sim d^{\pi^{(t)}}}\left[\left|(1-\alpha(t))Q_{\text{tar}}^{(t)}(s,a) + \alpha(t)Q_{\text{src}}(\phi^{(t)}(s),\psi^{(t)}(a)) - Q^{\pi^{(t)}}(s,a)\right|\right] \tag{44}
$$

$$
= \mathbb{E}_{(s,a)\sim d^{\pi^{(t)}}}\left[\left|(1-\alpha(t))\left(Q_{\text{tar}}^{(t)}(s,a) - r_{\text{tar}}(s,a) + r_{\text{tar}}(s,a)\right)\right.\right.
$$

$$
\left.\left. + \alpha(t)\left(Q_{\text{src}}(\phi^{(t)}(s),\psi^{(t)}(a)) - r_{\text{tar}}(s,a) + r_{\text{tar}}(s,a)\right) - Q^{\pi^{(t)}}(s,a)\right|\right] \tag{45}
$$

$$
\begin{aligned}
= \mathbb{E}_{(s,a)\sim d^{\pi^{(t)}}} \Bigg[ \Bigg| & \big(1-\alpha(t)\big)\big(Q_{\text{tar}}^{(t)}(s,a) - r_{\text{tar}}(s,a) + r_{\text{tar}}(s,a) - \gamma \mathbb{E}_{\substack{s'\sim P_{\text{tar}}(\cdot|s,a)\\a'\sim\pi^{(t)}(\cdot|s')}}[Q_{\text{tar}}^{(t)}(s',a')] \\
& + \gamma \mathbb{E}_{\substack{s'\sim P_{\text{tar}}(\cdot|s,a)\\a'\sim\pi^{(t)}(\cdot|s')}}[Q_{\text{tar}}^{(t)}(s',a')]\big) + \alpha(t)\Big(Q_{\text{src}}(\phi^{(t)}(s),\psi^{(t)}(a)) - r_{\text{tar}}(s,a) + r_{\text{tar}}(s,a) \\
& - \gamma \mathbb{E}_{\substack{s'\sim P_{\text{tar}}(\cdot|s,a)\\a'\sim\pi^{(t)}(\cdot|s')}}[Q_{\text{src}}(\phi^{(t)}(s'),\psi^{(t)}(a'))] + \gamma \mathbb{E}_{\substack{s'\sim P_{\text{tar}}(\cdot|s,a)\\a'\sim\pi^{(t)}(\cdot|s')}}[Q_{\text{src}}(\phi^{(t)}(s'),\psi^{(t)}(a'))]\Big) \\
& - Q^{\pi^{(t)}}(s,a) \Bigg| \Bigg]
\end{aligned}
\tag{46}
$$

$$
\begin{aligned}
= \mathbb{E}_{(s,a)\sim d^{\pi^{(t)}}} \Bigg[ \Bigg| & \big(1-\alpha(t)\big)\big(Q_{\text{tar}}^{(t)}(s,a) - r_{\text{tar}}(s,a) - \gamma \mathbb{E}_{\substack{s'\sim P_{\text{tar}}(\cdot|s,a)\\a'\sim\pi^{(t)}(\cdot|s')}}[Q_{\text{tar}}^{(t)}(s',a')]\big) \\
& + \alpha(t)\Big(Q_{\text{src}}(\phi^{(t)}(s),\psi^{(t)}(a)) - r_{\text{tar}}(s,a) - \gamma \mathbb{E}_{\substack{s'\sim P_{\text{tar}}(\cdot|s,a)\\a'\sim\pi^{(t)}(\cdot|s')}}[Q_{\text{src}}(\phi^{(t)}(s'),\psi^{(t)}(a'))]\Big) \\
& + \big(1-\alpha(t)\big)\gamma \mathbb{E}_{\substack{s'\sim P_{\text{tar}}(\cdot|s,a)\\a'\sim\pi^{(t)}(\cdot|s')}}[Q_{\text{tar}}^{(t)}(s',a')] + \alpha(t)\gamma \mathbb{E}_{\substack{s'\sim P_{\text{tar}}(\cdot|s,a)\\a'\sim\pi^{(t)}(\cdot|s')}}[Q_{\text{src}}(\phi^{(t)}(s'),\psi^{(t)}(a'))] \\
& + r_{\text{tar}}(s,a) - Q^{\pi^{(t)}}(s,a) \Bigg| \Bigg]
\end{aligned}
\tag{47}
$$

$$
\begin{aligned}
= \mathbb{E}_{(s,a)\sim d^{\pi^{(t)}}} \Bigg[ \Bigg| & \big(1-\alpha(t)\big)\Big(Q_{\text{tar}}^{(t)}(s,a) - r_{\text{tar}}(s,a) - \gamma \mathbb{E}_{\substack{s'\sim P_{\text{tar}}(\cdot|s,a)\\a'\sim\pi^{(t)}(\cdot|s')}}[Q_{\text{tar}}^{(t)}(s',a')]\Big) \\
& + \alpha(t)\Big(Q_{\text{src}}(\phi^{(t)}(s),\psi^{(t)}(a)) - r_{\text{tar}}(s,a) - \gamma \mathbb{E}_{\substack{s'\sim P_{\text{tar}}(\cdot|s,a)\\a'\sim\pi^{(t)}(\cdot|s')}}[Q_{\text{src}}(\phi^{(t)}(s'),\psi^{(t)}(a'))]\Big) \\
& + \gamma \mathbb{E}_{\substack{s'\sim P_{\text{tar}}(\cdot|s,a)\\a'\sim\pi^{(t)}(\cdot|s')}}[f^{(t)}(s',a')] + r_{\text{tar}}(s,a) - Q^{\pi^{(t)}}(s,a) \Bigg| \Bigg],
\end{aligned}
\tag{48}
$$

where we obtain (45) by adding the dummy terms $\big(1-\alpha(t)\big)\big(-r_{\text{tar}}(s,a) + r_{\text{tar}}(s,a)\big)$ and $\alpha(t)\big(-r_{\text{tar}}(s,a) + r_{\text{tar}}(s,a)\big)$ to the inner part of (44), (46) is obtained by adding $\big(1-\alpha(t)\big)\big(-\gamma\mathbb{E}_{\substack{s'\sim P_{\text{tar}}(\cdot|s,a)\\a'\sim\pi^{(t)}(\cdot|s')}}[Q_{\text{tar}}^{(t)}(s',a')] + \gamma\mathbb{E}_{\substack{s'\sim P_{\text{tar}}(\cdot|s,a)\\a'\sim\pi^{(t)}(\cdot|s')}}[Q_{\text{tar}}^{(t)}(s',a')]\big)$ and $\alpha(t)\big(-\gamma\mathbb{E}_{\substack{s'\sim P_{\text{tar}}(\cdot|s,a)\\a'\sim\pi^{(t)}(\cdot|s')}}[Q_{\text{src}}(\phi^{(t)}(s'),\psi^{(t)}(a'))] + \gamma\mathbb{E}_{\substack{s'\sim P_{\text{tar}}(\cdot|s,a)\\a'\sim\pi^{(t)}(\cdot|s')}}[Q_{\text{src}}(\phi^{(t)}(s'),\psi^{(t)}(a'))]\big)$ to the inner part of (45), (47) holds by rearranging the terms in (46), and (48) holds by the definition of $f^{(t)}$. Then, by adding $\gamma\mathbb{E}_{\substack{s''\sim P_{\text{tar}}(\cdot|s,a)\\a''\sim\pi^{(t)}(\cdot|s'')}}[Q^{\pi^{(t)}}(s'',a'')] - \gamma\mathbb{E}_{\substack{s''\sim P_{\text{tar}}(\cdot|s,a)\\a''\sim\pi^{(t)}(\cdot|s'')}}[Q^{\pi^{(t)}}(s'',a'')]$ to the inner part of (48), we can rewrite (48) as

$$
\begin{aligned}
\mathbb{E}_{(s,a)\sim d^{\pi^{(t)}}} \Bigg[ \Bigg| & \big(1-\alpha(t)\big)\Big(Q_{\text{tar}}^{(t)}(s,a) - r_{\text{tar}}(s,a) - \gamma \mathbb{E}_{\substack{s'\sim P_{\text{tar}}(\cdot|s,a)\\a'\sim\pi^{(t)}(\cdot|s')}}[Q_{\text{tar}}^{(t)}(s',a')]\Big) \\
& + \alpha(t)\Big(Q_{\text{src}}(\phi^{(t)}(s),\psi^{(t)}(a)) - r_{\text{tar}}(s,a) - \gamma \mathbb{E}_{\substack{s'\sim P_{\text{tar}}(\cdot|s,a)\\a'\sim\pi^{(t)}(\cdot|s')}}[Q_{\text{src}}(\phi^{(t)}(s'),\psi^{(t)}(a'))]\Big) \\
& + \gamma \mathbb{E}_{\substack{s'\sim P_{\text{tar}}(\cdot|s,a)\\a'\sim\pi^{(t)}(\cdot|s')}}[f^{(t)}(s',a')] + r_{\text{tar}}(s,a) - Q^{\pi^{(t)}}(s,a) \\
& + \gamma \mathbb{E}_{\substack{s''\sim P_{\text{tar}}(\cdot|s,a)\\a''\sim\pi^{(t)}(\cdot|s'')}}[Q^{\pi^{(t)}}(s'',a'')] - \gamma \mathbb{E}_{\substack{s''\sim P_{\text{tar}}(\cdot|s,a)\\a''\sim\pi^{(t)}(\cdot|s'')}}[Q^{\pi^{(t)}}(s'',a'')] \Bigg| \Bigg]
\end{aligned}
\tag{49}
$$

$$\leq \mathbb{E}_{(s,a)\sim d^{\pi^{(t)}}} \left[ \left| (1-\alpha(t)) \left( Q_{\text{tar}}^{(t)}(s,a) - r_{\text{tar}}(s,a) - \gamma \mathbb{E}_{\substack{s'\sim P_{\text{tar}}(\cdot|s,a) \\ a'\sim \pi^{(t)}(\cdot|s')}}[Q_{\text{tar}}^{(t)}(s',a')] \right) \right| \right.$$

$$+ \left| \alpha(t) \left( Q_{\text{src}}(\phi^{(t)}(s), \psi^{(t)}(a)) - r_{\text{tar}}(s,a) - \gamma \mathbb{E}_{\substack{s'\sim P_{\text{tar}}(\cdot|s,a) \\ a'\sim \pi^{(t)}(\cdot|s')}}[Q_{\text{src}}(\phi^{(t)}(s'), \psi^{(t)}(a'))] \right) \right|$$

$$+ \left| \gamma \mathbb{E}_{\substack{s'\sim P_{\text{tar}}(\cdot|s,a) \\ a'\sim \pi^{(t)}(\cdot|s')}}[f^{(t)}(s',a')] + r_{\text{tar}}(s,a) - Q^{\pi^{(t)}}(s,a) \right| \tag{50}$$

$$+ \left. \left| \gamma \mathbb{E}_{\substack{s''\sim P_{\text{tar}}(\cdot|s,a) \\ a''\sim \pi^{(t)}(\cdot|s'')}}[Q^{\pi^{(t)}}(s'',a'')] - \gamma \mathbb{E}_{\substack{s''\sim P_{\text{tar}}(\cdot|s,a) \\ a''\sim \pi^{(t)}(\cdot|s'')}}[Q^{\pi^{(t)}}(s'',a'')] \right| \right]$$

$$\leq \mathbb{E}_{(s,a)\sim d^{\pi^{(t)}}} \left[ (1-\alpha(t)) \underbrace{\left| Q_{\text{tar}}^{(t)}(s,a) - r_{\text{tar}}(s,a) - \gamma \mathbb{E}_{\substack{s'\sim P_{\text{tar}}(\cdot|s,a) \\ a'\sim \pi^{(t)}(\cdot|s')}}[Q_{\text{tar}}^{(t)}(s',a')] \right|}_{=:\epsilon_{\text{td}}^{(t)}(s,a)} \right.$$

$$+ \alpha(t) \underbrace{\left| \left( Q_{\text{src}}(\phi^{(t)}(s), \psi^{(t)}(a)) - r_{\text{tar}}(s,a) - \gamma \mathbb{E}_{\substack{s'\sim P_{\text{tar}}(\cdot|s,a) \\ a'\sim \pi^{(t)}(\cdot|s')}}[Q_{\text{src}}(\phi^{(t)}(s'), \psi^{(t)}(a'))] \right) \right|}_{=:\epsilon_{\text{cd}}(s,a;Q_{\text{src}},\phi^{(t)},\psi^{(t)},\pi^{(t)})} \tag{51}$$

$$+ \left| \gamma \mathbb{E}_{\substack{s'\sim P_{\text{tar}}(\cdot|s,a) \\ a'\sim \pi^{(t)}(\cdot|s')}}[f^{(t)}(s',a')] - \gamma \mathbb{E}_{\substack{s''\sim P_{\text{tar}}(\cdot|s,a) \\ a''\sim \pi^{(t)}(\cdot|s'')}}[Q^{\pi^{(t)}}(s'',a'')] \right|$$

$$+ \left. \underbrace{\left| r_{\text{tar}}(s,a) - Q^{\pi^{(t)}}(s,a) + \gamma \mathbb{E}_{\substack{s''\sim P_{\text{tar}}(\cdot|s,a) \\ a''\sim \pi^{(t)}(\cdot|s'')}}[Q^{\pi^{(t)}}(s'',a'')] \right|}_{=0} \right]$$

$$\leq \mathbb{E}_{(s,a)\sim d^{\pi^{(t)}}} \left[ (1-\alpha(t))\epsilon_{\text{td}}^{(t)}(s,a) + \alpha(t)\epsilon_{\text{cd}}(s,a;Q_{\text{src}},\phi^{(t)},\psi^{(t)},\pi^{(t)}) \right.$$

$$+ \left. \gamma \mathbb{E}_{\substack{s'\sim P_{\text{tar}}(\cdot|s,a) \\ a'\sim \pi^{(t)}(\cdot|s'')}}\left[ \left| f^{(t)}(s',a') - Q^{\pi^{(t)}}(s',a') \right| \right] \right] \tag{52}$$

where (50) holds by triangle inequality, (51) holds by the facts that $0 \leq \alpha(t) \leq 1$ and $0 \leq 1 - \alpha(t) \leq 1$, (52) holds by coupling $(s',a')$ and $(s'',a'')$ and applying Bellman expectation equation as well as the definitions that $\epsilon_{\text{td}}^{(t)}(s,a) := \left| Q_{\text{tar}}^{(t)}(s,a) - r_{\text{tar}}(s,a) - \gamma \mathbb{E}_{\substack{s'\sim P_{\text{tar}}(\cdot|s,a) \\ a'\sim \pi^{(t)}(\cdot|s')}}[Q_{\text{tar}}^{(t)}(s',a')] \right|$ and $\epsilon_{\text{cd}}(s,a;Q_{\text{src}},\phi^{(t)},\psi^{(t)},\pi^{(t)}) := \left| Q_{\text{src}}(\phi^{(t)}(s),\psi^{(t)}(a)) - r_{\text{tar}}(s,a) - \gamma \mathbb{E}_{\substack{s'\sim P_{\text{tar}}(\cdot|s,a) \\ a'\sim \pi^{(t)}(\cdot|s')}}[Q_{\text{src}}(\phi^{(t)}(s'),\psi^{(t)}(a'))] \right|$. By recursively applying the procedure from (44) to (52) to $\left| f^{(t)}(s',a') - Q^{\pi^{(t)}}(s',a') \right|$, we obtain a bound on $\mathbb{E}_{(s,a)\sim d^{\pi^{(t)}}}\left[ \left( f^{(t)}(s,a) - Q^{\pi^{(t)}}(s,a) \right)^2 \right]$ as follows:

$$\mathbb{E}_{(s,a)\sim d^{\pi^{(t)}}} \left[ \left| f^{(t)}(s,a) - Q^{\pi^{(t)}}(s,a) \right| \right]$$

$$\leq \mathbb{E}_{(s,a)\sim d^{\pi^{(t)}}} \left[ \left| (1-\alpha(t))\epsilon_{\text{td}}^{(t)}(s,a) + \alpha(t)\epsilon_{\text{cd}}(s,a;Q_{\text{src}},\phi^{(t)},\psi^{(t)},\pi^{(t)}) \right. \right.$$

$$+ \left. \left. \gamma \mathbb{E}_{\substack{s'\sim P_{\text{tar}}(\cdot|s,a) \\ a'\sim \pi^{(t)}(\cdot|s')}}\left[ \left| f^{(t)}(s',a') - Q^{\pi^{(t)}}(s',a') \right| \right] \right| \right] \tag{53}$$

$$\leq \mathbb{E}_{(s,a)\sim d^{\pi^{(t)}}}\left[\left|\left|\left(1-\alpha(t)\right)\epsilon_{\text{td}}^{(t)}(s,a)+\alpha(t)\epsilon_{\text{cd}}(s,a;Q_{\text{src}},\phi^{(t)},\psi^{(t)},\pi^{(t)})\right.\right.\right.$$

$$+\gamma\mathbb{E}_{\substack{s'\sim P_{\text{tar}}(\cdot|s,a)\\ a'\sim\pi^{(t)}(\cdot|s')}}\left[\left(1-\alpha(t)\right)\epsilon_{\text{td}}^{(t)}(s',a')+\alpha(t)\epsilon_{\text{cd}}(s',a';Q_{\text{src}},\phi^{(t)},\psi^{(t)},\pi^{(t)})\right. \tag{54}$$

$$\left.\left.\left.+\gamma\mathbb{E}_{\substack{s''\sim P_{\text{tar}}(\cdot|s',a')\\ a''\sim\pi^{(t)}(\cdot|s'')}}\left[\left|f^{(t)}(s'',a'')-Q^{\pi^{(t)}}(s'',a'')\right|\right]\right]\right|\right]$$

$$\leq \mathbb{E}_{(s,a)\sim d^{\pi^{(t)}}}\left[\left|\left(1-\alpha(t)\right)\epsilon_{\text{td}}^{(t)}(s,a)+\alpha(t)\epsilon_{\text{cd}}(s,a;Q_{\text{src}},\phi^{(t)},\psi^{(t)},\pi^{(t)})\right.\right.$$

$$+\frac{1}{(1-\gamma)\mu_{\text{tar,min}}}\left(\gamma\left(1-\alpha(t)\right)\epsilon_{\text{td}}^{(t)}(s,a)+\gamma\alpha(t)\epsilon_{\text{cd}}(s,a;Q_{\text{src}},\phi^{(t)},\psi^{(t)},\pi^{(t)})\right. \tag{55}$$

$$\left.\left.+\gamma^2\left(1-\alpha(t)\right)\epsilon_{\text{td}}^{(t)}(s,a)+\gamma^2\alpha(t)\epsilon_{\text{cd}}(s,a;Q_{\text{src}},\phi^{(t)},\psi^{(t)},\pi^{(t)})+\cdots\right)\right|\right]$$

$$\leq \frac{1}{(1-\gamma)^2\mu_{\text{tar,min}}}\mathbb{E}_{(s,a)\sim d^{\pi^{(t)}}}\left[\left|\left(1-\alpha(t)\right)\epsilon_{\text{td}}^{(t)}(s,a)+\alpha(t)\epsilon_{\text{cd}}(s,a;Q_{\text{src}},\phi^{(t)},\psi^{(t)},\pi^{(t)})\right|\right] \tag{56}$$

$$= \frac{1}{(1-\gamma)^2\mu_{\text{tar,min}}}\left[(1-\alpha(t))\mathbb{E}_{(s,a)\sim d^{\pi^{(t)}}}\left[\epsilon_{\text{td}}^{(t)}(s,a)\right]+\alpha(t)\mathbb{E}_{(s,a)\sim d^{\pi^{(t)}}}\left[\epsilon_{\text{cd}}(s,a;Q_{\text{src}},\phi^{(t)},\psi^{(t)},\pi^{(t)})\right]\right] \tag{57}$$

where (54) follows by applying the procedure from (44)–(52) to $f^{(t)}(s',a')-Q^{\pi^{(t)}}(s',a')$, (55) follows by applying the same procedure to subsequent time steps with importance sampling using the ratio bound in Lemma 3 and the same dummy variables $(s,a)$ for all subsequent state-action pairs, and (57) follows from summing an infinite geometric series. $\square$

## B  PROOFS OF THE PROPOSITIONS

We first present the proof of Proposition 3 in Appendix B.1 and then establish Proposition 2 and 1 by a similar argument in Appendix B.2.

### B.1  PROOF OF PROPOSITION 3

**Proposition 3.** *(Average Sub-Optimality) Under the QAvatar in Algorithm 2 and Assumption 1, the average sub-optimality over $T$ iterations can be upper bounded as*

$$\frac{1}{T}\sum_{t=1}^T\mathbb{E}_{s\sim\mu_{tar}}\left[V^{\pi^*}(s)-V^{\pi^{(t)}}(s)\right]$$

$$\leq \underbrace{\frac{[\log|\mathcal{A}_{tar}|+1]}{\sqrt{T}(1-\gamma)}}_{(a)}+\underbrace{\frac{C_0}{T}\sum_{t=1}^T\mathbb{E}_{(s,a)\sim d^{\pi^{(t)}}}\left[\left|f^{(t)}(s,a)-Q^{\pi^{(t)}}(s,a)\right|\right]}_{(b)} \tag{7}$$

$$\leq \underbrace{\frac{[\log|\mathcal{A}_{tar}|+1]}{\sqrt{T}(1-\gamma)}}_{(a)}+\underbrace{\frac{C_1}{T}\sum_{t=1}^T\left(\alpha(t)\|\epsilon_{cd}(Q_{src},\phi^{(t)},\psi^{(t)})\|_{d^{\pi^{(t)}}}+(1-\alpha(t))\|\epsilon_{td}^{(t)}\|_{d^{\pi^{(t)}}}\right)}_{(c)}, \tag{8}$$

*where $C_0 := 2C_{\pi^*}/(1-\gamma)$ and $C_1 := 2C_{\pi^*}/((1-\gamma)^3\mu_{tar,\,min})$.*

*Proof.* Using Lemma 5 and setting $\nu^{(t)}=f^{(t)}$, we have

$$\frac{1}{T}\sum_{t=1}^T\mathbb{E}_{s\sim\mu_{\text{tar}}}\left[V^{\pi^*}(s)-V^{\pi^{(t)}}(s)\right]$$

$$\leq \frac{[\log|\mathcal{A}_{\text{tar}}|+1]}{\sqrt{T}(1-\gamma)}+\frac{2C}{1-\gamma}\frac{1}{T}\sum_{t=1}^T\mathbb{E}_{(s,a)\sim d^{\pi^{(t)}}}\left[\left|f^{(t)}(s,a)-Q^{\pi^{(t)}}(s,a)\right|\right] \tag{58}$$

This establishes the first inequality. Furthermore, recall the definitions of $\epsilon_{td}^{(t)}(s,a)$ and $\epsilon_{cd}(s,a; Q_{src}, \phi, \psi, \pi)$ as

$$\epsilon_{td}^{(t)}(s,a) := \left| Q_{tar}^{(t)}(s,a) - r_{tar}(s,a) - \gamma \mathbb{E}_{\substack{s' \sim P_{tar}(\cdot|s,a) \\ a' \sim \pi^{(t)}(\cdot|s')}}[Q_{tar}^{(t)}(s',a')] \right|, \tag{59}$$

$$\epsilon_{cd}(s,a; Q_{src}, \phi, \psi, \pi) := \left| Q_{src}(\phi(s), \psi(a)) - r_{tar}(s,a) - \gamma \mathbb{E}_{s' \sim P_{tar}(\cdot|s,a), a' \sim \pi(\cdot|s')}[Q_{src}(\phi(s'), \psi(a'))] \right|. \tag{60}$$

We also define the weighted $\ell_1$ norm under state-action distribution induced by any policy $\pi$ as

$$\|\epsilon_{td}^{(t)}\|_{d^\pi} := \mathbb{E}_{(s,a) \sim d^\pi}\left[\epsilon_{td}^{(t)}(s,a)\right], \tag{61}$$

$$\|\epsilon_{cd}(Q_{src}, \phi^{(t)}, \psi^{(t)})\|_{d^\pi} := \mathbb{E}_{(s,a) \sim d^\pi}\left[\epsilon_{cd}(s,a; Q_{src}, \phi^{(t)}, \psi^{(t)}, \pi)\right]. \tag{62}$$

For the second inequality, by Lemma 6, we have

$$\frac{1}{T}\sum_{t=1}^{T}\mathbb{E}_{s \sim \mu_{tar}}\left[V^{\pi^*}(s) - V^{\pi^{(t)}}(s)\right]$$

$$\leq \frac{[\log|\mathcal{A}_{tar}|+1]}{\sqrt{T}(1-\gamma)} + \frac{2C}{1-\gamma}\frac{1}{T}\sum_{t=1}^{T}\mathbb{E}_{(s,a) \sim d^{\pi^{(t)}}}\left[\left|f^{(t)}(s,a) - Q^{\pi^{(t)}}(s,a)\right|\right] \tag{63}$$

$$\leq \frac{[\log|\mathcal{A}_{tar}|+1]}{\sqrt{T}(1-\gamma)} + \frac{2C}{(1-\gamma)^3\mu_{tar,min}}\frac{1}{T}\sum_{t=1}^{T}\left[(1-\alpha(t))\|\epsilon_{td}^{(t)}\|_{d^{\pi^{(t)}}} + \alpha(t)\|\epsilon_{cd}(Q_{src}, \phi^{(t)}, \psi^{(t)})\|_{d^{\pi^{(t)}}}\right] \tag{64}$$

This completes the proof of Proposition 3. Additionally, by choosing $\alpha(t) = \frac{\|\epsilon_{td}^{(t)}\|_{d^{\pi^{(t)}}}}{\|\epsilon_{cd}(Q_{src}, \phi^{(t)}, \psi^{(t)})\|_{d^{\pi^{(t)}}} + \|\epsilon_{td}^{(t)}\|_{d^{\pi^{(t)}}}}$ (as discussed in Section 4), we have

$$\frac{1}{T}\sum_{t=1}^{T}\mathbb{E}_{s \sim \mu_{tar}}\left[V^{\pi^*}(s) - V^{\pi^{(t)}}(s)\right]$$

$$\leq \frac{2}{(1-\gamma)^2}\sqrt{\frac{\log(\mathcal{A}_{tar})}{T}} + \frac{4\sqrt{2}C}{(1-\gamma)^3\mu_{tar,min}}\frac{1}{T}\sum_{t=1}^{T}\frac{\|\epsilon_{cd}(Q_{src}, \phi^{(t)}, \psi^{(t)})\|_{d^{\pi^{(t)}}} \cdot \|\epsilon_{td}^{(t)}\|_{d^{\pi^{(t)}}}}{\|\epsilon_{cd}(Q_{src}, \phi^{(t)}, \psi^{(t)})\|_{d^{\pi^{(t)}}} + \|\epsilon_{td}^{(t)}\|_{d^{\pi^{(t)}}}}. \tag{65}$$

$\square$

## B.2 PROOF OF PROPOSITION 2

**Proposition 2.** *Under the DQT method in Algorithm 1 and Assumption 1, the average sub-optimality over $T$ iterations is upper bounded as*

$$\frac{1}{T}\sum_{t=1}^{T}\mathbb{E}_{s \sim \mu_{tar}}\left[V^{\pi^*}(s) - V^{\pi^{(t)}}(s)\right] \leq \underbrace{\frac{[\log|\mathcal{A}_{tar}|+1]}{\sqrt{T}(1-\gamma)}}_{(a)} + \underbrace{\frac{C_0}{T}\sum_{t=1}^{T}\left\|\left|Q_{src}(\phi^{(t)}, \psi^{(t)}) - Q^{\pi^{(t)}}\right|\right\|_{d^{\pi^{(t)}}}}_{(b)}$$

$$\leq \underbrace{\frac{[\log|\mathcal{A}_{tar}|+1]}{\sqrt{T}(1-\gamma)}}_{(a)} + \underbrace{\frac{C_1}{T}\sum_{t=1}^{T}\|\epsilon_{cd}(Q_{src}, \phi^{(t)}, \psi^{(t)})\|_{d^{\pi^{(t)}}}}_{(c)}, \tag{4}$$

*where $C_0 := 2C_{\pi^*}/(1-\gamma)$ and $C_1 := 2C_{\pi^*}/((1-\gamma)^3\mu_{tar,min})$.*

*Proof.* Notably, since the Proposition 2 is a special case of Proposition 3, we can simply follow all the steps taken for Proposition 3 and set $\alpha(t) = 1$ for all $t$ to establish Proposition 2. More specifically,

we can replace $f^{(t)}(s,a)$ with $Q_{\text{src}}(\phi^{(t)}(s), \psi^{(t)}(a))$. Accordingly, under $\alpha(t) = 1$ for all $t$, Lemma 6 can be simply rewritten as

$$\mathbb{E}_{(s,a)\sim d^{\pi^{(t)}}}\left[\left|Q_{\text{src}}(\phi^{(t)}(s), \psi^{(t)}(a)) - Q^{\pi^{(t)}}(s,a)\right|\right] \tag{66}$$

$$\leq \frac{1}{(1-\gamma)^2 \mu_{\text{tar,min}}} \mathbb{E}_{(s,a)\sim d^{\pi^{(t)}}}\left[\epsilon_{\text{cd}}(s,a; Q_{\text{src}}, \phi^{(t)}, \psi^{(t)}, \pi^{(t)})\right]. \tag{67}$$

Similarly, Lemma 5 can be be rewritten as

$$\frac{1}{T}\sum_{t=1}^{T}\mathbb{E}_{s\sim\mu_{\text{tar}}}\left[V^{\pi^*}(s) - V^{\pi^{(t)}}(s)\right]$$

$$\leq \frac{[\log|\mathcal{A}_{\text{tar}}|+1]}{\sqrt{T}(1-\gamma)} + \frac{2C_{\pi^*}}{1-\gamma}\frac{1}{T}\sum_{t=1}^{T}\mathbb{E}_{(s,a)\sim d^{\pi^{(t)}}}\left[\left|Q_{\text{src}}(\phi^{(t)}(s), \psi^{(t)}(a)) - Q^{\pi^{(t)}}(s,a)\right|\right]$$

From the combination of the two results,

$$\frac{1}{T}\sum_{t=1}^{T}\mathbb{E}_{s\sim\mu_{\text{tar}}}\left[V^{\pi^*}(s) - V^{\pi^{(t)}}(s)\right] \leq \frac{[\log|\mathcal{A}_{\text{tar}}|+1]}{\sqrt{T}(1-\gamma)} + \frac{2C_{\pi^*}}{(1-\gamma)^3\mu_{\text{tar,min}}T}\sum_{t=1}^{T}\left\|\epsilon_{\text{cd}}(Q_{\text{src}}, \phi^{(t)}, \psi^{(t)})\right\|_{d^{\pi^{(t)}}}. \tag{68}$$

$\square$

### B.3 PROOF OF PROPOSITION 1

**Proposition 1.** *Under the tabular and approximate-Q settings, and Assumption 1, the average sub-optimality of Q-NPG over $T$ iterations is upper bounded by*

$$\frac{1}{T}\sum_{t=1}^{T}\mathbb{E}_{s\sim\mu_{\text{tar}}}\left[V^{\pi^*}(s) - V^{\pi^{(t)}}(s)\right]$$

$$\leq \underbrace{\frac{[\log|\mathcal{A}_{\text{tar}}|+1]}{\sqrt{T}(1-\gamma)}}_{(a)} + \underbrace{\frac{C_0}{T}\sum_{t=1}^{T}\left\|\left|Q_{\text{tar}}^{(t)} - Q^{\pi^{(t)}}\right|\right\|_{d^{\pi^{(t)}}}}_{(b)} \leq \underbrace{\frac{[\log|\mathcal{A}_{\text{tar}}|+1]}{\sqrt{T}(1-\gamma)}}_{(a)} + \underbrace{\frac{C_1}{T}\sum_{t=1}^{T}\left\|\epsilon_{\text{td}}^{(t)}\right\|_{d^{\pi^{(t)}}}}_{(c)}, \tag{2}$$

*where $C_0 := 2C_{\pi^*}/(1-\gamma)$ and $C_1 := 2C_{\pi^*}/((1-\gamma)^3\mu_{\text{tar, min}})$.*

*Proof.* Notably, since the Proposition 1 is a special case of Proposition 3, we can simply follow all the steps taken for Proposition 3 and set $\alpha(t) = 0$ for all $t$ to establish Proposition 1. More specifically, we can replace $f^{(t)}(s,a)$ with $Q_{\text{tar}}^{(t)}(s,a)$. Accordingly, under $\alpha(t) = 0$ for all $t$, Lemma 6 can be simply rewritten as

$$\mathbb{E}_{(s,a)\sim d^{\pi^{(t)}}}\left[\left|Q_{\text{tar}}^{(t)}(s,a) - Q^{\pi^{(t)}}(s,a)\right|\right] \tag{69}$$

$$\leq \frac{1}{(1-\gamma)^2 \mu_{\text{tar,min}}}\mathbb{E}_{(s,a)\sim d^{\pi^{(t)}}}\left[\epsilon_{\text{td}}(s,a)\right]. \tag{70}$$

Similarly, Lemma 5 can be be rewritten as

$$\frac{1}{T}\sum_{t=1}^{T}\mathbb{E}_{s\sim\mu_{\text{tar}}}\left[V^{\pi^*}(s) - V^{\pi^{(t)}}(s)\right]$$

$$\leq \frac{[\log|\mathcal{A}_{\text{tar}}|+1]}{\sqrt{T}(1-\gamma)} + \frac{2C_{\pi^*}}{1-\gamma}\frac{1}{T}\sum_{t=1}^{T}\mathbb{E}_{(s,a)\sim d^{\pi^{(t)}}}\left[\left|Q_{\text{tar}}^{(t)}(s,a) - Q^{\pi^{(t)}}(s,a)\right|\right]$$

From the combination of the two results,

$$\frac{1}{T}\sum_{t=1}^{T}\mathbb{E}_{s\sim\mu_{\text{tar}}}\left[V^{\pi^*}(s) - V^{\pi^{(t)}}(s)\right] \leq \frac{[\log|\mathcal{A}_{\text{tar}}|+1]}{\sqrt{T}(1-\gamma)} + \frac{2C_{\pi^*}}{(1-\gamma)^3\mu_{\text{tar,min}}T}\sum_{t=1}^{T}\left\|\epsilon_{\text{td}}\right\|_{d^{\pi^{(t)}}}. \tag{71}$$

$\square$

### B.4 SAMPLE COMPLEXITY BOUND

To convert the convergence result in Proposition 3 into sample complexity guarantees, we adopt the standard technique of the least squares generalization bound for sequential function estimation (Agarwal et al., 2019; Song et al., 2023) as follows.

**Lemma 7** (Least squares generalization bound, Lemma 3 in (Song et al., 2023)). *Consider a sequential function estimation setting with an instance space $\mathcal{X}$ and target space $\mathcal{Y}$. Let $R > 0$, $\delta \in (0, 1)$. Let $\mathcal{H} : \mathcal{X} \to [-R, R]$ be a class of real-valued functions. Let $\mathcal{D} = \{(x_1, y_1), \ldots, (x_M, y_M)\}$ be a dataset of $M$ points where $x_m \sim \rho_m := \rho_m(x_{1:m-1}, y_{1:m-1})$, and $y_m$ is sampled via the conditional probability $p(\cdot|x_m)$: $y_m \sim p(\cdot|x_m) := h^*(x_m) + \varepsilon_m$. Suppose the following conditions hold:*

*1. $h^*$ satisfies approximate realizability, i.e., $\inf_{h \in \mathcal{H}} \frac{1}{M} \sum_{m=1}^{M} \mathbb{E}_{x \sim \rho_m}\left[(h^*(x) - h(x))^2\right] \leq \kappa$.*

*2. $\{\varepsilon_m\}_{m=1}^{M}$ are independent random variables such that $\mathbb{E}[y_m|x_m] = h^*(x_m)$.*

*3. $\max_m |y_m| \leq R$ and $\max_x |h^*(x)| \leq R$.*

*Then, the least-squares solution $\hat{h} := \arg\min_{h \in \mathcal{H}} \sum_{m=1}^{M} (h(x_m) - y_m)^2$ satisfies that with probability at least $1 - \delta$,*

$$\sum_{m=1}^{M} \mathbb{E}_{x \sim \rho_m}\left[(\hat{h}(x) - h^*(x))^2\right] \leq 3\kappa M + 256 R^2 \log\left(\frac{2|\mathcal{H}|}{\delta}\right). \tag{72}$$

We define

$$\kappa_{\text{tar}}^{(t)} := \inf_{Q^{(t)} \in \mathcal{Q}} \mathbb{E}_{(s,a) \sim d^{\pi^{(t)}}}\left[\left|r_{\text{tar}}(s, a) + \gamma \mathbb{E}_{s' \sim P_{\text{tar}}, a' \sim \pi^{(t)}}[Q^{(t)}(s', a')] - Q^{(t)}(s, a)\right|^2\right], \tag{73}$$

where $\mathcal{Q}$ denotes a (finite) class of possible action-value functions. For ease of exposition, we suppose $\kappa_{\text{tar}}^{(t)} \leq \kappa_{\text{tar,max}}$, for all $t$. Note that this can be achieved since $\kappa_{\text{tar,max}}$ can be configured by choosing the function class $\mathcal{Q}$. We also let $\mathcal{F}$ denote the product of the (finite) classes of possible inter-domain mappings $\phi$ and $\psi$.

**Definition 6** (Cross-Domain Realizability). *A source-domain critic $Q_{src}$ is said to satisfy the cross-domain realizability under a target-domain policy $\pi$ if there exists a pair of inter-domain mappings $(\phi, \psi)$ in $\mathcal{F}$ such that $\|\epsilon_{cd}(Q_{src}, \phi, \psi)\|_{d^\pi} = 0$.*

**Corollary 1.** *Consider the setting of Proposition 3 and assume a source-domain critic with cross-domain realizability for all $t$. In order to obtain an $\epsilon$-optimal policy in $\mathcal{M}_{tar}$ with probability at least $1 - \delta$, the number of target-domain samples needed under QAvatar is*

$$\mathcal{O}\left(\left(\frac{[\log |\mathcal{A}_{tar}| + 1]}{(1 - \gamma)}\right)^2 \frac{1}{\epsilon^2} \cdot \min\left\{\frac{C_1^2 C_{cd}}{\epsilon^2}, \frac{C_{tar}}{\left[\frac{\epsilon^2}{C_1^2} - 3\kappa_{tar,\text{max}}\right]^+}\right\}\right) \tag{74}$$

*where $C_{tar} := \frac{1024}{(1-\gamma)^2} \log\left(\frac{4|\mathcal{Q}|}{\delta}\right)$ and $C_{cd} := \frac{1024}{(1-\gamma)^2} \log(\frac{4|\mathcal{F}|}{\delta})$. Moreover, to obtain an $\epsilon$-optimal policy in $\mathcal{M}_{tar}$ with probability at least $1 - \delta$, the number of target-domain samples needed under Q-NPG is*

$$\mathcal{O}\left(\left(\frac{[\log |\mathcal{A}_{tar}| + 1]}{(1 - \gamma)}\right)^2 \frac{1}{\epsilon^2} \cdot \frac{C_{tar}}{\left[\frac{\epsilon^2}{C_1^2} - 3\kappa_{tar,\text{max}}\right]^+}\right) \tag{75}$$

*Proof.* To establish the sample complexity bound, we connect the sub-optimality gap in Proposition 3 with the number of samples needed in learning the Q function and the inter-domain mappings. To begin with, we bound the $\|\epsilon_{\text{td}}^{(t)}\|_{d^{\pi^{(t)}}}$ as follows: Let $Q : \mathcal{S}_{\text{tar}} \times \mathcal{A}_{\text{tar}} \to \mathbb{R}$ denote an action-value function in the target domain. Recall that we use $r_{\text{tar}}$ and $P_{\text{tar}}$ to denote the reward function and the transition kernel of the target domain, respectively. For ease of exposition, define two helper functions $\zeta : \mathcal{S}_{\text{tar}} \times \mathcal{A}_{\text{tar}} \to \mathbb{R}$ and $\zeta^* : \mathcal{S}_{\text{tar}} \times \mathcal{A}_{\text{tar}} \to \mathbb{R}$ as

$$\zeta(s, a; Q, \pi) := r_{\text{tar}}(s, a) + \gamma \mathbb{E}_{s' \sim P_{\text{tar}}, a' \sim \pi}[Q(s', a')] - Q(s, a), \tag{76}$$

$$\zeta^*(s, a; \pi) := r_{\text{tar}}(s, a) + \gamma \mathbb{E}_{s' \sim P_{\text{tar}}, a' \sim \pi}[Q^\pi(s', a')] - Q^\pi(s, a). \tag{77}$$

By the Bellman expectation equations, we know $\zeta^*(s,a;\pi) = 0$, for any $(s,a)$ and any target-domain policy $\pi$. Recall from Algorithm 2, in each iteration $t$, we sample a batch $\mathcal{D}^{(t)}$ of $N_{\text{tar}}$ target-domain samples to obtain the $Q_{\text{tar}}^{(t)}$ by minimizing the empirical TD loss, i.e.,

$$Q_{\text{tar}}^{(t)} = \arg \min_{Q^{(t)} \in \mathcal{Q}} \sum_{(s,a,r,s') \in \mathcal{D}^{(t)}} \left[ \left| r + \gamma \mathbb{E}_{a' \sim \pi^{(t)}}[Q^{(t)}(s',a')] - Q^{(t)}(s,a) \right|^2 \right]. \tag{78}$$

Now we are ready to reinterpret (76)-(78) through the lens of Lemma 7. Let $\zeta(s,a;Q,\pi)$ and $\zeta^*(s,a;\pi)$ play the roles of $h(x)$ and $h^*(x)$. For each data sample $(s,a,r,s')$, by treating $\left( \zeta(s,a;Q,\pi) - (r + \gamma \mathbb{E}_{a' \sim \pi^{(t)}}[Q^{(t)}(s',a')] - Q^{(t)}(s,a)) \right)$ as the noise term $\epsilon_m$ in Lemma 7, we know $Q_{\text{tar}}^{(t)}$ actually plays the role of the least-squares solution (i.e., , $\hat{h}$ in Lemma 7). Through this interpretation, we know that the three conditions in Lemma 7 are satisfied with $\kappa = \kappa_{\text{tar}}$ and $R = 2/(1-\gamma)$. By applying Lemma 7 and Jensen's inequality, the result in (72) implies that with probability at least $1 - \delta/2$,

$$\|\epsilon_{\text{td}}^{(t)}\|_{d^{\pi^{(t)}}}^2 \le 3\kappa_{\text{tar}}^{(t)} + \frac{C_{\text{tar}}}{N_{\text{tar}}}, \tag{79}$$

where $C_{\text{tar}} := \frac{1024}{(1-\gamma)^2} \log\left( \frac{4|\mathcal{Q}|}{\delta} \right)$. Similarly, we proceed to bound the $\|\epsilon_{\text{cd}}(Q_{\text{src}}, \phi^{(t)}, \psi^{(t)})\|_{d^{\pi^{(t)}}}$ as follows. Define an additional helper function $\zeta_{\text{cd}} : \mathcal{S}_{\text{tar}} \times \mathcal{A}_{\text{tar}} \to \mathbb{R}$ as

$$\zeta_{\text{cd}}(s,a;Q_{\text{src}}, \pi, \phi, \psi) := r_{\text{tar}}(s,a) + \gamma \mathbb{E}_{s' \sim P_{\text{tar}}, a' \sim \pi}[Q_{\text{src}}(\phi(s'), \psi(a'))] - Q_{\text{src}}(\phi(s), \psi(a)). \tag{80}$$

Recall that in each iteration $t$, we also use the batch $\mathcal{D}^{(t)}$ of $N_{\text{tar}}$ target-domain samples to obtain the $\phi^{(t)}, \psi^{(t)}$ by minimizing the empirical cross-domain Bellman loss, i.e.,

$$\phi^{(t)}, \psi^{(t)} \leftarrow \arg \min_{\phi, \psi} \mathcal{L}_{\text{CD}}(\phi, \psi; Q_{\text{src}}, \pi^{(t)}, \mathcal{D}_{\text{tar}}^{(t)}). \tag{81}$$

In each iteration $t$, we let $\phi_*^{(t)}$ and $\psi_*^{(t)}$ denote the inter-domain mappings that yield $\|\epsilon_{\text{cd}}(Q_{\text{src}}, \phi_*^{(t)}, \psi_*^{(t)})\|_{d^{\pi^{(t)}}} = 0$. We know $\zeta_{\text{cd}}(s,a;Q_{\text{src}}, \pi^{(t)}, \phi_*^{(t)}, \psi_*^{(t)}) = 0$, for all $(s,a)$.

Now we are ready to reinterpret (80) through the lens of Lemma 7. Let $\zeta_{\text{cd}}(s,a;Q_{\text{src}}, \pi, \phi, \psi)$ and $\zeta_{\text{cd}}(s,a;Q_{\text{src}}, \pi^{(t)}, \phi_*^{(t)}, \psi_*^{(t)})$ play the roles of $h(x)$ and $h^*(x)$, respectively. For each data sample $(s,a,r,s')$, by treating $\left( \zeta_{\text{cd}}(s,a;Q_{\text{src}}, \pi, \phi, \psi) - (r + \gamma \mathbb{E}_{a' \sim \pi^{(t)}}[Q_{\text{src}}^{(t)}(\phi(s'), \psi(a'))] - Q_{\text{src}}^{(t)}(\phi(s), \psi(a))) \right)$ as the noise term $\epsilon_m$ in Lemma 7, we know that $\phi^{(t)}$ and $\psi^{(t)}$ actually play the role of the least-squares solution (i.e., , $\hat{h}$ in Lemma 7). Again, through the above interpretation, we know that the three conditions in Lemma 7 hold with $\kappa = 0$ and $R = 2/(1-\gamma)$. By applying Lemma 7 and Jensen's inequality, the result in (72) implies that with probability at least $1 - \delta/2$,

$$\|\epsilon_{\text{cd}}(Q_{\text{src}}, \phi^{(t)}, \psi^{(t)})\|_{d^{\pi^{(t)}}}^2 \le \frac{C_{\text{cd}}}{N_{\text{tar}}}, \tag{82}$$

where $C_{\text{cd}} := \frac{1024}{(1-\gamma)^2} \log(\frac{4|\mathcal{F}|}{\delta})$. We are ready to put everything together. We can rewrite the sub-optimality gap in Proposition 3 as follows. With probability at least $1 - \delta$,

$$\frac{1}{T} \sum_{t=1}^{T} \mathbb{E}_{s \sim \mu_{\text{tar}}} \left[ V^{\pi^*}(s) - V^{\pi^{(t)}}(s) \right]$$

$$\le \frac{[\log |\mathcal{A}_{\text{tar}}| + 1]}{\sqrt{T}(1-\gamma)} + \frac{C_1}{T} \sum_{t=1}^{T} \left( \alpha(t) \|\epsilon_{\text{cd}}(Q_{\text{src}}, \phi^{(t)}, \psi^{(t)})\|_{d^{\pi^{(t)}}} + (1 - \alpha(t)) \|\epsilon_{\text{td}}^{(t)}\|_{d^{\pi^{(t)}}} \right), \tag{83}$$

$$\le \frac{[\log |\mathcal{A}_{\text{tar}}| + 1]}{\sqrt{T}(1-\gamma)} + \frac{C_1}{T} \sum_{t=1}^{T} \left( \alpha(t) \sqrt{\frac{C_{\text{cd}}}{N_{\text{tar}}}} + (1 - \alpha(t)) \sqrt{3\kappa_{\text{tar}}^{(t)} + \frac{C_{\text{tar}}}{N_{\text{tar}}}} \right), \tag{84}$$

$$\le \frac{[\log |\mathcal{A}_{\text{tar}}| + 1]}{\sqrt{T}(1-\gamma)} + \frac{C_1}{T} \sum_{t=1}^{T} \min \left\{ \sqrt{\frac{C_{\text{cd}}}{N_{\text{tar}}}}, \sqrt{3\kappa_{\text{tar}}^{(t)} + \frac{C_{\text{tar}}}{N_{\text{tar}}}} \right\}, \tag{85}$$

where (84) follows from (79) and (82), and (85) holds by choosing $\alpha(t)$ as an indicator function as described in Section 4.3. Accordingly, we can convert this into a sample complexity bound. We use

$[z]^+$ as the shorthand for $\max\{0, z\}$. Moreover, suppose $\kappa_{\text{tar}}^{(t)} \leq \kappa_{\text{tar,max}}$, for all $t$. Note that $\kappa_{\text{tar,max}}$ can be configured by choosing the function class $\mathcal{Q}$. Then, given any $\epsilon > 0$, for any $\beta \in (0, 1)$, we have that for any $T \geq \left( \frac{[\log |\mathcal{A}_{\text{tar}}| + 1]}{(1-\gamma)\beta} \right)^2 \frac{1}{\epsilon^2} =: T(\epsilon)$ and $N_{\text{tar}} \geq \min \left\{ \frac{C_1^2 C_{\text{cd}}}{(1-\beta)^2 \epsilon^2}, \frac{C_{\text{tar}}}{\left[ \frac{(1-\beta)^2 \epsilon^2}{C_1^2} - 3\kappa_{\text{tar,max}} \right]^+} \right\}$,

the average sub-optimality gap is no more than $\epsilon$. Hence, by the fact that the final target-domain policy $\pi_{\text{tar}}^{(T)} \sim \text{Uniform}(\{\pi^{(1)}, \cdots, \pi^{(T)}\})$, we have that $\mathbb{E}_{s \sim \mu_{\text{tar}}} \left[ V^{\pi^*}(s) - V^{\pi_{\text{tar}}^{(T)}}(s) \right] \leq \epsilon$, for any

$T \geq \left( \frac{[\log |\mathcal{A}_{\text{tar}}| + 1]}{(1-\gamma)\beta} \right)^2 \frac{1}{\epsilon^2}$ and $N_{\text{tar}} \geq \min \left\{ \frac{C_1^2 C_{\text{cd}}}{(1-\beta)^2 \epsilon^2}, \frac{C_{\text{tar}}}{\left[ \frac{(1-\beta)^2 \epsilon^2}{C_1^2} - 3\kappa_{\text{tar,max}} \right]^+} \right\}$. This implies a total

number of target-domain samples

$$\mathcal{O}\left( \left( \frac{[\log |\mathcal{A}_{\text{tar}}| + 1]}{(1-\gamma)} \right)^2 \frac{1}{\epsilon^2} \cdot \min \left\{ \frac{C_1^2 C_{\text{cd}}}{\epsilon^2}, \frac{C_{\text{tar}}}{\left[ \frac{\epsilon^2}{C_1^2} - 3\kappa_{\text{tar,max}} \right]^+} \right\} \right) \tag{86}$$

is needed to achieve an $\epsilon$-optimal target-domain policy under QAvatar. Moreover, recall that one can recover the vanilla Q-NPG by setting $\alpha(t) = 0$ for all $t$. Hence, by setting $\alpha(t) = 0$ in (83)-(85), we can also obtain that a total number of target-domain samples

$$\mathcal{O}\left( \left( \frac{[\log |\mathcal{A}_{\text{tar}}| + 1]}{(1-\gamma)} \right)^2 \frac{1}{\epsilon^2} \cdot \frac{C_{\text{tar}}}{\left[ \frac{\epsilon^2}{C_1^2} - 3\kappa_{\text{tar,max}} \right]^+} \right) \tag{87}$$

is needed to achieve an $\epsilon$-optimal target-domain policy under Q-NPG. $\qquad \square$

## C   A DETAILED DESCRIPTION OF RELATED WORK

**CDRL across domains with distinct state and action spaces.** The existing approaches can divided into the following main categories:

- (i) *Manually designed latent mapping*: In (Ammar & Taylor, 2012) and (Ammar et al., 2012), the trajectories are mapped manually and by sparse coding from the source domain and the target domain to a common latent space, respectively. The distance between latent states can then be calculated to find the correspondence of the states from the different domains. In (Gupta et al., 2017), the correspondence of the states is found by dynamic time warping and the mapping function which can map the states from two domains to the latent space is found by the correspondence.

- (ii) *Learned inter-domain mapping*: In the literature (Taylor et al., 2008; Zhang et al., 2021; You et al., 2022; Gui et al., 2023; Zhu et al., 2024), the inter-domain mapping is mainly learned by enforcing dynamics alignment (or termed dynamics cycle consistency in (Zhang et al., 2021)), i.e., aligning the one-step transitions of the two domains. Additional properties have also been incorporated as auxiliary loss functions in learning the inter-domain mapping in the prior works, including domain cycle consistency (Zhang et al., 2021; You et al., 2022), effect cycle consistency (Zhu et al., 2024), maximizing mutual information between states and embeddings (You et al., 2022), and alignment of target-domain rewards with the embeddings (You et al., 2022). Moreover, as the state and action spaces are typically bounded sets and these methods directly map the data samples between the two domains, adversarial learning has been used to restrict the output range of the mapping functions (Zhang et al., 2021; Gui et al., 2023). On the other hand, in (Ammar et al., 2015), the state mapping function is found by Unsupervised Manifold Alignment (Wang & Mahadevan, 2009).

Despite the above progress, the existing approaches all presume that the domains are sufficiently similar and do not have any performance guarantees (and hence can suffer from negative transfer in bad-case scenarios). By contrast, this paper proposes a robust CDRL method that can achieve transfer regardless of source-domain model quality or domain similarity with guarantees.

**CDRL across domains with identical state and action spaces.** In CDRL, a variety of methods have been proposed for the case where source and target domains share the same state and action spaces but are subject to dynamics mismatch.

- (i) *Using the data samples from both source and target domains for policy learning*: One popular approach is to use the data from both domains for model updates (Eysenbach et al., 2021; Liu et al., 2022; Xu et al., 2023). For example, for compensating the discrepancy between domains in transition dynamics, (Eysenbach et al., 2021) proposes to modify the reward function, which is learned by an auxiliary domain classifier that distinguishes between the source-domain and target-domain transitions. (Liu et al., 2022) handles the dynamics shift problem in offline RL by augmenting rewards in the source-domain dataset. (Xu et al., 2023) proposes to address dynamics mismatch by a value-guided data filtering scheme, which ensures selective sharing of the source-domain transitions based on the proximity of paired value targets.

- (ii) *Explicit domain similarity*: (Sreenivasan et al., 2023) proposes to selectively apply direct transfer of the source-domain policy to the target domain based on a learnable similarity metric, which is essentially the TD error of target domain trajectories with source Q function. Moreover, based on the policy invariant explicit shaping (Behboudian et al., 2022), (Sreenivasan et al., 2023) further uses the potential function as a bias term for selecting actions.

- (iii) *Using both Q-functions for the Q-learning updates*: Target Transfer Q-Learning (Wang et al., 2020) calculates the TD error by the source and target domains Q functions in order to select the TD target from the two Q functions.

- (iv) *Domain randomization*: To tackle sim-to-real transfer with dynamics mismatch, domain randomization (Rajeswaran et al., 2016; Peng et al., 2018; Chebotar et al., 2019; Du et al., 2021) and (Du et al., 2021) collects data from multiple similar source domains with different configurations to learn a high-quality policy that can work robustly in a possibly unseen but similar target domain.

# D  ADDITIONAL EXPERIMENTAL RESULTS

## D.1  A TOY EXAMPLES FOR MOTIVATING THE BENEFIT OF CROSS-DOMAIN BELLMAN LOSS

We consider the 3-by-3 grid navigation problem, as shown in Figure 8. In both domains, there are only two actions: 'going top' and 'going right.' The state of the source domain is described in decimal coordinates, while the state of the target domain is described in binary coordinates. The white squares represent obstacles that cannot be traversed. There are three special states: (i) Start state: The episode always begins at this state. (ii) End state: The episode will only end at this state, and the agent will receive an ending reward of +1. (iii) Treasure state: When the agent first navigates to this state, it will receive +0.5 rewards. In other states or at other times navigating the treasure state, the agent will not

(a) Source Domain          (b) Target Domain

Figure 8: Source and target domains of the grid navigation example.

receive any reward. In the source domain, the start state, end state, and treasure state are set to $(0,0)$, $(0,2)$, and $(2,2)$, respectively. In the target domain, the start state, end state, and treasure state are set to $(0,0,0,0)$, $(0,0,1,1)$, and $(1,1,1,1)$, respectively. We assume that the source Q-function $Q_{\text{src}}$ is optimal in the source domain and the environment discount factor $\gamma$ is set to 0.99. It is easy to verify that the optimal trajectory of the source domain is $(0,0) \rightarrow (0,1) \rightarrow (0,2) \rightarrow (1,2) \rightarrow (2,2)$ and the optimal trajectory of the target domain is $(0,0,0,0) \rightarrow (0,0,0,1) \rightarrow (0,0,1,1) \rightarrow (0,1,1,1) \rightarrow (1,1,1,1)$. Consider two trajectories in the source domain: Traj-A, which is the optimal trajectory, and Traj-B, defined as $(0,0) \rightarrow (0,1) \rightarrow (1,1) \rightarrow (1,2) \rightarrow (2,2)$. When we map the optimal trajectory of the target domain to Traj-A and the optimal trajectory of the target domain to Traj-B, both mappings result in 0 cycle consistency loss. This suggests that the cycle consistency cannot determine which mapping is superior. This phenomenon results from the unsupervised nature of dynamics cycle consistency. In contrast, when we mapping the optimal trajectory of the target domain to Traj-A yields a cross-domain Bellman-like loss of 0, while mapping the optimal trajectory of

the target domain to Traj-B results in a cross-domain Bellman-like loss of 1. Thus, we can achieve optimal mapping results based on the cross-domain Bellman error, while the cycle consistency loss provides sub-optimal mapping results.

## D.2 FINAL REWARDS

In this section, we evaluate the asymptotic performance of all baselines and our algorithm. In the experiments, all target-domain models are trained for 500k steps in MuJoCo and 100k steps in Robosuite. The results are summarized in Table 2.

Table 2: Final rewards of $Q$Avatarand all baselines in the experiments.

| Algorithm | HalfCheetah | Ant | Door Opening | Table Wiping | Navigation |
|---|---|---|---|---|---|
| $Q$Avatar | **11586.0 ± 1224.4** | **2858.8 ± 848.0** | **216.6 ± 131.3** | **76.6 ± 13.5** | **38.5 ± 13.2** |
| SAC | 10986.0 ± 1821.8 | 1620.0 ± 527.2 | 94.8 ± 23.9 | 47.6 ± 11.0 | 19.7 ± 13.6 |
| FT | 10756.8 ± 1070.8 | 1644.3 ± 748.2 | 129.9 ± 34.6 | 42.1 ± 15.4 | 12.5 ± 9.0 |
| PAR | 8097.4 ± 3962.0 | 737.6 ± 45.3 | 33.7 ± 18.6 | 17.9 ± 11.8 | 0.0 ± 0.0 |
| CAT-SAC | 8756.5 ± 1264.3 | 1628.9 ± 200.6 | 63.2 ± 33.3 | 23.7 ± 10.7 | 2.7 ± 2.4 |
| CAT | 46.1 ± 149.9 | 17.1 ± 27.3 | 34.7 ± 8.4 | 55.5 ± 29.7 | -0.1 ± 0.2 |
| CMD | -253.1 ± 344.1 | 777.5 ± 144.1 | 7.8 ± 6.4 | 0.8 ± 0.4 | -0.0 ± 0.0 |

## D.3 ABLATION STUDY: EXPERIMENTAL RESULTS

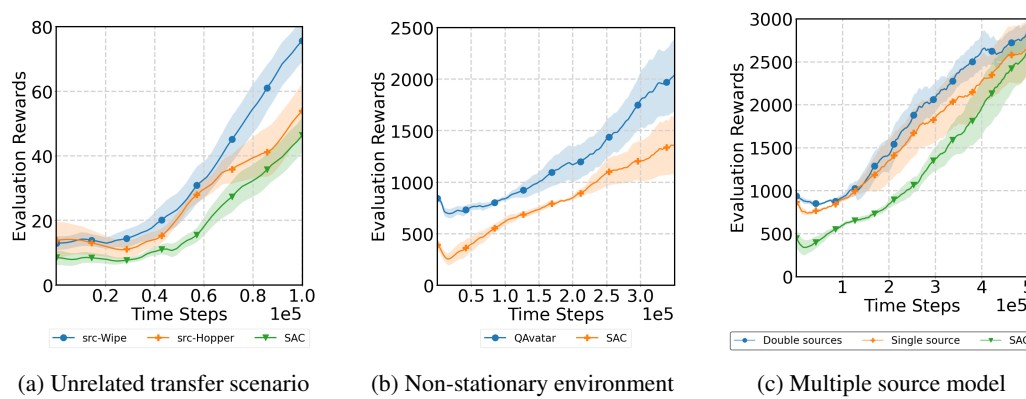

(a) Unrelated transfer scenario   (b) Non-stationary environment   (c) Multiple source model

Figure 9: Training curves of ablation experiments. (a) Unrelated transfer scenario where the target domain is Table-Wiping with UR5e. (b) Training curves in non-stationary Ant-v3. (c) Transfer from two source domains to a target domain.

## D.4 ADDITIONAL EXPERIMENT AND EXPLANATION DURING REBUTTAL

**$Q$Avatar on image-based experiment.**

We additionally evaluate $Q$Avatar on image-based continuous control tasks from the DeepMind Control Suite (DMC) (Tassa et al., 2018). In DMC, each observation consists of a stack of three 84×84 RGB frames, and an action repeat of 4 is applied. The protocol details are described as follows:

**SAC.** For SAC, both the actor and critic use 3 hidden layers with 1024 units each. The image encoder follows the IMPALA (Espeholt et al., 2018) architecture to extract low-dimensional visual features. All remaining hyperparameters are the same as those used in Stable-Baselines3.

**Flow model training.** Training a flow model directly on the raw high-dimensional image observations is challenging. Therefore, we first pass each image stack through the source encoder to obtain its feature representation, and train the flow model to match the distribution of these extracted features rather than the raw images. Notably, this modification does not alter the $Q$Avatar framework, since the source model remains fixed during target-domain transfer.

**Cross-domain transfer** To leverage the source critic for transfer, each target image observation is first passed through the target encoder to obtain its feature representation, which is then used as the input to the state decoder; the rest of the procedure follows the standard $Q$Avatar framework.

**Comparison to Effective Cycle Consistency (Zhu et al., 2024).**

Effective cycle consistency (ECC) (Zhu et al., 2024) operates under the same **unsupervised** cross-domain assumption as DCC and CMD, where the agent has no access to target-domain rewards. This makes the problem fundamentally more challenging than the supervised CDRL setting (i.e., with target-domain reward signal) considered in our work. Building on the DCC objective, ECC further introduces effect cycle-consistency to learn the mapping functions. We evaluate on Cheetah, Ant, Door Opening, and Table Wiping in Section 5.1. Figure 10 demostrate that although ECC produces more stable alignment than CMD, its overall performance remains significantly below SAC and other supervised CDRL baselines, which is consistent with the inherent limitations of the unsupervised CDRL methods.

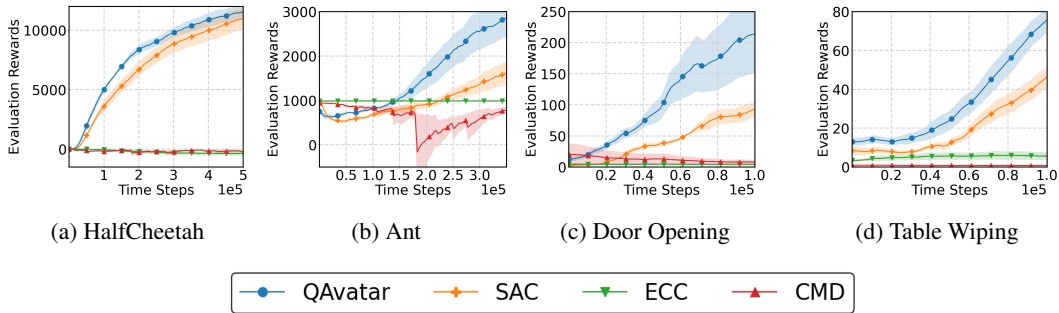

(a) HalfCheetah     (b) Ant     (c) Door Opening     (d) Table Wiping

Figure 10: Performance evaluation of ECC.

# E    IMPLEMENTATION DETAILS OF $Q$AVATAR

## E.1    PSEUDO CODE OF THE PRACTICAL IMPLEMENTATION OF $Q$AVATAR

In this section, we provide the pseudo code of the practical version of $Q$Avatarin Algorithm 3.

---

**Algorithm 3** Practical Implementation of $Q$Avatar

---

1: **Require:** Source-domain Q-network $Q_{\text{src}}$, update $\alpha$ frequency $N_\alpha$, batch size $N$.
2: Initialize the state mapping function $\phi$, the action mapping function $\psi$, the initial target-domain
   policy network $\pi^{(1)}$, entropy coefficient $\beta$, replay buffer $D$, and $\alpha = 0$.
3: **for** iteration $t = 1, \cdots, T$ **do**
4:     Interact with the environment and store the transition $(s_t, a_t, r_t, s_{t+1})$ in the replay buffer $D$.
5:     Sample two sets of $N$ transitions, denoted as $B_{\text{SAC}}$ and $B_{\text{Map}}$, from the replay buffer $D$.
6:     Update the target-domain $\{Q_{\text{tar},1}, Q_{\text{tar},2}\}$ by SAC's critic loss:

$$Q_{\text{tar},j}^{(t)} = \arg\min_{Q_{\text{tar}}} \hat{\mathbb{E}}_{(s,a,r,s')\in B_{\text{SAC}}} \left[ \left| r + \gamma \mathbb{E}_{a'\sim\pi^{(t)}(\cdot|s')} \left[ Q_{\text{tar}}(s',a') - \beta\log(\pi(a'|s')) \right] - Q_{\text{tar}}(s,a) \right|^2 \right]. \tag{88}$$

7:     Update the state mapping function $\phi$ and action mapping function $\psi$ by minimizing
8:     the following loss:

$$\phi^{(t)}, \psi^{(t)} = \arg\min_{\phi,\psi} \hat{\mathbb{E}}_{(s,a,r,s')\in B_{\text{Map}}} \left[ \left| r + \gamma \mathbb{E}_{a'\sim\pi^{(t)}(\cdot|s')} \left[ Q_{\text{src}}(\phi(s'), \psi(a')) \right] - Q_{\text{src}}(\phi(s), \psi(a)) \right|^2 \right]. \tag{89}$$

9:     **if** $t \bmod N_\alpha = 0$ **then**
10:        Define $\|\epsilon_{\text{td}}^{(t)}\|_D = \hat{\mathbb{E}}_{(s,a,r,s')\in D} \left[ \left| r + \gamma\mathbb{E}_{a'\sim\pi^{(t)}(\cdot|s')} \left[ \min_{j=1,2} Q_{\text{tar},j}^{(t)}(s',a') \right] - \min_{j=1,2} Q_{\text{tar},j}^{(t)}(s,a) \right| \right]$,
11:        $\|\epsilon_{\text{cd}}(Q_{\text{src}}, \phi^{(t)}, \psi^{(t)})\|_D = \hat{\mathbb{E}}_{(s,a,r,s')\in D} \left[ \left| r + \gamma\mathbb{E}_{a'\sim\pi^{(t)}(\cdot|s')} \left[ Q_{\text{src}}(\phi^{(t)}(s'), \psi^{(t)}(a')) \right] - Q_{\text{src}}(\phi^{(t)}(s), \psi^{(t)}(a)) \right| \right]$.
12:        Update the weight $\alpha = \|\epsilon_{\text{td}}^{(t)}\|_D / (\|\epsilon_{\text{cd}}(Q_{\text{src}}, \phi^{(t)}, \psi^{(t)})\|_D + \|\epsilon_{\text{td}}^{(t)}\|_D)$.
13:    **end if**
14:    Update the target-domain policy $\pi$:

$$\pi^{(t+1)} = \arg\min_{\pi} \hat{\mathbb{E}}_{\substack{(s,a,r,s')\in B_{\text{SAC}} \\ a'\sim\pi^{(t)}(\cdot|s)}} \left[ \beta\log\pi(a'|s) - f^{(t)}(s,a') \right], \tag{90}$$

$$f^{(t)}(s,a') = (1-\alpha)\min_{j=1,2} Q_{\text{tar},j}^{(t)}(s,a') + \alpha Q_{\text{src}}(\phi^{(t)}(s), \psi^{(t)}(a')). \tag{91}$$

15: **end for**

---

## E.2    SOURCE-DOMAIN MODELS AND THEIR PERFORMANCE

For the locomotion tasks including HalfCheetah and Ant, we train each source model for 1M steps. The average performance of the 5 source-domain models (under 5 distinct random seeds) in HalfCheetah and Ant are $7355 \pm 2892$ and $3689 \pm 1013$, respectively. For the Robosuite tasks including Door Opening and Table Wiping, we train each source-domain model for 500K steps. The average performance of 5 random seed is $383 \pm 139$ and $94 \pm 16$, respectively. For the navigation environment, we train the model for 500K steps, and the average performance is $39.85$.

## E.3    INTER-DOMAIN MAPPING NETWORK AUGMENTED WITH A NORMALIZING FLOW MODEL

As discussed in Section 4, a flow-based generative model is employed to transform the outputs of the mapping functions into their corresponding feasible regions. Therefore, there are two architectural paradigms of the flow model can be considered. In the first paradigm, the state and action are concatenated and jointly treated as the codomain of the flow model. This joint formulation is adopted in Cheetah, Ant environment. In the second paradigm, the state and action are modeled separately,

with two independent flow models trained respectively for the state and the action. This decoupled formulation is applied in Hopper-v3, Table Wiping, and Door Opening tasks.

### E.4 QUALITY AND STABILITY OF LEARNED STATE/ACTION MAPPING FUNCTIONS $\phi$ AND $\psi$.

In $Q$Avatar, a key diagnostic for assessing alignment quality and stability is the cross-domain Bellman error. When this error approaches zero, it implies that $Q_{src}$ is $\delta$-Bellman-consistent with a sufficiently small $\delta$ under the learned mappings, indicating that $\phi$ and $\psi$ are well aligned. Figure 11 demostrate the curves of cross-domain Bellman error versus the TD error of the target critic across all main experiments in Section 5. The results consistently show that the cross-domain Bellman error remains low relative to the TD error when the learned mappings align well with the target domain.

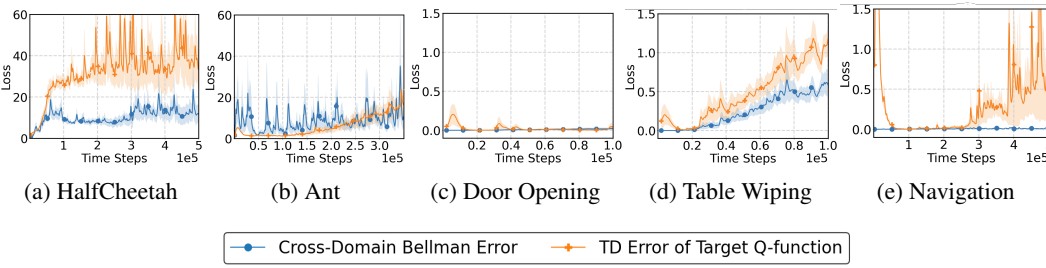

| (a) HalfCheetah | (b) Ant | (c) Door Opening | (d) Table Wiping | (e) Navigation |

Figure 11: Cross-domain Bellman error and TD error of the target Q-function during training across all main experiments.

## F CONFIGURATION DETAILS OF THE EXPERIMENTS

### F.1 STATE AND ACTION DIMENSIONS OF BENCHMARK ENVIRONMENTS

We summarize the state and action dimensions of each pair of source-domain and target-domain benchmark tasks in the following Table 3.

Table 3: Dimensions of the source and target domains ("Src" and "Tar" represent the source domain and the target domain.)

| Environment | State | | Action | |
|---|---|---|---|---|
| | Src | Tar | Src | Tar |
| HalfCheetah | 17 | 23 | 6 | 9 |
| Ant | 111 | 133 | 8 | 10 |
| Door Opening | 46 | 51 | 8 | 7 |
| Table Wiping | 37 | 34 | 7 | 6 |
| Goal Navigation | 40 | 72 | 2 | 12 |

### F.2 MUJOCO AND ROBOSUITE ENVIRONMENTS

As mentioned in Section 5, We evaluate $Q$Avatarin both MuJoCo and Robosuite environments. In the MuJoCo environments, the source domains of our experiments are the original MuJoCo environments such as HalfCheetah-v3 and Ant-v3. The target domains are the modified MuJoCo environments such as HalfCheetah with three legs and Ant with five legs. In Robosuite environments, We evaluate $Q$Avataron two tasks, including door opening and table wiping. For each task, we consider cross-domain transfer from controlling a Panda robot arm to controlling a UR5e robot arm. These four tasks are illustrated in Figure 12 and 13.

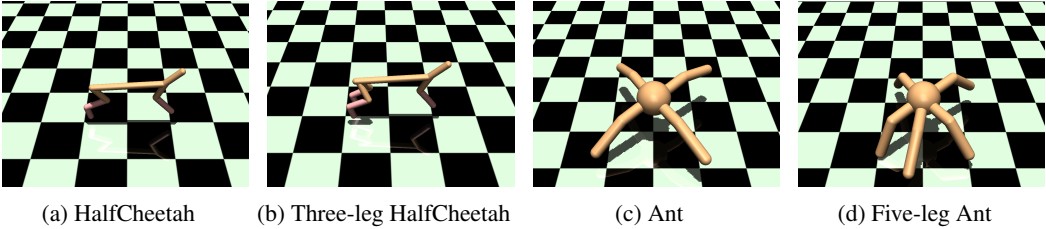

| (a) HalfCheetah | (b) Three-leg HalfCheetah | (c) Ant | (d) Five-leg Ant |

Figure 12: The environments of the source domains and the target domains. (a),(c): Source domains – Original MuJoCo environments. (b),(d): Target domains – Modified MuJoCo environments.

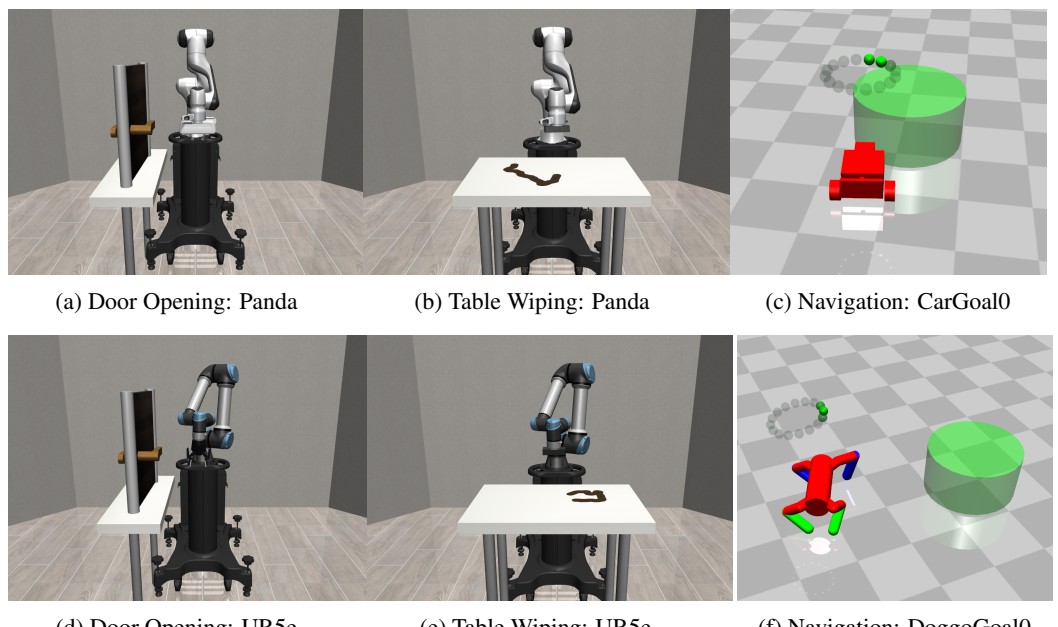

| (a) Door Opening: Panda | (b) Table Wiping: Panda | (c) Navigation: CarGoal0 |

| (d) Door Opening: UR5e | (e) Table Wiping: UR5e | (f) Navigation: DoggoGoal0 |

Figure 13: The environments of the source and target domains. (a)–(c): Source domains. (d)–(f): Target domains.

### F.3 THE IMPLEMENTATION DETAILS OF BASELINES

**SAC.** The implementation of SAC used in our experiments is released by Stable-Baselines3 (Raffin et al., 2021). The settings of all hyperparameters except for the discouted factor $\gamma$ follows the default settings of SAC in the documentation of Stable-Baselines3. The discouted factor is set 0.99 in all other MuJoCo environments. In Robosuite environments, we set the discouted factor to 0.9.

**CMD.** Since there is no publicly available implementation of CMD, we leverage and adapt the codebase of DCC (Zhang et al., 2021) (https://github.com/sjtuzq/Cycle_Dynamics) and reproduce CMD by following the pseudo code of CMD in its original paper (Gui et al., 2023). We follow the setting of the hyperparameters which is revealed in its original paper. Additionally, we change CMD from collecting the fixed amount of data to collecting data continuously for a fair comparison. As for the source model, we use the same model used in our algorithm. Moreover, we observe that the original setting could suffer because the collected trajectories mostly have low returns due to a random behavior policy. Therefore, we consider a stronger version of CMD with target-domain data collected under the target-domain policy, which is induced by the source-domain pre-trained policy and the current inter-domain mappings.

**FT.** FT can be seen as a standard SAC algorithm with source feature initialization. Specifically, we modify the input and output layers of the source policy to match the target domain's state and action dimensions, using random initialization, while keeping the middle layers with the same weights as the

source model. Similarly, for the source Q function, we adjust the input layer to fit the target domain's state and action dimensions with random initialization, while the remaining layers retain the source model's weights. After initialization, the agent is fine-tuned using the standard SAC algorithm.

**CAT.** We use the authors' implementation ([https://github.com/TJU-DRL-LAB/transfer-and-multi-task-reinforcement-learning/tree/main/Single-agent%20Transfer%20RL/Cross-domain%20Transfer/CAT](https://github.com/TJU-DRL-LAB/transfer-and-multi-task-reinforcement-learning/tree/main/Single-agent%20Transfer%20RL/Cross-domain%20Transfer/CAT)) and use PPO as the target-domain base algorithm following the original paper. For a fair comparison, we use the same source model used in $Q$Avatar.

**CAT-SAC.** As CAT can be integrated with any off-the-shelf RL method, we adapt the original PPO-based CAT to CAT-SAC by using the SAC implementation in Spinning Up (Achiam, 2018) as the backbone of CAT-SAC. All the SAC-related hyperparameters are the same as those used by SAC and the CAT-related parameters are configured as in the original implementation. For a fair comparison, we use the same source model used by $Q$Avatar.

**PAR.** We use the authors' implementation ([https://github.com/dmksjfl/PAR.git](https://github.com/dmksjfl/PAR.git)) and consider the offline to online version of PAR, which is more compatible with the CDRL setting in our paper. For the source-domain data required by PAR, we use the samples in the buffer collected during the training of the source-domain policies (shared by $Q$Avataraand other baselines). As a result, to adapt PAR to the more general CDRL setting in our paper, similar to the data pre-processing methods used in handling sequences (Zahavy et al., 2018; Dwarampudi & Reddy, 2019; Morad et al., 2024; Wu & Hu, 2018), we use padding and truncation to handle the differences in state and action dimensions. More specifically,

- **Padding**: If the target domain has $n$ more dimensions than the source, we append $n$ zeros to the end of each source sample.

- **Truncation**: If the target domain has $n$ fewer dimensions than the source, we discard the last $n$ from each source sample.

Note that this design is reasonable, as neither the baselines nor $Q$Avatarhave any knowledge about the physical meaning of each entry in the state or action representations. For the hyperparameters, to ensure a fair comparison with $Q$Avataras well as the baselines CAT-SAC and SAC, we set the ratio between environment interaction and agent training to 1 (*i.e.,* `config['tar_env_interact_freq']` in their original code). Other parameters (*e.g.,* beta, weight, etc.) and network architecture follow the recommendations provided in the original PAR paper. In addition, we observe that in some environments, temperature tuning can improve performance. Therefore, we apply temperature tuning during the training process (as adopted by PAR's original code), and select the better one between using and not using temperature tuning as the final result.

### F.4 Hyperparamter tuning of the baselines

In this section, we provide the value of hyperparamter tuning detail of in the Section 5. For fairness, all SAC-based methods (QAvatar, SAC, FT, CAT-SAC, and PAR) use exactly the same SAC-related hyperparameters. This ensures that any performance differences arise solely from whether transfer is applied and how it is implemented. For all locomotion tasks, we follow the recommended SAC hyperparameters from Stable-Baselines3 (Raffin et al., 2021). Thus, there are no hyperparameter need to tune in SAC and FT. For the other baselines,

**CAT and CAT-SAC.** We conducted tuning for the schedule of $p(t)$, which is the weight of the linear combination of hidden layer parameters: $p(t) = 0$ means that only the source-domain parameters are used; $p(t) = 1$ means that only the target-domain parameters are used). Specifically, You et al.

(2022) sets $p(t)$ to be piecewise linear as follows: Let $T$ be the total training steps.

$$p(t) = \begin{cases} 0, & t \in [0,\, c_1 T], \\ \dfrac{t - c_1 T}{(c_2 - c_1)T}, & t \in [c_1 T,\, c_2 T], \\ 1, & t \in [c_2 T,\, T]. \end{cases}$$

The official CAT chooses $c_1 = 0.45, c_2 = 0.9$. For each environment, we choose the best among the following candidate choices: $(c_1, c_2) \in \{(0.15, 0.4), (0.4, 0.7), (0.45, 0.9)\}$.

**PAR.** We performed a grid search over the penalty coefficient $\beta : \{0.1, 0.5, 1.0, 2.0\}$, which are the values suggested by the ablation study in the original paper. We also searched over the policy objective normalization coefficient $\nu : \{2.5, 5.0\}$, as recommended by the official code and Appendix E.1 of the original paper. In addition, we evaluated both configurations that enable temperature tuning during training, which is used in their official implementation, and configurations that exclude temperature tuning, which is the default setting. We selected the best-performing configuration in the Cheetah environment and applied this configuration to all remaining environments.

**CMD.** We conduct a grid search over the loss weights $(\rho_0, \rho_1, \rho_2)$ in

$$L_{2nd} = \rho_0 L_{gan}(D_{\text{source}}, G_1) + \rho_1 L_{gan}(D_{\text{target}}, G_2) + \rho_2 L_3(G_1, G_2)$$

The official CMD setting is $(\rho_0, \rho_1, \rho_2) = (1, 1, 3)$, we fix $\rho_1 = 1$ and conduct the grid search on $\rho_2 : \{0.3, 1.0, 3.0\}, \rho_2 = \{1.0, 3.0, 10.0\}$. Across all combinations, the performance is similar and consistently much lower than that of SAC. Based on this observation, we adopt the original recommended CMD setting for all remaining environments.

### F.5 DETAILED CONFIGURATION OF $Q$AVATAR

The base algorithm, Soft Actor-Critic (SAC), is implemented using Stable-Baselines3 (Raffin et al., 2021). All experiments were conducted on a computing server equipped with dual Intel Xeon Gold 6154 CPUs (36 cores in total) and an NVIDIA Tesla V100-SXM2 GPU with 32 GB of GPU memory.

The hyperparameters of $Q$Avatar are summarized in the following table. Unless otherwise specified, the hyperparameter settings, including the learning rates of the actor and critic networks, batch size, replay buffer size, and discount factor, follow the default configurations of SAC.

Table 4: A list of hyperparameters of $Q$Avatar.

| Parameter | Value |
|---|---|
| critic/actor learning rate | 0.0003 |
| mapping function learning rate | 0.0001 |
| batch size | 256 |
| replay buffer size | $10^6$ |
| optimizer for Actor/critic | Adam |
| optimizer for mapping function | AdamW |
| hidden layer size | 256 |
| update $\alpha$ frequency $N_\alpha$ | 1000 |

# G ANALYSIS OF THE OFF-POLICY VARIANT OF PROPOSITION 3

In Proposition 3, we provide an upper bound on the average sub-optimality of the on-policy version of $Q$Avatar. In this section, we derive the corresponding upper bound for the off-policy variant of $Q$Avatar. The primary difference between the on-policy and off-policy settings lies in the data collection policy. The on-policy approach collects data using the learned policy $\pi^{(t)}$, while the off-policy variant collects data using a behavior policy $\pi_\beta^{(t)}$. Based on notation use in the main paper, we provide the average sub-optimality of the off-policy version of QAvatar as following:

**Corollary 2.** *Under the QAvatarin Algorithm 2, but with the data collection policy $\pi^{(t)}$ replaced by $\pi_\beta^{(t)}$, and under Assumption 1, the average sub-optimality over $T$ iterations can be upper bounded as follows:*

$$\frac{1}{T}\sum_{t=1}^{T}\mathbb{E}_{s\sim\mu_{tar}}\left[V^{\pi^*}(s)-V^{\pi^{(t)}}(s)\right]$$

$$\leq \underbrace{\frac{[\log|\mathcal{A}_{tar}|+1]}{\sqrt{T}(1-\gamma)}}_{(a)} + \underbrace{\frac{C_0}{T}\sum_{t=1}^{T}\mathbb{E}_{(s,a)\sim d_\beta^{\pi^{(t)}}}\left[\left|f^{(t)}(s,a)-Q^{\pi^{(t)}}(s,a)\right|\right]}_{(b)} \tag{92}$$

$$\leq \underbrace{\frac{[\log|\mathcal{A}_{tar}|+1]}{\sqrt{T}(1-\gamma)}}_{(a)} + \underbrace{\frac{C_1}{T}\sum_{t=1}^{T}\left(\alpha(t)\|\epsilon_{cd}(Q_{src},\phi^{(t)},\psi^{(t)})\|_{d_\beta^{\pi^{(t)}}} + (1-\alpha(t))\|\epsilon_{td}^{(t)}\|_{d_\beta^{\pi^{(t)}}}\right)}_{(c)}, \tag{93}$$

*where $C_0 := 2C_{\pi^*,\beta}/(1-\gamma)$ and $C_1 := 2C_{\pi^*,\beta}/((1-\gamma)^3\mu_{tar,\,min})$.*

*Proof.* The proof follows exactly the same steps as those used for Proposition 3. The only difference lies in the proof of Lemma 5, specifically in Equation (34). There, instead of assuming the learned policy $\pi^{(t)}$ together with the corresponding $\left\|\frac{d^{\pi^*}}{d^{\pi^{(t)}}}\right\|_\infty \leq C$, we replace it with the behavior policy $\pi_\beta^{(t)}$ together with the corresponding $\left\|\frac{d^{\pi^*}}{d_\beta^{\pi^{(t)}}}\right\|_\infty \leq C_\beta$. Notably, both $C_\beta$ and $C$ are bounded constants given the condition of exploratory initial distribution in Assumption 1. For the subsequent derivations in Lemma 5 as well as those in Lemma 6, we may directly replace $d^{\pi^{(t)}}$ with $d_\beta^{\pi^{(t)}}$. This substitution preserves all arguments, because the relevant manipulations are carried out entirely inside expectations with respect to a distribution, and the structure of the inequalities does not depend on which particular distribution the expectation is taken over.

Lemma 4 does not require any modification, because it is designed to control the learning process of the NPG-style policy update. This part of the analysis does not depend on whether the data are collected by $\pi^{(t)}$ or by $\pi_\beta^{(t)}$. With the above substitutions, we may directly use the modified versions of Lemma 5, Lemma 6, and Lemma 4. Following the same sequence of steps as in Equations (58) to (64), we can then complete the proof of this corollary. $\square$

In our practical SAC-based implementation of $Q$Avatar, we set the behavior policy $\pi_\beta^{(t)}$ to match the buffer distribution, which can be viewed as a mixture of all past learned policies. Consequently, the value of $\alpha(t)$ can naturally be evaluated over the buffer distribution.

