# OpenReview forum: "Cross-Domain Policy Optimization via Bellman Consistency and Hybrid Critics"
_ICLR.cc/2026/Conference — ICLR 2026 Poster_

### Official Review · Reviewer_bUDR · 2025-10-30

**Soundness:** 2
**Presentation:** 3
**Contribution:** 2
**Rating:** 4
**Confidence:** 4

**Summary:**

The paper studies cross-domain transfer RL: the source and the target MDPs could differ in both state and action spaces. The transfer could be infeasible in that good inter-domain mappings may not exist. It introduces QAvatar, an algorithm that uses a convex combination of the Q-values from the source and the Q-values from the learned critic in the target to update the target policy. They show the efficacy of their method through locomotion, manipulation, and goal navigation tasks.

**Strengths:**

- The paper is mostly well-written.
- Cross-domain transfer in RL is an interesting problem, and the authors tackle it with a novel approach.
- The algorithm, QAvatar, is principled and is motivated by theoretical insights.

**Weaknesses:**

- There are discrepancies between what the theory suggests and what the practical version of QAvatar does: while $\alpha_t$ suggested by the theory involves norm defined w.r.t. visitation distributions of the policy at time t (line 332), empirically, the authors use the entire replay buffer for it (lines 1315 - 1318).
- In the experiments, other CDRL baselines considered by the authors perform worse than learning from scratch (SAC) and direct finetuning (FT).  Did the authors optimize the hyperparameters for these baselines?
- While QAvatar is theoretically motivated, the method as such does not come with any guarantees. In this setting, one would be interested in sample complexity guarantees: the proposed algorithm is provably more sample efficient than learning from scratch. The authors do not provide guarantees of this form. But I understand that this might be beyond the scope of the paper.

**Questions:**

See weaknesses.

---

> ### Author Response · Authors · 2025-11-26
> **Response to Reviewer bUDR (1/3)**
>
> We greatly appreciate the reviewer for the thoughtful and constructive feedback. We address each concern point-by-point below and have revised the paper accordingly.
>
> **[Q1]: Regarding the on-policy and off-policy variants of QAvatar**
> > There are discrepancies between what the theory suggests and what the practical version of QAvatar does: while $\alpha_{t}$ suggested by the theory involves norm defined w.r.t. visitation distributions of the policy at time t (line 332), empirically, the authors use the entire replay buffer for it (lines 1315 - 1318).
>
> Thank you for pointing out this important issue. We’d like to clarify that while our theoretical analysis is presented in the on-policy tabular NPG setting, **the same analysis can be directly extended to the off-policy RL setting used in our implementation, with only minimal changes to the proof**.
> Specifically: The main change lies in the data distribution:
> Instead of the $d^{\pi^{(t)}}$ induced by the current policy $\pi^{(t)}$, the data distribution is $d^{\pi_{\beta}^{(t)}}$ determined by a $\textit{behavior policy}$ $\pi^{(t)}\_{\beta}$​. In practice, $d^{\pi_{\beta}^{(t)}}$ can be the data distribution induced by some exploratory policy or corresponds to a probabilistic mixture of past distributions $\lbrace d^{\pi^{(t)}}\rbrace_{t}$ induced by the replay-buffer sampling.
> Under this off-policy variant, the proof only requires modifying the coverage assumption in Lemma 5. Specifically, instead of assuming $\Big\lvert \frac{d^{\pi^\*}}{d^{\pi^{(t)}}}\Big\rvert\_{\infty}\le C$, we assume that $\Bigg\lvert\frac{d^{\pi^\*}}{d^{\pi^{(t)}\_{\beta}}} \Bigg\rvert\_{\infty}\le C\_{\beta}$, and the remaining steps follow by substituting $d^{\pi^{(t)}}$ with $d^{\pi^{(t)}\_{\beta}}$​. As a result, in the off-policy case, the hybrid-critic weight $\alpha_t$​​ is naturally defined with respect to $d^{\pi^{(t)}\_{\beta}}$, which is consistent with our implementation where $\alpha_t$​ is computed from replay-buffer samples.
>
> This discussion is also provided in Corollary 2 in Appendix G of the updated manuscript to strengthen the connection between the theoretical analysis and the practical implementation for the readers.

---

> ### Author Response · Authors · 2025-11-26
> **Response to Reviewer bUDR (2/3)**
>
> **[Q2]: Hyperparameters tuning for the baselines**
> > In the experiments, other CDRL baselines considered by the authors perform worse than learning from scratch (SAC) and direct finetuning (FT). Did the authors optimize the hyperparameters for these baselines?
>
> We appreciate the opportunity to clarify this point. Yes, we did optimize the hyperparameters for these baselines as much as we could. Specifically: To ensure a fair comparison, all SAC-based methods (QAvatar, SAC, FT, CAT-SAC, and PAR) use exactly the same SAC-related hyperparameters. This ensures that any performance differences arise solely from whether transfer is applied and how it is implemented. For all locomotion tasks, we follow the recommended SAC hyperparameters from Stable-Baselines3 (see Appendix F.3).
>
> Regarding hyperparameter tuning for each individual CDRL baseline:
>
> **CAT (or CAT-SAC)**: We conducted tuning for the schedule of $p(t)$, which is the weight of the linear combination of hidden layer parameters: $p(t) = 0$ means that only the source-domain parameters are used; $p(t) = 1$ means that only the target-domain parameters are used). In the original paper, the authors sets $p(t)$ to be piecewise linear as follows:
>
> Let $T$ be the total training steps.
>   1. $p(t) = 0$, for $t \in [0, c_1 T]$
>   2. $p(t) $ grows linearly from 0 to 1 during $t \in [c_1 T, c_2 T] $
>   3. $ p(t) = 1 $, for $ t \in [c_2 T, T] $
>
> The official CAT chooses $ c_1 = 0.45, c_2 = 0.9 $. For each task, we choose the best among the following candidate choices:  $ (c_1, c_2) \in \lbrace (0.15, 0.4), (0.4, 0.7), (0.45, 0.9)\rbrace $.
>
> **PAR**: We performed a grid search over the penalty coefficient $\beta: \lbrace0.1, 0.5, 1.0, 2.0\rbrace$, which are the values suggested by the ablation study in the original paper. We also searched over the policy objective normalization coefficient $\nu:\lbrace 2.5, 5.0\rbrace$, as recommended by the official code and Appendix E.1 of the original paper. In addition, we evaluated both configurations that enable temperature tuning during training, which is used in their official implementation, and configurations that exclude temperature tuning, which is the default setting. We selected the best-performing configuration in the Cheetah environment and applied this configuration to all remaining environments.
>
> **CMD**: We conduct a grid search over the loss weights $(\rho_{0}, \rho_{1}, \rho_{2})$ in
> 	$$L_{2nd}=\rho_{0}L_{gan}(D_{\text{source}}, G_{1})+\rho_{1}L_{gan}(D_{\text{target}}, G_{2})+\rho_{2}L_{3}(G_{1}, G_{2})$$
> 	The official CMD setting is $(\rho_{0}, \rho_{1}, \rho_{2})=(1,1,3)$, we fix $\rho_{0}=1$ and conduct the grid search on $\rho_{1}:\lbrace0.3, 1.0, 3.0\rbrace, \rho_{2}=\lbrace1.0, 3.0, 10.0\rbrace$. **Across all combinations, the performance remains similar and consistently much lower than that of SAC**. Based on this observation, we adopt the original recommended CMD setting for all remaining environments.
>
> We have added the above detailed configurations in Appendix F.4 for completeness.
>
> **Explanation of poor performance of the baselines**
>
> **CAT (or CAT-SAC)**: CAT transfers knowledge by forming a weighted combination of corresponding layers between the source and target networks, which can be viewed as a parameter-based transfer method. This approach implicitly **assumes that the source and target domains share the same feature representations**. When the domains differ, this assumption often does not hold, which leads to degraded transfer quality. CAT also does not provide convergence guarantees, and therefore it offers no theoretical safeguard when the feature representations are mismatched.
>
> **CMD**: CMD is designed for an unsupervised transfer setting where the reward function in the target domain is not accessible. As a result, the learned policy cannot benefit from the target-domain reward signal, which directly leads to its weaker performance in our setting.
>
> **PAR**: PAR assumes that the source and target domains share the same state space, action space, and reward function. These assumptions do not hold in our general cross-domain scenario. As a result, the method is not directly applicable and consequently performs poorly in this setting.
>
> Thus, we hypothesize that **the performance differences observed in the baselines are largely attributable to the assumptions and problem settings these methods are originally designed for**. This helps explain why their results cannot fully carry over to our more general cross-domain setting.

---

> ### Author Response · Authors · 2025-11-26
> **Response to Reviewer bUDR (3/3)**
>
> **[Q3] Convergence guarantees and sample complexity**
> > While QAvatar is theoretically motivated, the paper does not provide explicit guarantees showing that it is more sample-efficient than learning from scratch. One would expect sample-complexity results in this setting, though this may fall outside the scope of the current work.
>
> Thank you for the thoughtful question. We clarify that the convergence guarantee in Proposition 3 can be directly converted into sample-complexity bounds, following the standard RL analyses [1-3]. The derivation proceeds as follows.
>
> **Step 1: Connect the errors bounds $\epsilon_{td}$ and $\epsilon_{cd}$ with the number of samples per iteration**
>
> To achieve this, we use a least squares generalization bound (LSGB) (e.g., [1] and [2]) for sequential function estimation, which learns a least-squares estimate $\hat{h}(x)$ (within a function class $\mathcal{H}={h(x)}$) of an underlying function $h^*(x)$ from stochastic samples. This allows us to rewrite the error bounds in terms of the number of samples using a PAC-style argument. For ease of exposition, we focus on the case of cross-domain realizability, where $Q_{src}$ is 0-Bellman consistent (cf. Definition 4).
>
> **Step 1-1: Bound $\epsilon_{td}$**
>
> Recall that at iteration $t$, QAvatar collects a dataset $\mathcal{D}^{(t)}\_{tar}=\lbrace(s,a,r,s')\rbrace$ and computes $Q^{(t)}\_{tar}$ as the minimizer of the TD loss over $\mathcal{Q}$, where $\mathcal{Q}$ is finite class of target-domain action-value functions.
>
> In QAvatar, the function $\hat{h}(x)$ in LSGB corresponds to the one-step TD error induced by $Q^{(t)}_{tar}$​, and $h^{*}(x)$ corresponds to the TD error induced by $Q^{\pi^{(t)}}$, which is zero under the Bellman expectation equation.
>
> We then verify that the three LSGB conditions (approximate realizability, zero-mean noise, boundedness) indeed hold. Let $\kappa_{tar}$ denote the approximation error. With probability at least $1-\delta$, we obtain
> $$
>     \lVert\epsilon_{td}^{(t)}\rVert_{d^{\pi^{(t)}}}^2\leq 3\kappa_{tar}+\frac{C_{tar}}{N_{tar}},
> $$
> for some problem- and $\mathcal{Q}$-dependent constant $C_{tar}$.
>
> **Step 1-2: Bound $\epsilon_{cd}$**
>
> A similar argument applies to the cross-domain Bellman error, where the TD error is replaced by the CD-Bellman error induced by $Q_{src}(\phi,\psi)$. Under cross-domain realizability, the optimal error is zero, so $h^{*}(x)=0$. The three LSGB conditions (approximate realizability, zero-mean noise, boundedness) also hold in this case. Hence, with probability at least $1-\delta$,
> $$
> \lVert\epsilon_{cd}(Q_{src},\phi^{(t)},\psi^{(t)})\rVert_{d^{\pi^{(t)}}}^2\leq\frac{C_{cd}}{N_{tar}},
> $$
> for some problem- and $\mathcal{F}$-dependent constant $C_{cd}$.
>
> **Step2: Connect Step1 and Proposition3**
>
> Substituting these bounds into Proposition 3 yields a sample-complexity expression depending only on $T$ (iterations) and $N_{tar}$ (per-iteration samples). Let
> $T(\epsilon)=\Big(\frac{\left[\log|\mathcal{A}\_{tar}|+1\right]}{(1-\gamma)}\Big)^2\frac{1}{\epsilon^2}$. To achieve an $\epsilon$-optimal policy with probability at least $1-\delta$, the total number of target samples is
> $$
> \mathcal{O}\Big(T(\epsilon)\cdot\min\Big\lbrace\frac{C_1^2C_{\text{cd}}}{\epsilon^2},\frac{C_{tar}}{\big[\frac{\epsilon^2}{C_1^2}-3 \kappa_{tar,\max}\big]^{+}} \Big\rbrace\Big).
> $$
> Recovering vanilla Q-NPG by setting $\alpha(t)=0$ in QAvatar gives
> $$
> \mathcal{O}\bigg(T(\epsilon)\cdot\frac{C_{tar}}{\big[\frac{\epsilon^2}{C_1^2}-3 \kappa_{tar,\max}\big]^+}\bigg)
> $$
> Thus, under cross-domain realizability, the derived sample-complexity bound shows that QAvatar is always comparable or better than learning from scratch. Moreover, when the $\kappa_{tar}$ is relatively large, $\frac{C_1^2C_{\text{cd}}}{\epsilon^2}<\frac{C_{tar}}{\big[\frac{\epsilon^2}{C_{1}^{2}}-3 \kappa_{tar,\max}\big]^{+}}$,
> so QAvatar achieves higher sample efficiency by leveraging the source Q-function. This benefit already appears in Proposition 3, where QAvatar attains a tighter upper bound on average sub-optimality than vanilla Q-NPG.
>
> We have added the above discussion in Corollary 1 with the detailed proof in Appendix B.4.
>
> Notably, in this discussion, we focus on the cross-domain realizability scenario to illustrate that QAvatar can achieve higher sample efficiency in the most favorable transfer scenario. It also appears feasible to extend this analysis to the more general case of $\delta$-Bellman consistency for any $\delta$, although this requires extending the lemma to a more general form. We appreciate this insightful point; it indeed opens a promising avenue for future work, though it lies beyond the scope of the current paper.
>
> [1] Agarwal et al., “Reinforcement learning: Theory and algorithm,” 2019.
>
> [2] Song et al., “Hybrid RL: Using both offline and online data can make RL efficient,” ICLR 2023.
>
> [3] Zhang et al., “Conservative Dual Policy Optimization for Efficient Model-Based Reinforcement Learning,” NeurIPS 2022.

---

### Official Review · Reviewer_5Ycb · 2025-10-31

**Soundness:** 3
**Presentation:** 3
**Contribution:** 3
**Rating:** 6
**Confidence:** 4

**Summary:**

The paper proposes an interesting and novel approach with solid mathematical proof and promising empirical performance.
The paper proposes cross‑domain Bellman consistency $\delta$ and QAvatar method. By learning a internal mapping,  the $Q_{src}$ aligns with the target-domain transitions and use a hybrid critic that interpolates target and mapped‑source Q values with a closed‑form weight $\alpha$. The authors prove an average sub‑optimality bound and show empirical improvements on tasks with different state/action dimensionalities, including locomotion, manipulation, and navigation.

**Strengths:**

1. Clean problem framing for CDRL with mismatched spaces; explicit transferability notion via a Bellman‑style residual
2. Hybrid critic that can down‑weight a poor source, limiting negative transfer
3. A formal tabular NPG analysis that separates learning progress and approximation error
4. The experimental evaluation is comprehensive and convincing, covering diverse tasks and challenging transfer scenarios

**Weaknesses:**

1. The theory-practice gap is substantial: the convergence analysis assumes on-policy, tabular NPG, while the implementation is off-policy, deep RL with replay buffers and twin Q-networks. The critical assumptions of the bound (e.g., on-policy error norms) do not hold in the implemented algorithm.

2. The claim of being “hyperparameter-free” is overstated: the update frequency $N_\alpha$ for the adaptive weight acts as a hidden hyperparameter, and no ablation study is provided.

3. The loss to update the mapping $\phi, \psi$ Eqn 5 (unquared) and Eqn 73 (squared) are mismatch

4. mior format issue: The title of tables houle be ahead of the table

**Questions:**

Same as weakness

additional questions:

Can you add ablation test regards to the update coefficient $N_\alpha$? Is the performance sensitive to the udpate frequency?

---

> ### Author Response · Authors · 2025-11-23
> **Response to Reviewer 5Ycb (1/2)**
>
> We thank the reviewer for the insightful and constructive feedback. We address each concern point-by-point below and have revised the paper accordingly.
>
> **[Q1]: Regarding the differences between theory and practice**
> > The theory-practice gap is substantial: the convergence analysis assumes on-policy, tabular NPG, while the implementation is off-policy, deep RL with replay buffers and twin Q-networks. The critical assumptions of the bound (e.g., on-policy error norms) do not hold in the implemented algorithm.
>
> Thank you for pointing out this important issue. We’d like to clarify that while our theoretical analysis is presented in the on-policy tabular NPG setting, **the same analysis can be directly extended to the off-policy RL setting used in our implementation, with only minimal changes to the proof**.
>
> Specifically: The main change lies in the data distribution:
>
> - Instead of the $d^{\pi^{(t)}}$ induced by the current policy $\pi^{(t)}$, the data distribution is $d^{\pi_{\beta}^{(t)}}$ determined by a *behavior policy* $\pi_{\beta}^{(t)}$. In practice, $d^{\pi_{\beta}^{(t)}}$ can be the data distribution induced by some exploratory policy or corresponds to a probabilistic mixture of past distributions $\lbrace d^{\pi^{(t)}} \rbrace_{t}$ induced by the replay-buffer sampling.
>
> - Under this off-policy variant, the proof only requires modifying the coverage assumption in Lemma 5. Specifically, instead of assuming
>
>
> $$\left\lVert \frac{d^{\pi^*}}{d^{\pi^{(t)}}} \right\rVert_{\infty} \le C$$,
> we assume that
>
>
> $$
> \left\lVert \frac{d^{\pi^*}}{d^{\pi_{\beta}^{(t)}}} \right\rVert_{\infty} \le C_\beta.
> $$
>
> and the remaining steps follow by substituting $d^{\pi^{(t)}}$ with $d^{\pi_{\beta}^{(t)}}$. As a result, in the off-policy case, the hybrid-critic weight $\alpha_t$ is naturally defined with respect to $d^{\pi^{(t)}_{\beta}}$, which is consistent with our implementation where $\alpha(t)$ is computed from replay-buffer samples.
>
> This discussion is also provided in Corollary 2 in Appendix G of the paper to strengthen the connection between the theoretical analysis and the practical implementation for the readers.
>
> **Regarding the practical implications of our theory**: While our convergence results are established in the tabular NPG setting, our analysis indeed offers very useful and directly applicable insights to practical cross-domain RL:
>
> - **Cross-domain Bellman consistency**: The form of cross-domain Bellman loss is completely motivated and derived from the convergence analysis. This notion of consistency serves as a practical way to measure domain discrepancy and leads to a loss function readily compatible with deep RL.
>
> - **Hybrid critic with an adaptive weight $\alpha(t)$**: Our Proposition 3 directly indicates that using the hybrid critic (i.e., a combination of source and target Q functions) with the weight $\alpha(t)$ determined adaptively by the two error terms can lead to effective transfer. This insight still applies in the deep RL setting, as also verified by the training curves and $\alpha(t)$ in Figures 3 and 4.
>
> More broadly, while the settings in our theoretical analysis do not hold verbatim in deep RL (as is common in cross-domain RL and also RL in general [1-5]), the theory still provides meaningful structural guidance for practical algorithm design, especially on clarifying (1) **what** should be transferred and (2) **how** transfer should be controlled.
> We therefore believe that our theoretical results remain valuable to the RL community precisely because they yield insights that enhance practical algorithms.
>
> [1] Jiafei Lyu, Chenjia Bai, Jingwen Yang, Zongqing Lu, and Xiu Li, “Cross-Domain Policy Adaptation by Capturing Representation Mismatch,” ICML 2024.
>
> [2] Kang Xu, Chenjia Bai, Xiaoteng Ma, Dong Wang, Bin Zhao, Zhen Wang, Xuelong Li, and Wei Li, “Cross-Domain Policy Adaptation via Value-Guided Data Filtering,” NeurIPS 2023.
>
> [3] Benjamin Eysenbach, Swapnil Asawa, Shreyas Chaudhari, Sergey Levine, Ruslan Salakhutdinov, “Off-Dynamics Reinforcement Learning: Training for Transfer with Domain Classifiers,” ICLR 2021.
>
> [4] Marc G. Bellemare, Will Dabney, Rémi Munos, “A Distributional Perspective on Reinforcement Learning,” ICML 2017.
>
> [5] Scott Fujimoto, Herke van Hoof, and David Meger, “Addressing Function Approximation Error in Actor-Critic Methods,” ICML 2018.

---

> ### Author Response · Authors · 2025-11-23
> **Response to Reviewer 5Ycb (2/2)**
>
> **[Q2]: Ablation test on $N_{\alpha}$**
> > Can you add an ablation test with regards to the update coefficient $N_{\alpha}$? Is the performance sensitive to the update frequency?
>
> Thank you for the question. In our implementation, $\alpha(t)$​ is recomputed periodically, and $N_{\alpha}$ only determines how frequently this closed-form update is performed. As long as $N_{\alpha}$​ is not too large (which would delay updates) or too small (which may cause $\alpha$ to fluctuate too frequently and lead to unstable learning), the overall learning behavior remains largely unaffected.
>
> To validate this, we evaluated three choices of update interval, $N_{\alpha}=300,1000, 3000$, in both the Cheetah and Table Wiping environments using 5 random seeds. The resulting performance curves and learned $\alpha(t)$​ trajectories are shown at: https://imgur.com/a/GRN1y5C.
>
> Across all settings, we observe that:
>
> (1) **the learned $\alpha_t$​ trajectories remain highly similar**.
>
> (2) **the performance exhibits only mild sensitivity**.
>
> None of the settings cause degradation or instability, suggesting that the method is reasonably robust to the choice of $N_{\alpha}$​ within this range.
>
> **[Q3]: Regarding the cross-domain Bellman loss**
> > The loss to update the mapping $\phi, \psi$ Eqn 5 (unsquared) and Eqn 73 (squared) are mismatch.
>
> Thank you for pointing this out. Equation (73) is correct. Equation (5) contains a typographical error, and the correct form should use the squared loss. This design is similar to the TD loss of the target Q-function in Equation (1). This correction does not affect the validity of our theorem, because all of our proofs presume that both the TD loss and the cross-domain Bellman loss take the form of squared loss. We have fixed this in the updated manuscript.
>
> **[Q4]: Formatting issue**
> > Minor format issue: The title of tables should be ahead of the table.
>
> Thank you for catching this. We have fixed this in the updated manuscript.

---

### Official Review · Reviewer_dbj9 · 2025-11-01

**Soundness:** 3
**Presentation:** 3
**Contribution:** 2
**Rating:** 6
**Confidence:** 2

**Summary:**

This paper addresses the problem of cross-domain reinforcement learning (CDRL) where the source and target domains have different state and action spaces. The authors identify two key challenges: (1) learning the mapping between these distinct spaces, and (2) avoiding "negative transfer" when the source-domain model is not actually beneficial for the target task. The core ideas include cross-domain Bellman Consistency, hybrid critic, hyperparameter-free weighting, and mapping via Bellman Loss.

The authors provide a theoretical analysis in the tabular NPG setting to show that QAvatar achieves a good sub-optimality bound that gracefully degrades to the standard NPG bound in cases of negative transfer. They then present a practical implementation using SAC and normalizing flows. Experiments on MuJoCo, Robosuite, and Safety-Gym tasks show that QAvatar significantly outperforms other CDRL baselines.

**Strengths:**

- The central concept of using a hybrid critic weighted by a measure of "transferability" is very clever. The automatic, hyperparameter-free weighting scheme, which pits the target TD error against the cross-domain Bellman error, is an elegant solution to the problem of negative transfer.
- Learning the state-action mappings by minimizing the cross-domain Bellman loss (a reward- and dynamics-aware objective) is a strong alternative to unsupervised, dynamics-agnostic methods like cycle-consistency. The toy example in Appendix D.1 provides a good motivation for this.
- The method shows consistent and significant sample efficiency improvements over all baselines (SAC, FT, PAR, CAT, CMD) across a good variety of cross-domain tasks (locomotion, manipulation, navigation) with differing state/action spaces.

**Weaknesses:**

- The practical implementation of QAvatar is very complex. It requires running a full SAC algorithm while also training two normalizing flow models for the state and action mappings, and also calculating two separate Bellman errors (target and cross-domain) at every step (or every $N_\alpha$ steps) to compute the weight. The paper notes this results in "about twice the training time of SAC," which is a major practical limitation and cost.
- The paper uses normalizing flows for the mappings, which works for the low-dimensional state/action spaces in MuJoCo and Robosuite. It is not clear how this approach would scale to high-dimensional state spaces, such as images, where learning a normalizing flow is significantly more challenging.
- Stability of Mapping Loss: The mapping functions $(\phi, \psi)$ are trained to minimize a crossdomain Bellman loss (Eq. 5) that depends on both the fixed $Q_{s r c}$ and the changing target policy $\pi^{(t)}$. This creates a complex, non-stationary optimization problem for the mappings. It's unclear how stable this is, especially if $Q_{s r c}$ is sub-optimal or the policy is exploring.

**Questions:**

- The computational cost of doubling training time (per Section 6) is a significant drawback. Could you discuss the trade-off? At what point does the 2x computational cost per step outweigh the ~2x sample efficiency gain?
- The mapping functions are trained to minimize a Bellman loss dependent on $Q_{\text {src }}$. What happens if $Q_{s r c}$ is sub-optimal? Does the loss find the "correct" physical mapping, or does it find a distorted mapping that simply makes the sub-optimal $Q_{\text {src }}$ look consistent with the target dynamics?

---

> ### Author Response · Authors · 2025-11-23
> **Response to Reviewer dbj9 (1/3)**
>
> We thank the reviewer for the insightful and constructive feedback. We address each concern point-by-point below and have revised the paper accordingly.
>
> **[Q1]: Complexity of QAvatar’s implementation**
> > The practical implementation of QAvatar is very complex. The computational cost of doubling training time (per Section 6) is a significant drawback. Could you discuss the trade-off? At what point does the 2x computational cost per step outweigh the ~2x sample efficiency gain?
>
> Thank you for raising this important point. We clarify both the computational trade-off and the conditions under which the additional cost outweighs the sample-efficiency gains.
>
> **Complexity comparison**
>
> Although QAvatar introduces several auxiliary modules, its architectural complexity is comparable to existing CDRL baselines.
>
> - QAvatar uses 1 source model plus 4 modules (two normalizing flows and two decoders), and incurs an additional cross-domain Bellman loss and periodic computation of $\alpha$.
>
> In contrast:
>
> - CAT also introduces 1 source model plus 4 auxiliary networks (state/action encoders and reverse state/action encoders), trained with mutual information, cycle-consistency, and correction losses.
>
> - CMD requires 1 source model plus 5 auxiliary networks (state/action generative alignment modules, state/action discriminators, and a dynamics model), together with adversarial losses and next-state consistency losses.
>
> Thus, QAvatar is not more complex than other CDRL methods, and in fact avoids adversarial training, which is often unstable and more computationally expensive.
>
> **Trade-off discussion**
>
> **The key motivation of CDRL is that target-domain data collection is expensive**, which has been widely assumed in prior work [1-3]. Under this setting, a moderate increase in computation per gradient step is typically acceptable if it yields substantial reductions in real-world interaction cost. If the environment allows cheap or unlimited data collection (e.g., simulated domains where rollout is extremely fast), but each model update step is very expensive (e.g., very large foundation models), then the trade-off reverses. In such settings, the extra networks and losses may no longer be worthwhile, and vanilla RL may be preferable.
>
> [1] Xu et al., “Cross-domain policy adaptation via value-guided data filtering.”, Neurips 2023.
>
> [2] Zhu et al., “Cross Domain Policy Transfer with Effect Cycle-Consistency.”, ICRA 2024.
>
> [3] Lyu et al., “Cross-Domain Policy Adaptation by Capturing Representation Mismatch”, ICML 2024.
>
> **[Q2]: Stability of Mapping Loss**
> > The mapping functions $(\phi,\psi)$ are trained to minimize a cross-domain Bellman loss (Eq. 5) that depends on both the fixed $Q_{src}$ and the changing target policy $\pi^{(t)}$. This creates a complex, non-stationary optimization problem for the mappings. It's unclear how stable this is, especially if $Q_{src}$ is sub-optimal or the policy is exploring.
>
> We would like to clarify that the non-stationarity highlighted in the question is common in general RL and also appears in standard actor–critic algorithms such as SAC, where the critic is learned under a continually changing policy. Training the mapping functions $(\phi,\psi)$ in our framework is analogous: both are optimized in the presence of policy exploration and policy-induced non-stationarity. Importantly, however, the source critic $Q_{src}$​ respects the Bellman structure of the source domain, which provides a stable reference. When the source and target domains are reasonably similar, learning $(\phi,\psi)$ can in practice be easier than learning a full target-domain critic from scratch.
>
> To support this, we report the target-domain TD loss and the cross-domain Bellman loss from our main experiments. Both curves are smoothed with EWMA and shown in the following link: https://imgur.com/a/jbsNAUO.
>
> Across all environments, the cross-domain Bellman loss remains consistently low and stable compared to the TD loss, indicating that the learned mappings stay well-aligned with the target domain throughout training.

---

> ### Author Response · Authors · 2025-11-23
> **Response to Reviewer dbj9 (2/3)**
>
> **[Q3]: Regarding normalizing flows and how QAvatar scales to high-dimensional state spaces like images**
> > The paper uses normalizing flows for the mappings, which works for the low-dimensional state/action spaces in MuJoCo and Robosuite. It is not clear how this approach would scale to high-dimensional state spaces, such as images, where learning a normalizing flow is significantly more challenging.
>
> Thank you for the thoughtful question. In image-based RL, it is standard to learn an image encoder that maps raw observations into a low-dimensional feature space [6-7]. When the source model is trained from images, one can simply encode each source observation and train the normalizing flow in this latent feature space rather than on raw pixels. Importantly, the encoder’s feature dimensionality in modern image-based RL is typically low, which is orders of magnitude smaller than the raw image space [8-9].
> Therefore, **training a normalizing flow in this feature space is entirely tractable and does not introduce any scalability issues**. This modification does not affect the QAvatar framework, because the source model is fixed during target-domain transfer.
>
> To further demonstrate this, we additionally evaluated QAvatar on image-based continuous-control tasks in the DeepMind Control Suite (DMC) [4]. Each DMC observation consists of a stack of three 84×84 RGB frames with an action repeat of 4.
> Below, we summarize the SAC setup, flow-model training, and transfer protocol.
>
> **SAC**
>
> For SAC, both the actor and critic use 3 hidden layers with 1024 units each. The image encoder follows the IMPALA [5] architecture to extract low-dimensional visual features. All remaining hyperparameters are the same as those used in Stable-Baselines3.
>
> During training, each frame is padded with 4 pixels on all sides, followed by a random 84×84 crop (allowing a translation of up to $\pm$ 4 pixels), which is the standard procedure in image-based RL. The cropped frame is then passed through the encoder to obtain its feature representation, which is used as the state input to the SAC algorithm; the remainder of training follows standard SAC.
>
> **Flow Model Training**
>
> Training a flow model directly on the raw high-dimensional image observations is challenging. Therefore, we first pass each image stack through the source encoder to obtain its feature representation, and train the flow model to match the distribution of these extracted features rather than the raw images. Importantly, this modification does not alter the QAvatar framework, since the source model remains fixed during target-domain transfer.
>
> **QAvatar and Cross-Domain Transfer**
>
> For cross-domain experiments, we consider two transfer scenarios:
> - The source model is obtained by training an SAC agent on the **walker_walk** task for both scenarios.
> - The target tasks are **walker_run** and **cheetah_run**, respectively.
>
> To leverage the source critic for transfer, each target image observation is first passed through the target encoder to obtain its feature representation, which is then used as the input to the state decoder; the rest of the procedure follows the standard QAvatar framework.
>
> **Experimental Results**
>
> Results averaged over 5 random seeds are available at [ https://imgur.com/a/Up1zeat ]
>
> These indicate that QAvatar achieves substantially higher performance than SAC trained from scratch on both target tasks, and notably, QAvatar succeeds even when SAC struggles to learn effectively on cheetah_run.
>
> Finally, if one must handle a setting where the source state must remain in a high-dimensional form, the flow model can be replaced with diffusion-based or other SoTA generative models, while preserving the core structure of QAvatar, since the purpose of the normalizing flow in QAvatar is to generate samples from the source state distribution.
>
> [4] Tassa et al., “DeepMind Control Suite,” arXiv 2018.
>
> [5] Espeholt et al., “IMPALA: Scalable Distributed Deep-RL with Importance Weighted Actor-Learner Architectures,” ICML 2018.
>
> [6 ] Yarats et al., "Image augmentation is all you need: Regularizing deep reinforcement learning from pixels,” ICLR 2021.
>
> [7] Yuan et al., "Pre-Trained Image Encoder for Generalizable Visual Reinforcement Learning
> ," Neurips 2022.
>
> [8] Ghugare et al., "Simplifying Model-based RL: Learning Representations, Latent-space Models, and Policies with One Objective," ICLR 2023
>
> [9] Laskin et al., "CURL: Contrastive Unsupervised Representations for Reinforcement Learning," ICML 2020.

---

> ### Author Response · Authors · 2025-11-23
> **Response to Reviewer dbj9 (3/3)**
>
> **[Q4]: Explain the effect of $Q_{src}$ on the learned mapping functions**
> > The mapping functions are trained to minimize a Bellman loss dependent on $Q_{src}$. What happens if $Q_{src}$ is sub-optimal? Does the loss find the "correct" physical mapping, or does it find a distorted mapping that simply makes the sub-optimal $Q_{src}$ look consistent with the target dynamics?
>
> Thank you for the thoughtful question. We respectfully interpret “correct physical mapping’’ as referring to the mappings that would arise if the source model $Q_{src}$ were optimally trained. If this interpretation differs from what the reviewer intended, we sincerely apologize and would appreciate further clarification.
>
> We fully agree that if the $Q_{src}$ is sub-optimal, the learned mappings may not coincide with the “correct physical mapping.” However, this does not mean that the mappings are poorly learned. By minimizing the cross-domain Bellman loss, the learned mappings adapt so that $Q_{src}$ becomes consistent with the target-domain Bellman equation.
> Crucially, when the cross-domain Bellman error is smaller than the TD-error of the target critic, $Q_{src}$ offers a more reliable update direction for policy improvement than the target critic, even under not “correct mapping’’.
> We also highlight that in our main experiments, each source model is trained for 1M steps per seed. As a result, some source models are only of medium quality. Nevertheless, the empirical results consistently show that such source models still provide positive benefits during target-domain learning.

---

### Official Review · Reviewer_hvoW · 2025-11-03

**Soundness:** 3
**Presentation:** 3
**Contribution:** 2
**Rating:** 6
**Confidence:** 3

**Summary:**

The paper introduces QAvatar, a hybrid-critic framework for cross-domain reinforcement learning (CDRL), which targets transfer scenarios where source and target domains possess distinct state and/or action spaces mismatch and the transferability of source models is hard to measure. Experiments demonstrate substantial empirical improvements of QAvatar on diverse RL benchmarks with both positive and negative transfer cases.

**Strengths:**

This paper proposes the formal definition of cross-domain Bellman consistency and its incorporation into the analysis of transferability represents a strong conceptual advance over prior methods.

The paper presents a detailed theoretical analysis with proofs supporting the soundness of QAvatar’s adaptive weighting scheme.

**Weaknesses:**

Although the hybrid critic can theoretically downweight poor source Q-functions, the effectiveness of knowledge transfer fundamentally depends on the quality of the state/action mapping functions $\phi$ and $\psi$. When mapping is highly ambiguous or misaligned, both the source-supplied and hybrid Q-values may be misleading.

While this paper classifies major approaches and discusses adversarial alignment and cycle consistency, the comparison or discussion with alternative alignment strategies (e.g., [1-3]) is lacking.

It is hard to tell how the proposed method would perform in high-dimensional practical tasks.

There is little discussion regarding the challenges posed by partial observability or scaling to dozens of source-target pairs.

[1] Cross Domain Policy Transfer with Effect Cycle-Consistency.

[2] Cross-Domain Policy Transfer by Representation Alignment via Multi-Domain Behavioral Cloning.

[3] Cross-Domain Knowledge Transfer for RL via Preference Consistency.

**Questions:**

In Figure 1, while QAvatar outperforms in terms of sample efficiency, the gap with FT or CAT-SAC is at times moderate in locomotion (HalfCheetah task). A deeper dissection of where relative benefits are highest would be useful.

Can the authors provide empirical results or qualitative diagnostics assessing when the learned mappings $\phi$ and $\psi$ are well aligned?  What if the mapping is misleading, would the proposed method avoid collapse?

Have the authors tested QAvatar’s hybrid critic and mapping approach on image-based RL or other high-dimensional observation settings? If not, do they foresee major obstacles?

---

> ### Author Response · Authors · 2025-11-23
> **Response to Reviewer hvoW (1/2)**
>
> We sincerely thank the reviewer for the thoughtful and constructive feedback. We address each concern point-by-point below.
>
> **[Q1]: Regarding the quality of learned state/action mapping functions $\phi$ and $\psi$**
> > Although the hybrid critic can theoretically downweight poor source Q-functions, the effectiveness of knowledge transfer fundamentally depends on the quality of the state/action mapping functions $\phi$ and $\psi$. Can the authors provide empirical results or qualitative diagnostics assessing when the learned mappings $\phi$ and $\psi$ are well aligned?
>
> Thank you for the insightful question. We agree that the quality of the learned mapping functions $\phi$ and $\psi$ is critical for successful knowledge transfer. In our framework, a key diagnostic for assessing alignment quality is the **cross-domain Bellman error**. When this error approaches zero, it implies that $Q_{src}$ is $\delta$-Bellman-consistent with a sufficiently small $\delta$ under the learned mappings, indicating that $\phi$ and $\psi$ are well aligned.
>
> To substantiate this, we provide curves of cross-domain Bellman error versus the TD error of the target critic across all main experiments (see: https://imgur.com/a/jbsNAUO). The results consistently show that the cross-domain Bellman error remains low relative to the TD error when the learned mappings align well with the target domain.
>
> > What if the mapping is misleading, would the proposed method avoid collapse.
>
> When the mappings become misleading, the cross-domain Bellman error *cannot* be sufficiently reduced and remains significantly larger than the TD error. In such cases, the adaptive weight $\alpha(t)$ becomes small, causing the hybrid critic to rely primarily on the target-domain critic. Consequently, QAvatar naturally degenerates to SAC, preventing collapse and avoiding negative transfer.
>
> To further strengthen this claim, we refer to our ablation where the source domain is “Ant-v3” and the target domain only reverses the locomotion direction. We intentionally fix the mapping to identity, which becomes misleading in this setting. As shown in Figure 3(b) (orange curve), the $\alpha(t)$ remains low, and the agent’s learning behavior effectively falls back to SAC. This demonstrates that even under incorrect mappings, QAvatar remains stable and avoids collapse.
>
> **[Q2]: Comparison with alternative alignment strategies**
> > While this paper classifies major approaches and discusses adversarial alignment and cycle consistency, the comparison or discussion with alternative alignment strategies (e.g., [1-3]) is lacking.
>
> [1] Cross Domain Policy Transfer with Effect Cycle-Consistency.
>
> [2] Cross-Domain Policy Transfer by Representation Alignment via Multi-Domain Behavioral Cloning.
>
> [3] Cross-Domain Knowledge Transfer for RL via Preference Consistency.
>
> Thank you for the constructive suggestion. We now provide a clearer comparison with [1–3].
>
> **Regarding [1]**:
>
> - [1] operates under the same **unsupervised** cross-domain assumption as DCC and CMD, where the agent has no access to target-domain rewards. This makes the problem fundamentally more challenging than the supervised CDRL setting (i.e., with target-domain reward signal) considered in our work. Building on the DCC objective, [1] further introduces effect cycle-consistency to learn the mapping functions.
> - We additionally evaluated [1] on Cheetah, Ant, Door Opening, and Table Wiping, with results available at: https://imgur.com/a/KupIanX. As shown in these plots, although [1] produces more stable alignment than CMD, its overall performance remains significantly below SAC and other supervised CDRL baselines, which is consistent with the inherent limitations of the unsupervised CDRL methods.
>
> **Regarding [2] and [3]**:
>
> Both methods rely on supervision signals that are fundamentally different from those available in our setting:
>
> - [2] studies multi-task imitation learning and requires expert demonstrations from both the source and target domains, which provides much richer supervision than what is assumed in our problem. In contrast, our method assumes no prior knowledge from the target domain beyond online RL interaction.
>
>
> - [3] formulates cross-domain transfer as preference-based RL, where the true reward function is not observable. Instead, a preference model indicates which target-domain trajectories are preferred. This method builds upon the dynamics cycle-consistency framework used in DCC, CMD, and [1], but adds additional supervision through preference labels. This assumption is orthogonal to ours, as the true reward in our target environment is directly available.
>
>
> In summary, [2-3] rely on supervision signals (expert demos or preference models) that are not available in our learning setup. Therefore, these approaches are not directly comparable to our supervised CDRL setting.

---

> ### Author Response · Authors · 2025-11-23
> **Response to Reviewer hvoW (2/2)**
>
> **[Q3]: Explain when relative benefits of QAvatar are highest**
> > In Figure 1, while QAvatar outperforms in terms of sample efficiency, the gap with FT or CAT-SAC is at times moderate in locomotion (HalfCheetah task). A deeper dissection of where relative benefits are highest would be useful.
>
> Thank you for the suggestion. Both CAT (or CAT-SAC) and FT can be viewed as parameter-based transfer methods that **assume the source and target domains share similar feature representations**. These methods can enable efficient transfer when this assumption holds. However, this assumption often does not hold in general transfer scenarios. In contrast, our approach minimizes the cross-domain Bellman loss by learning inter-domain mappings $\phi$ and $\psi$ that enforce Bellman consistency between the source and target domains. This ensures that $Q_{src}$ aligns with the target-domain Bellman equation, allowing $Q_{src}$ to provide a more accurate policy update direction for the target learner.
>
> **[Q4]: QAvatar on image-based control tasks**
> > Have the authors tested QAvatar’s hybrid critic and mapping approach on image-based RL or other high-dimensional observation settings? If not, do they foresee major obstacles?
>
> Thank you for the question. As a generic CDRL method, QAvatar can be readily applied to image-based RL tasks.
>
> To demonstrate this, we additionally evaluate QAvatar on image-based continuous control tasks from the DeepMind Control Suite (DMC) [4]. In DMC, each observation consists of a stack of three 84×84 RGB frames, and we apply an action repeat of 4. Below, we outline the SAC setup, the flow-model training, and the cross-domain transfer protocol.
>
> **SAC**
>
> For SAC, both the actor and critic use 3 hidden layers with 1024 units each. The image encoder follows the IMPALA [5] architecture to extract low-dimensional visual features. All remaining hyperparameters are the same as those used in Stable-Baselines3. During training, each frame is padded with 4 pixels on all sides, followed by a random 84×84 crop (allowing a translation of up to $\pm$ 4 pixels), which is the standard procedure in image-based RL. The cropped frame is then passed through the encoder to obtain its feature representation, which is used as the state input to the SAC algorithm; the remainder of training follows standard SAC.
>
> **Flow Model Training**
>
> Training a flow model directly on high-dimensional image observations is challenging. Therefore, we first pass each image stack through the source encoder to obtain its feature representation, and train the flow model to match the distribution of these extracted features rather than the raw images. Importantly, this modification does not alter the QAvatar framework, since the source model remains fixed during target-domain transfer.
>
> **Cross-Domain Transfer**
>
> For cross-domain experiments, we consider two transfer scenarios:
> - The source model is obtained by training an SAC agent on the **walker_walk** task for both scenarios.
> - The target tasks are **walker_run** and **cheetah_run**, respectively.
>
> To leverage the source critic for transfer, each target image observation is first passed through the target encoder to obtain its feature representation, which is then used as the input to the state decoder; the rest of the procedure follows the standard QAvatar framework.
>
> **Experimental Results**
>
> Results averaged over 5 random seeds (shown in [ https://imgur.com/a/Up1zeat ]) indicate that QAvatar achieves substantially higher performance than SAC trained from scratch on both target tasks, and notably, QAvatar succeeds even when SAC struggles to learn effectively on Cheetah-Run.
>
> [4] Tassa et al., “DeepMind Control Suite,” arXiv 2018.
>
> [5] Espeholt et al., “IMPALA: Scalable Distributed Deep-RL with Importance Weighted Actor-Learner Architectures,” ICML 2018.

---

### Author Response · Authors · 2025-11-28
**Response to AC and All Reviewers**

Dear AC and Reviewers,

We sincerely thank the reviewers for their thoughtful and constructive feedback and appreciate the AC's effort in taking over the assessment of our submission. We have addressed all the concerns and revised the manuscript accordingly with the changes highlighted in blue. Below, we summarize the key updates.

## (1) New experimental results: Image-based control, sensitivity test, and an additional baseline
- **Image-based experiments:** As recommended by Reviewers hvoW and dbj9, we further evaluated QAvatar in image-based environments from the DeepMind Control Suite (DMC) [1] and provide a brief step-by-step description of our implementation. To leverage the $Q_{src}$, we train a flow model to generate the source state-feature distribution rather than raw images, thereby avoiding the difficulty of modeling high-dimensional pixels. This modification does not change the original algorithm or the core principles of QAvatar.
- **Sensitivity test on the choice of $N_{\alpha}$:** Following Reviewer 5Ycb’s suggestion, we conducted a sensitivity analysis on different update intervals ($N_{\alpha}=300, 1000, 3000$), confirming that the $\alpha(t)$ curves and the performance is insensitive to $N_{\alpha}$ within this range.
- **Comparison on alternative alignment strategies:** As suggested by Reviewer hvoW, we additionally compared QAvatar with Effect Cycle-Consistency (ECC) [2] on both locomotion and robot manipulation environments, and QAvatar consistently outperforms ECC. We also clarify why ECC performs substantially worse than other supervised methods: ECC operates in an unsupervised setting where the target-domain reward function is unavailable, which prevents the policy from leveraging target-domain reward signals during training.

We have added all the new experimental results and discussions in Appendix D.4.

## (2) Regarding the differences between theory and implementation

In response to Reviewers 5Ycb and bUDR, we show that Proposition 3 can be directly extended to the off-policy setting with only minor modifications to the proof. The corresponding results and discussion are included in Corollary 2 in Appendix G. We also emphasize that this off-policy version aligns with our practical implementation, such as computing $\alpha(t)$ using data in the replay buffer.
## (3) Conversion between convergence guarantee to sample complexity
Following Reviewer bUDR’s suggestion, we present a feasible way to convert our convergence guarantee into a sample-complexity bound. We also provide the final bound under cross-domain realizability, where $Q_{src}$​​ is 0-Bellman consistent (cf. Definition 4). The results show that, under this condition, QAvatar is always comparable to or better than learning from scratch, and in some cases achieves higher sample efficiency by leveraging the source Q-function. These results are included in Corollary 1, with the full derivation in Appendix B.4.
## (4) The quality and stability of learned state/action mapping functions $\phi$ and $\psi$.

In response to Reviewers hvoW and dbj9, we illustrate the quality and stability of the learned mappings by adding plots comparing the cross-domain Bellman error with the TD error of the target critic across all main experiments, together with additional discussion in Appendix E.4.
The results consistently show that the cross-domain Bellman error remains low relative to the TD error, indicating that the mappings are stable and accurate enough for effective transfer of $Q_{src}$. We also clarify how QAvatar avoids performance collapse when the mappings become misleading and note that this behavior is already evident in our existing ablation study.
## (5) Descriptions of hyper-parameter tuning for CDRL baselines

In response to Reviewer bUDR, we provide detailed descriptions of the hyper-parameter tuning procedures for all CDRL baselines used in the main experiments. These details are included in Appendix F.4 in the updated manuscript. We also discuss why other CDRL methods perform worse than QAvatar, primarily due to whether transfer is applied and, more importantly, how each method implements its transfer mechanism.

## (6) Complexity of QAvatar’s implementation

In response to Reviewer dbj9, we show that QAvatar is no more complex to implement than other CDRL baselines. We also note that data collection is typically far more costly than computation in the context of CDRL, making the use of some extra computes a reasonable trade-off.

We acknowledge that the reviewers can no longer comment on whether the new experiments and explanations fully resolve the earlier concerns. Nonetheless, we believe that the additional experiments and comprehensive clarifications further strengthen our work, and we are confident that they fully demonstrate the merit of this approach.

[1] Tassa et al., “DeepMind Control Suite,” arXiv 2018.

[2] Zhu et al., “Cross Domain Policy Transfer with Effect Cycle-Consistency,” ICRA 2024.

---

### Meta-Review · Area_Chair_iZ25 · 2025-12-06

**Summary:**

This paper investigates cross-domain reinforcement learning (CDRL), where the source and target domains have different state and action spaces. The authors identify two key challenges: (1) learning the mapping between these distinct spaces, and (2) avoiding "negative transfer" when the source-domain model is not actually beneficial for the target task. The core ideas include cross-domain Bellman Consistency, hybrid critic, hyperparameter-free weighting, and mapping via Bellman Loss. Theoretical analysis in the tabular NPG setting is given, and the practical implementation is based on SAC and normalizing flows. Experiments on multiple benchmark tasks show that QAvatar outperforms other CDRL baselines.

The paper received generally positive reviews (3 weak accept, 1 weak reject). The authors also addressed the reviewers' concerns by providing additional numerical and theoretical results. However, there are still some concerns that remain not well-addressed, including algorithm complexity, theory-practice gap, etc. Weighing all the pros and cons of this paper, I'm inclined to vote for acceptance, but encourage the authors to diligently address all the remaining concers from reviewers.

**Reviewer Concerns:**

Some concerns that I think are still not well-addressed:
- Complex practical implementation (Reviewer dbj9)
- Stability of learning mapping functions (Reviewer dbj9, 5Ycb)
- Theory-practice gap (Reviewer 5Ycb, bUDR)
- Overstated "hyperparameter-free" (Reviewer 5Ycb)

Some comments that I believe have been addressed or partly addressed during rebuttal, based on additional experimental results:
- Applicability in high-dimensional state space, e.g., image (Reviewer dbj9)
- Lack of theoretical guarantee (Reviewer bUDR)

**Reviewer Scores:**

I think Reviewer dbj9 and bUDR might consider increasing the scores, due to the additional numerical and theoretical results provided.

---

### Decision · Program_Chairs · 2026-01-26

Accept (Poster)